# Online Two-Stage Submodular Maximization

**Iasonas Nikolaou**[*]
Boston University

**Miltiadis Stouras**[†]
EPFL

**Stratis Ioannidis**[‡]
Northeastern University

**Evimaria Terzi**[§]
Boston University

## Abstract

Given a collection of monotone submodular functions, the goal of Two-Stage Submodular Maximization (2SSM) [Balkanski et al., 2016] is to restrict the ground set so an objective selected u.a.r. from the collection attains a high maximal value, on average, when optimized over the restricted ground set. We introduce the Online Two-Stage Submodular Maximization (O2SSM) problem, in which the submodular objectives are revealed in an online fashion. We study this problem for weighted threshold potential functions, a large and important subclass of monotone submodular functions that includes influence maximization, data summarization, and facility location, to name a few. We design an algorithm that achieves sublinear $(1 - 1/e)^2$-regret under general matroid constraints and $(1 - 1/e)(1 - e^{-k}k^k/k!)$-regret in the case of uniform matroids of rank $k$; the latter also yields a state-of-the-art bound for the (offline) 2SSM problem. We empirically validate the performance of our online algorithm with experiments on real datasets.

## 1 Introduction

Motivated by applications such as dictionary learning [Maas et al., 2011], data sumarization [Lin and Bilmes, 2012], and recommender systems [Yue and Guestrin, 2011, El-Arini et al., 2009, Mirzasoleiman et al., 2016], Balkanski et al. [2016] introduced the following *Two-stage Submodular Maximization* problem (2SSM). Given a ground set $V$ of $n \in \mathbb{N}$ elements, in the first stage, an algorithm selects a set $S \subseteq V$ of size $|S| = \ell \in \mathbb{N}$. In the second stage, a reward function $f : 2^V \to \mathbb{R}_+$ is sampled u.a.r. from a collection $\mathcal{F}$ of monotone submodular functions; the function $f$ is then optimized over subsets of $S$ with size $k < \ell$. The goal of 2SSM is to select the restricted ground set $S$ so that the expected second-stage reward is maximized; formally:

$$\text{Maximize:} \quad F(S) \equiv \mathbb{E}\left[\max_{S' \subseteq S : |S'| \leq k} f(S)\right], \tag{1a}$$

$$\text{subject to:} \quad S \subseteq V, |S| \leq \ell, \tag{1b}$$

where the expectation is over the random function $f \in \mathcal{F}$. Balkanski et al. [2016] proposed an approximation algorithm for 2SSM. Subsequently, Stan et al. [2017] provided a generalization when the second stage optimization is over an arbitrary matroid (rather than a cardinality) constraint.

The 2SSM problem is of significant interest when the second-stage optimization needs to happen in near real-time. For example, if the original ground set $V$ is quite large, restricting the ground truth set prior to the revelation of the objective can yield significant computational dividends. Though applications are numerous (see Appendix A), a canonical example is a recommender system [Stan et al., 2017, Liu et al., 2022a]. In this scenario, $V$ is catalog of items to be recommended to a user cohort. Each user is described by a mononote submodular utility function $f : 2^V \to \mathbb{R}_+$, sampled

---

[*]Equal Contribution, `nikolaou@bu.edu`

[†]Equal Contribution, `miltiadis.stouras@epfl.ch`

[‡]`ioannidis@northeastern.edu`

[§]`evimaria@bu.edu`

| Reference | 2SSM Offline Approx. Ratio | O2SSM Online $\alpha$-Regret | Setting | Objective Function | Constraints |
|---|---|---|---|---|---|
| Balkanski et al. [2016] | $1 - 1/e - 1/k^{1/2-\epsilon} - e^{-\Omega(k^{2\epsilon})}$ | – | Offline | Submodular | Cardinality |
| Balkanski et al. [2016] | $1/2(1 - 1/e)$ | – | Offline | Coverage | Cardinality |
| Stan et al. [2017] | $1/2(1 - 1/e^2)$ | – | Offline | Submodular | Matroid |
| Yang et al. [2021] | $1/(p+1)(1 - e^{-(p+1)})$ | – | Offline | Submodular | $p$-Exchange System |
| Mitrovic et al. [2018] | $1/7$ | – | Streaming | Submodular | Matroid |
| Mitrovic et al. [2018] | $1/4(1 - 1/e^2)$ | – | Distributed | Submodular | Matroid |
| This paper | $(1 - 1/e)(1 - e^{-k}k^k/k!)$ | $\alpha = (1 - 1/e)(1 - e^{-k}k^k/k!)$ | Online+Offline | WTP | Cardinality |
| This paper | $(1 - 1/e)^2$ | $\alpha = (1 - 1/e)^2$ | Online+Offline | WTP | Matroid |

Table 1: Summary of known results for different settings of the Two-Stage Submodular Maximization (2SSM) problem and its online version (O2SSM). Our work is the first to consider (and provide guarantees for) O2SSM. Our online algorithm can be converted to a new offline algorithm for 2SSM under WTP objectives (see Appendix J). For cardinality constraints, the guarantees of both our (offline) algorithm and the one by Balkanski et al. [2016] converge to $1 - 1/e \approx 0.63$ as $k \to \infty$, but our algorithm has a strictly better guarantee for finite $k \in \mathbb{N}$. For general matroids, our guarantee for 2SSM ($(1 - 1/e)^2 \approx 0.40$) is slightly worse than the one by Stan et al. [2017] ($1/2(1 - 1/e^2) \approx 0.43$).

from a known distribution, that captures the value of a recommended set of items to the user. To speed-up recommendations and/or item delivery, the recommender a priori restricts recommendations to occur within a set $S \subset V$ of size $|S| = \ell$ items, where $k < \ell < n$. When users arrive, their utility is revealed, and the recommender suggests $k$ items by solving the corresponding cardinality-constrained optimization problem. Restricting recommendations to $S$ speeds up this computation; moreover, items in $S$ can be, e.g., prefetched (non-realtime) to a cache, thereby also accelerating their delivery.

This offline version of 2SSM, studied by Balkanski et al. [2016] and Stan et al. [2017], assumes prior knowledge of the candidate objectives set $\mathcal{F}$ (and the distribution over it). When set $\mathcal{F}$ is unknown, it is natural to consider an *online* version, in which functions $f \in \mathcal{F}$ are revealed sequentially, and the restricted ground set is adapted online. This *Online Two-Stage Submodular Optimization* (O2SSM) problem poses significant challenges: even though functions $f \in \mathcal{F}$ are submodular, the first-stage objective $F$ in (1a) is not. Hence, O2SSM cannot be solved with existing online submodular maximization techniques [Niazadeh et al., 2021, Matsuoka et al., 2021, Si Salem et al., 2024]. To make matters worse, even when the set $\mathcal{F}$ is known, computing the objective $F$ is itself NP-hard. As a result, any online algorithm solving O2SSM *cannot even assume oracle access to the first-stage objective $F(\cdot)$, even after a sequence of second-stage objectives $f$ has been revealed.*

Our main contribution is to overcome these challenges: we construct a no-regret online algorithm for O2SSM when objectives belong to a specific subclass of submodular functions, namely, Weighted Threshold Potential (WTP) functions [Karimi et al., 2017, Stobbe and Krause, 2010]. This subclass strictly generalizes weighted coverage functions and appears in applications such as recommender systems, team formation, and influence maximization. When $\mathcal{F}$ is a subset of this class, we can construct a polynomial-time online algorithm that attains sublinear regret with respect to an offline approximation algorithm for 2SSM (with hindsight). We consider both general matroid and cardinality constraints for the second stage. In the latter case, our online algorithm yields also an improved approximation guarantee for the offline problem, when the objective is a WTP function (see Table 1).

From a technical standpoint, we resolve the challenge of making online decisions without oracle access to the first-stage objective $F(\cdot)$. We do so by introducing a fractional relaxation of the objective and a corresponding rounding scheme. For Weighted Threshold Potential (WTP) objectives, we prove that this relaxation is concave and that its supergradients are efficiently computable. This enables us to use Online Convex Optimization (OCO) methods [Hazan, 2016], such as Follow the Regularized Leader (FTRL) and Online Gradient Ascent (OGA), to compute fractional restricted ground sets with sublinear regret. We then seek a rounding scheme that converts fractional solutions to integral ones while preserving their expected reward up to a constant factor. We prove that *Randomized Pipage Rounding* [Chekuri et al., 2009] has this property through a contention-resolution–based analysis that exploits submodular dominance, as exhibited the distribution induced by this rounding.

## 2   Related work

**Two-stage Submodular Maximization.** Balkanski et al. [2016] introduced the *Two-stage Submodular Maximization* problem (2SSM) under cardinality constraints and proposed an $(1-1/e-\Omega(1/\sqrt{k}))$-

approximation algorithm. Their algorithm also optimizes a continuous relaxation and applies a basic randomized rounding scheme twice: first, to select the restricted ground set, and then to generate second-stage solutions. Our work departs from [Balkanski et al., 2016] in three ways: (a) we introduce the online variant of the problem, where the objectives arrive sequantially, (b) we generalize the second stage to general matroids and (c) improve both rounding stages: we use randomized pipage rounding [Chekuri et al., 2009] in the first stage (which incurs no loss) and state-of-the-art contention resolution schemes in the second stage [Dughmi, 2020]; this yields improved approximation guarantees when applied to the offline setting (see Table 1 and Appendix J).

Balkanski et al. [2016] also propose a local search algorithm for coverage functions under cardinality constraints that achieves an $(1/2(1 - 1/e))$-approximation. Building on this, Stan et al. [2017] design an $(1/2(1 - 1/e^2))$-approximation algorithm that works for arbitrary monotone submodular functions under general matroid constraints. The latter is the state of the art in the offline setting, though a gap persists between its guarantee and the hardness threshold of $(1 - 1/e)$. Several works have since extended this algorithm to settings with weakly submodular functions [Chang et al., 2023], bounded curvature submodular functions [Li et al., 2023], or knapsack constraints in the first stage [Liu et al., 2022b]. Stan et al. [2017] leave the *online* version of the problem, which we address in this paper, as an open question. Our algorithm achieves a $(1 - 1/e)^2 \approx 0.40$ approximation under general matroid constraints and WTP functions, which is close to the state-of-the-art $(1/2(1 - 1/e^2)) \approx 0.43$ approximation for the offline problem. Notably, our online algorithm is competitive with the best-known approximation algorithm in hindsight, without accessing the objective sequence in advance.

**Streaming Two-Stage Submodular Maximization.** Mitrovic et al. [2018] study the two-stage submodular maximization problem in the *streaming* setting, where the algorithm makes a single sequential pass over the ground set and must irrevocably decide whether to include or discard each element, while using limited memory. They design algorithms that achieve constant-factor approximations under matroid constraints. In contrast, we consider an *online* setting in which the objective functions, not the elements, are revealed sequentially. Note that, in the streaming setting, the algorithm has access to all functions while making its decisions, while the online setting requires making decisions without knowledge of the objectives.

**(One-stage) Online Submodular Maximization.** Several *online* algorithms have been proposed for maximizing submodular functions under various matroid constraints [Streeter et al., 2009, Harvey et al., 2020, Niazadeh et al., 2021, Matsuoka et al., 2021, Si Salem et al., 2024]. There has also been recent work on the online maximization of continuous DR-submodular functions [Chen et al., 2018, Zhang et al., 2019, 2022]. Our work builds on the recent results of Si Salem et al. [2024], who reduce Online Submodular Maximization (OSM) to Online Convex Optimization (OCO) for the class of WTP functions. WTP functions, introduced by Stobbe and Krause [2010], form a broad and important subclass of monotone submodular functions, with applications including influence maximization [Kempe et al., 2003], facility location [Krause and Golovin, 2014], caching networks [Ioannidis and Yeh, 2016, Li et al., 2021], similarity caching [Si Salem et al., 2022], recommender systems [Krause and Golovin, 2014], and weighted coverage functions [Karimi et al., 2017]; we review some of these in Appendix A. Our main point of departure from Si Salem et al. [2024] and, more broadly, OSM, is that the objective in O2SSM is not submodular; in fact, we cannot even assume that we have polytime oracle access to it. Overcoming this obstacle is one of our main contributions.

## 3  Technical Preliminary

**Submodular Functions and Matroids.** Given a ground set $\mathcal{V} = [n] \equiv \{1, \ldots, n\}$, a set function $f : 2^{\mathcal{V}} \to \mathbb{R}_+$ is *submodular* if $f(A \cup \{v\}) - f(A) \geq f(B \cup \{v\}) - f(B)$ for all $A \subseteq B \subseteq \mathcal{V}$ and $v \in \mathcal{V} \setminus B$, and *monotone* if $f(A) \leq f(B)$ for all $A \subseteq B \subseteq 2^{\mathcal{V}}$. A *matroid* is a pair $\mathcal{M} = (\mathcal{V}, \mathcal{I})$, where $\mathcal{I} \subseteq 2^{\mathcal{V}}$, for which the following holds: (*i*) if $B \in \mathcal{I}$ and $A \subseteq B$, then $A \in \mathcal{I}$, (*ii*) if $A, B \in \mathcal{I}$ and $|A| < |B|$, then there exists a $v \in B \setminus A$ s.t. $A \cup \{v\} \in \mathcal{I}$. The elements of $\mathcal{I}$ are called independent sets. The cardinality of the largest independent set is called the *rank* of the matroid $\mathcal{M}$.

The *characteristic vector* $\boldsymbol{x} \in \{0, 1\}^n$ of a set $S \subseteq [n]$ is the vector whose $i$-th coordinate $x_i$ is 1 if and only if $i \in S$. The *matroid polytope* $\mathcal{P}(\mathcal{M})$ is the convex hull of the characteristic vectors of $\mathcal{M}$'s independent sets. For the remainder of the paper, we will identify sets $S \subseteq \mathcal{V} = [n]$ with their characteristic vectors $\boldsymbol{x} \in \{0, 1\}^n$. Hence, we treat matroids $\mathcal{M}$ as subsets of $\{0, 1\}^n$, while

$\mathcal{P}(\mathcal{M}) \subseteq [0,1]^n$. Similarly, with a slight abuse of notation, we refer to functions $f : 2^{\mathcal{V}} \to \mathbb{R}_+$ via their equivalent representations as functions of binary vectors $f : \{0,1\}^n \to \mathbb{R}_+$.

**Two-Stage Submodular Maximization.** In the two-stage submodular maximization setting of Balkanski et al. [2016], we are given a collection $\mathcal{F} = \{f_t\}_{t=1}^T$ of monotone and submodular objective functions over the ground set $\mathcal{V} = [n]$. The goal is to select a *smaller ground set* that yields a high objective value in expectation, when the function to be maximized is selected u.a.r. from $\mathcal{F}$.

Formally, let $\mathcal{X}_\ell \subseteq \{0,1\}^n$ be the uniform matroid of rank $\ell$ over the ground set $\mathcal{V} = [n]$, i.e.,

$$\mathcal{X}_\ell = \left\{ \boldsymbol{x} \in \{0,1\}^n : \sum_{i=1}^n x_i \le \ell \right\}.$$

For $\boldsymbol{x} \in \mathcal{X}_\ell$ and matroid $\mathcal{M}$, let $\mathcal{Y}_\mathcal{M}(\boldsymbol{x}) \subseteq \{0,1\}^n$ be all independent sets in $\mathcal{M}$ that are subsets of $\boldsymbol{x}$, i.e.:

$$\mathcal{Y}_\mathcal{M}(\boldsymbol{x}) = \left\{ \boldsymbol{y} \in \{0,1\}^n : \boldsymbol{y} \in \mathcal{M}, \ \boldsymbol{y} \le \boldsymbol{x} \right\}.$$

Given $f_t \in \mathcal{F}$, we define:

$$F_t(\boldsymbol{x}) = \max_{\boldsymbol{y} \in \mathcal{Y}_\mathcal{M}(\boldsymbol{x})} f_t(\boldsymbol{y}), \quad \text{for } \boldsymbol{x} \in \mathcal{X}_\ell, \tag{2}$$

to be the objective attained by maximizing $f_t$ over the restriction of $\mathcal{M}$ to subsets of $\boldsymbol{x}$. Then, the Two-Stage Submodular Maximization (2SSM) problem can be stated as:

$$\text{Maximize}: \quad F(\boldsymbol{x}) = \sum_{t=1}^T F_t(\boldsymbol{x}), \tag{3a}$$
$$\text{subj. to}: \quad \boldsymbol{x} \in \mathcal{X}_\ell. \tag{3b}$$

In the recommender system example from the introduction, $\mathcal{M} = \mathcal{X}_k$, i.e., the uniform matroid of rank $k \le \ell$, and $\mathcal{F}$ is a set of user utility functions $f_t$, capturing the value of a recommended set to user $t$. Notice that 2SSM generalizes (one-stage) submodular maximization subject to a cardinality constraint, if we set $\ell = k$, and is therefore NP-hard to approximate within a factor better than $1 - 1/e$ [Feige, 1998]. Moreover, even computing $F_t(\boldsymbol{x})$ for a given $\boldsymbol{x}$ is NP-hard, so a polynomial-time algorithm for 2SSM cannot assume oracle access to $F(\cdot)$.[5] Finally, the function $F(\cdot)$ is *not* submodular [Balkanski et al., 2016], and approximation algorithms cannot readily rely on this property. Nevertheless, as discussed in Sec. 2, there exist polytime approximation algorithms for 2SSM.

**Online Convex Optimization (OCO).** In Online Convex Optimization (OCO), an algorithm makes sequential decisions in order to maximize concave reward functions that are revealed in an online fashion. Formally, at each time step $t = 1, \ldots, T$, a decision maker selects an $\boldsymbol{x}_t$ from a compact and convex set $\mathcal{K}$, and then observes a concave reward function $g_t : \mathcal{K} \to \mathbb{R}$. Decision $\boldsymbol{x}_t$ is made via a (possibly randomized) online algorithm $\mathcal{P}_\mathcal{X}(\cdot)$, which takes as input the history of reward functions and decisions before time $t$; that is,

$$\boldsymbol{x}_t = \mathcal{P}_\mathcal{X}(\{(g_\tau, \boldsymbol{x}_\tau)\}_{\tau=1}^{t-1}).$$

The goal of the decision maker is to minimize the regret, defined as:

$$\mathcal{R}_T = \max_{\boldsymbol{x} \in \mathcal{K}} \sum_{t=1}^T g_t(\boldsymbol{x}) - \sum_{t=1}^T g_t(\boldsymbol{x}_t).$$

Intuitively, sublinear regret (a.k.a. no-regret) implies that the cummulative reward of the online algorithm is comparable to the one attained by the optimal algorithm in hindsight. Standard algorithms, such as *Follow-the-Regularized-Leader* (FTRL) [Shalev-Shwartz, 2012] and *Online Gradient Ascent* (OGA) [Zinkevich, 2003] achieve sublinear regret under mild assumptions:

**Theorem 3.1.** *[Hazan, 2016] Let $\{g_t\}_{t=1}^T$ be a sequence of concave functions, each $L$-Lipschitz (with respect to the $\ell_2$ norm) over a convex and compact domain $\mathcal{K}$ with $\ell_2$-diameter $D$. Then, FTRL and OGA achieve a regret bound of $\mathcal{O}\left(DL\sqrt{T}\right)$.*

We provide a detailed description of FTRL and OGA in Appendix H.

---

[5] It is worth noting that 2SSM remains NP-hard even if one has oracle access to $F(\cdot)$.

$\alpha$**- Regret.** In many online combinatorial optimization problems (e.g., [Streeter et al., 2009, Harvey et al., 2020, Gergatsouli and Tzamos, 2022, Christou et al., 2023]), especially those whose offline counterparts are NP-hard, the performance of an online algorithm $\mathcal{P}_{\mathcal{X}_\ell}(\cdot)$ is evaluated relative to the best $\alpha$-approximate solution in hindsight. This is measured through $\alpha$-Regret [Kakade et al., 2007]:

$$\mathcal{R}_{\alpha,T} = \alpha \max_{\boldsymbol{x} \in \mathcal{X}_\ell} \sum_{t=1}^{T} F_t(\boldsymbol{x}) - \sum_{t=1}^{T} F_t(\boldsymbol{x}_t). \tag{4}$$

The goal is then to design online algorithms that attain sublinear (e.g., $\mathcal{O}(\sqrt{T})$) expected $\alpha$-Regret,[6] having thus a cumulative reward competitive with that of an $\alpha$-approximate offline algorithm. This relaxation is necessary in our setting (O2SSM), as achieving sublinear regret for $\alpha = 1$ would imply a polynomial-time solution to the (NP-hard) offline problem.

**Weighted Threshold Potentials.** A *Weighted Threshold Potential (WTP) function* [Stobbe and Krause, 2010, Si Salem et al., 2024] is a submodular function of the form:

$$f(\boldsymbol{y}) = \sum_{j \in \mathcal{C}} c_j \min \{b_j, \boldsymbol{y} \cdot \mathbf{w}_j\}, \quad \text{for } \boldsymbol{y} \in \{0,1\}^n, \tag{5}$$

where $\mathcal{C}$ is an arbitrary index set, $c_j > 0$, $b_j \in \mathbb{R}_{\geq 0} \cup \{\infty\}$ and $\mathbf{w}_j \in \mathbb{R}_{\geq 0}$. For an extensive review of WTPs, we refer to Appendix B of Si Salem et al. [2024]. Given a WTP function $f$, its *concave relaxation* $\widetilde{f}$ is the concave function we obtain by relaxing the domain of $f$ to be $[0,1]^n$, i.e.,

$$\widetilde{f}(\boldsymbol{y}) = \sum_{j \in \mathcal{C}} c_j \min \{b_j, \boldsymbol{y} \cdot \mathbf{w}_j\}, \quad \text{for } \boldsymbol{y} \in [0,1]^n. \tag{6}$$

We will use $\mathcal{F}_{G,M}$ to denote the class of WTP functions as defined in Eq. (5), that have at most $G$ terms ($|\mathcal{C}| \leq G$) and whose parameters are upper bounded by $M$ ($c_j \leq M, b_j \leq M, ||w||_\infty \leq M, \forall j \in \mathcal{C}$).

## 4  Online Two-stage Submodular Maximization (O2SSM)

In this section, we define the *Online Two-stage Submodular Maximization* (O2SSM) problem. Using our running example of a recommender system, we assume that the set $\mathcal{F}$ *is not known a priori, and the user utilities are revealed sequentially.* In turn, the recommender system maintains and adapts its cache of $\ell \in \mathbb{N}$ items as utilities are revealed, via an online algorithm that has access only to the history of utilities and caching decisions.

**The O2SSM Problem.** Formally, at each time step $t \in \mathbb{N}$, a decision maker selects a set $\boldsymbol{x}_t \in \mathcal{X}_\ell$ of size at most $\ell \in \mathbb{N}$ to serve as a (restricted) ground set. After committing to $\boldsymbol{x}_t$, a monotone submodular function $f_t : \{0,1\}^n \to \mathbb{R}_+$ is revealed, and the decision maker accrues reward:

$$F_t(\boldsymbol{x}_t) = \max_{\boldsymbol{y}_t \in \mathcal{Y}_{\mathcal{M}}(\boldsymbol{x}_t)} f_t(\boldsymbol{y}_t), \tag{7}$$

where $\mathcal{M}$ is a matroid (e.g., $\mathcal{M} = \mathcal{X}_k$, for $k < \ell$).

The set $\boldsymbol{x}_t = P_{\mathcal{X}_\ell}(\{(f_\tau, \boldsymbol{x}_\tau)\}_{\tau=1}^{t-1})$ is selected without knowledge of the function $f_t$, by an online algorithm that has only access to past history.

O2SSM poses two significant challenges: first, the reward functions $F_t(\cdot)$ are not submodular, and the problem does not reduce to standard Online Submodular Maximization. Second, to make matters worse, evaluating functions $F_t(\cdot)$ for a given $\boldsymbol{x}_t$ is NP-hard. As a result, *the online algorithm $\mathcal{P}_{\mathcal{X}_\ell}(\cdot)$ cannot assume oracle access to reward function $F_t$, even after $f_t$ has been revealed.* Nevertheless, if functions $f_t$ belong to the WTP class, we can indeed construct a polynomial time online algorithm that attains sublinear $\alpha$-regret, as we show in the next section.

## 5  Our Algorithm

In this section, we present RAOCO (Algorithm 1), an algorithm that achieves sublinear $(1 - 1/e)^2$-Regret for WTP functions under arbitrary matroid constraints on the second stage. When the second-stage matroid is a uniform matroid of rank $k$ (i.e., cardinality constraints), RAOCO achieves an improved guarantee, namely sublinear $(1 - 1/e)(1 - e^{-k}k^k/k!)$-Regret.

---

[6]Where the expectation is w.r.t. any random choices of the online algorithm $\mathcal{P}_{\mathcal{X}_\ell}$.

---

**Algorithm 1** Rounding-Augmented `OCO` (`RAOCO`)

---
1: **Uses:**
2: (1) FTRL or OGA `OCO` policy $\mathcal{P}_{\widetilde{\mathcal{X}}_\ell}$ (see Appendix H)
3: (2) Randomized Pipage Rounding $\Xi : \widetilde{\mathcal{X}}_\ell \to \mathcal{X}_\ell$ (see Appendix I)
4: **for** $t = 1$ to $T$ **do**
5:    $\widetilde{\boldsymbol{x}}_t \leftarrow \mathcal{P}_{\widetilde{\mathcal{X}}_\ell}\left(\widetilde{F}_1, \ldots, \widetilde{F}_{t-1}\right)$               ▷ Compute fractional solution
6:    $\boldsymbol{x}_t \leftarrow \Xi(\widetilde{\boldsymbol{x}}_t)$                     ▷ Round to obtain the integral solution
7:    Play action $\boldsymbol{x}_t$                     ▷ Accrue reward $F_t(\boldsymbol{x}_t)$ (implicitly)
8:    Construct relaxation $\widetilde{F}_t$ from $f_t$           ▷ Function $f_t$ has been revealed
9: **end for**

---

## 5.1 Description of the Algorithm

At a high level, `RAOCO` uses an `OCO` policy (such as FTRL or OGA), which outputs a fractional restricted ground set $\widetilde{\boldsymbol{x}}_t \in \widetilde{\mathcal{X}}_\ell$. It then rounds the fractional ground set $\widetilde{\boldsymbol{x}}_t$ to an integral ground set $\boldsymbol{x}_t \in \mathcal{X}_\ell$ using Randomized Pipage Rounding [Chekuri et al., 2009]. Importantly, we prove that the rounding procedure preserves, in expectation, the quality of the fractional solution produced by the `OCO` policy. We now describe each step of the algorithm in more detail.

**Fractional Relaxation.** We begin by introducing a fractional relaxation of the reward function. Let

$$\widetilde{\mathcal{X}}_\ell = \{\widetilde{\boldsymbol{x}} \in [0,1]^n : \sum_{i=1}^n \widetilde{x}_i \leq \ell\}$$

denote the convex hull of the feasible restricted ground sets $\mathcal{X}_\ell$. Given $\widetilde{\boldsymbol{x}} \in \widetilde{\mathcal{X}}_\ell$ and a matroid $\mathcal{M}$, we also define the corresponding relaxed set of second-stage feasible solutions as

$$\widetilde{\mathcal{Y}}_\mathcal{M}(\widetilde{\boldsymbol{x}}) = \{\widetilde{\boldsymbol{y}} \in [0,1]^n : \widetilde{\boldsymbol{y}} \in \mathcal{P}(\mathcal{M}), \; \widetilde{\boldsymbol{y}} \leq \widetilde{\boldsymbol{x}}\},$$

where $\mathcal{P}(\mathcal{M})$ denotes the matroid polytope associated with $\mathcal{M}$.

Let $f$ be a `WTP` function, as defined in Eq. (5), and let $\widetilde{f}$ be its concave relaxation, as defined in Eq. (6). We then define the *concave relaxation $\widetilde{F}$* of the reward function/first-stage objective $F$ as

$$\widetilde{F}(\widetilde{\boldsymbol{x}}) = \max_{\widetilde{\boldsymbol{y}} \in \widetilde{\mathcal{Y}}_\mathcal{M}(\widetilde{\boldsymbol{x}})} \widetilde{f}(\widetilde{\boldsymbol{y}}), \quad \text{for } \widetilde{\boldsymbol{x}} \in \widetilde{\mathcal{X}}_\ell. \tag{8}$$

As discussed below, we prove that $\widetilde{F}$ is indeed concave and Lipschitz over $\widetilde{\mathcal{X}}_\ell$, and that both $\widetilde{F}$ and its supergradients can be computed in polynomial time. These properties are in stark contrast to the original reward function $\max_{\boldsymbol{y} \in \mathcal{Y}(\boldsymbol{x})} f(\boldsymbol{y})$, whose evaluation is NP-hard.

**Algorithm.** Our algorithm, `RAOCO`, is summarized in Algorithm 1. It operates using two main subroutines: (a) an `OCO` policy $\mathcal{P}_{\widetilde{\mathcal{X}}_\ell}$ that maintains fractional restricted ground sets over $\widetilde{\mathcal{X}}_\ell$, and (b) a rounding scheme $\Xi : \widetilde{\mathcal{X}}_\ell \to \mathcal{X}_\ell$ that maps fractional solutions to integral ones.

At each time step $t > 1$, the algorithm computes a fractional solution $\widetilde{\boldsymbol{x}}_t$ using the `OCO` policy, based on the history of *concave relaxations of reward functions revealed so far*; then, it rounds the fractional solution to an integral ground set via the randomized rounding scheme. Formally,

$$\widetilde{\boldsymbol{x}}_t = \mathcal{P}_{\widetilde{\mathcal{X}}_\ell}\left(\left\{\left(\widetilde{F}_\tau, \widetilde{\boldsymbol{x}}_\tau\right)\right\}_{\tau=1}^{t-1}\right), \quad \text{and} \quad \boldsymbol{x}_t = \Xi(\widetilde{\boldsymbol{x}}_t), \tag{9}$$

where $\mathcal{P}_{\widetilde{\mathcal{X}}_\ell}$ is instantiated either as FTRL or OGA (see Appendix H), and that the rounding scheme $\Xi : \widetilde{\mathcal{X}}_\ell \to \mathcal{X}_\ell$ is Randomized Pipage Rounding [Chekuri et al., 2009] (see Appendix I).

Note that, after selecting the restricted ground set $\boldsymbol{x}_t$, the algorithm observes the `WTP` function $f_t$ and constructs the fractional relaxation $\widetilde{F}_t$ of its corresponding reward function $F_t$; the history of these (concave) functions is then passed to `OCO` policy $\mathcal{P}_{\widetilde{\mathcal{X}}_\ell}$ to produce $\widetilde{\boldsymbol{x}}_{t+1}$. Moreover, as $\Xi$ is randomized, so is `RAOCO`; our characterization of the expected regret will be w.r.t. this randomness.

## 5.2 Main Result

Our main contribution is the following theoretical guarantee of `RAOCO`:

**Theorem 5.1.** *Fix a matroid $\mathcal{M} \subseteq [n]$ and $\ell, k \in \mathbb{N}$. For any sequence of `WTP` functions $\{f_\tau\}_{\tau=1}^T \subseteq \mathcal{F}_{G,M}$, Algorithm 1 runs in polynomial time in $n, G$ and $\log M$ and exhibits sublinear $c_{\mathcal{M}}(1 - 1/e)$-Regret, i.e.,*

$$\mathbb{E}[\mathcal{R}_{c_{\mathcal{M}}(1-1/e),T}] = \mathcal{O}\left(\ell G M^2 \sqrt{nT}\right),$$

*where $c_{\mathcal{M}} = (1 - e^{-k}k^k/k!)$ if $\mathcal{M}$ is a uniform matroid of rank $k$ and $c_{\mathcal{M}} = (1 - 1/e)$ otherwise.*

The proof of Theorem 5.1 relies on two key ingredients: First, we show that the sequence of fractional restricted ground sets computed by the `OCO` policy has sublinear regret in the fractional relaxation of the problem (Lemma 5.2). Second, we show that applying Randomized Pipage Rounding to a fractional restricted ground set yields an integral ground set that, in expectation, admits second-stage solutions with approximately the same value as those in the original fractional ground set. Consequently, the reward obtained from the rounded ground set is approximately the same as that of the original fractional ground set (Lemma 5.4).

We now describe each of these components in more detail and use them in order to prove Theorem 5.1.

**Regret of the fractional `OCO` policy.** Recall that the concave relaxation $\widetilde{F}$, defined in Eq. (8), is a function that maps fractional ground sets to the solution of an optimization problem, whose constraints are dictated by the given fractional ground set. Despite the intricate relation between a fractional ground set and its reward, we prove in Appendix D that $\widetilde{F}$ is concave and Lipschitz over $\widetilde{\mathcal{X}}_\ell$ when the second-stage objective $f$ is a `WTP` function with bounded coefficients. These two key properties allow us to employ the standard regret analysis of `OCO` policies (Theorem 3.1) to show that the sequence of fractional ground sets that the algorithm computes will have no-regret. We state this formally in Lemma 5.2, whose proof is in Appendix D:

**Lemma 5.2.** *Let $\{f_t\}_{t=1}^T \subseteq \mathcal{F}_{G,M}$ be a sequence of `WTP` functions, let $\{F_t\}_{t=1}^T$ be the corresponding reward functions as defined in Eq. (7) and $\{\widetilde{F}_t\}_{t=1}^T$ be the corresponding concave relaxations of the reward functions as defined in Eq. (8). Suppose that $\{\widetilde{\boldsymbol{x}}_t\}_{t=1}^T$ is the sequence of fractional restricted ground sets computed by the `OCO` policy of Algorithm 1. Then,*

$$\max_{\boldsymbol{x} \in \mathcal{X}_\ell} \sum_{t=1}^T F_t(\boldsymbol{x}) - \sum_{t=1}^T \widetilde{F}_t(\widetilde{\boldsymbol{x}}_t) = \mathcal{O}\left(\ell G M^2 \sqrt{n}\sqrt{T}\right).$$

Furthermore, we show that the supergradients of the fractional reward function $\widetilde{F}$ can be computed in polynomial time by solving a linear program. This implies that standard `OCO` policies like FTRL or OGA, which use the supergradient of the reward function to calculate the next response, have polynomial time implementations. In particular, the following lemma holds:

**Lemma 5.3.** *Let $f \in \mathcal{F}_{G,M}$ be a `WTP` function and let $\widetilde{F}$ be the corresponding fractional reward function, as defined in Eq. (8). For any fractional restricted ground set $\widetilde{\boldsymbol{x}} \in \widetilde{\mathcal{X}}_\ell$, both the time to compute $\widetilde{F}$ and the time to compute a supergradient of $\widetilde{F}$ at $\widetilde{\boldsymbol{x}}$ are polynomial in $n, G$ and $\log M$.*

The proof of Lemma 5.3 is in Appendix E.

**Randomized Rounding Bound.** Our next goal is to show that the value of the reward function is preserved, in expectation, when passing from a fractional ground set to an integral one via Randomized Pipage Rounding. Specifically, we aim to prove that optimizing over the rounded ground set yields second-stage solutions whose value closely approximates those obtainable in the fractional domain.

To this end, we design a two-stage randomized rounding procedure, used only in our analysis, that is coupled with the rounding performed by `RAOCO`. Given any fractional second-stage solution, this procedure first produces an intermediate integral solution within the rounded ground set, which potentially violates the matroid constraints, and then applies a contention resolution scheme (CRS) to obtain a feasible solution (for more details on CRSs, see Appendix B). A key technical challenge is that standard CRS guarantees typically rely on the input vector being sampled independently across coordinates, an assumption that does not hold in our setting. However, when the ground set is obtained

via Randomized Pipage Rounding, we prove that the resulting distribution exhibits a submodular dominance property that allows us to apply a CRS with provable approximation guarantees. In particular, the CRS ensures that, in expectation, the quality of the resulting feasible solution remains close to that of the original fractional solution.

We stress that we use this CRS construction purely to characterize the performance of the (first-stage) randomized rounding, and not in the online algorithm itself: the only rounding procedure in `RAOCO` is Randomized Pipage Rounding (see Algorithm 1). We leverage the CRS construction to show that the expected reward from the rounded ground set is within a constant factor of the reward from the fractional solution. This is formalized in Lemma 5.4 and proved in Appendix C.

**Lemma 5.4.** *Let $\widetilde{x} \in \widetilde{\mathcal{X}}_\ell$ be a fractional ground set and $x = \Xi(\widetilde{x})$ be an integral ground set obtained by rounding $\widetilde{x}$ using Randomized Pipage Rounding. Let also $f$ be any `WTP` function, and $F, \widetilde{F}$ be the associated reward function and its concave relaxation (Eq. (7) and (8)) respectively. Then,*

$$\mathbf{E}_\Xi \left[ F(\Xi(\widetilde{x})) \right] \geq c_\mathcal{M} \cdot \left( 1 - \frac{1}{e} \right) \cdot \widetilde{F}(\widetilde{x}),$$

*where $c_\mathcal{M} = 1 - e^{-k} k^k / k!$ if the matroid $\mathcal{M}$ of the second stage is uniform with rank $k$, and $c_\mathcal{M} = (1 - 1/e)$ otherwise.*

Finally, using Lemmas 5.2, 5.3 and 5.4, our main theorem, Theorem 5.1, can be proved as follows.

*Proof of Theorem 5.1.* **Regret Guarantee.** Let $\{F_t\}_{t=1}^T$ be a sequence of reward functions and $\widetilde{F}_1, \ldots, \widetilde{F}_T$ be their concave relaxations (see Eq. (7) and (8) respectively). The expected regret of the Algorithm 1 is

$$\mathbb{E}[\mathcal{R}_{c_\mathcal{M}(1-1/e),T}] = c_\mathcal{M}(1 - 1/e) \cdot \max_{x \in \mathcal{X}_\ell} \sum_{t=1}^T F_t(x) - \sum_{t=1}^T \mathbf{E}_\Xi[F_t(\Xi(\widetilde{x}_t))]$$

$$\overset{\text{Lem. } 5.4}{\leq} c_\mathcal{M}(1 - 1/e) \left( \max_{x \in \mathcal{X}_\ell} \sum_{t=1}^T F_t(x) - \sum_{t=1}^T \widetilde{F}_t(\widetilde{x}_t) \right)$$

$$\overset{\text{Lem. } 5.2}{=} O(\ell G M^2 \sqrt{n} \sqrt{T}).$$

**Computational Complexity.** Each round of `RAOCO` consists of two steps: (1) computing a fractional solution through an `OCO` policy (either FTRL or OGA, see Appendix H for details), and (2) rounding this solution to an integral ground set. In order to run the `OCO` policy, `RAOCO` first computes a super-gradient of the concave relaxation function $\widetilde{F}_{t-1}$ at the previous solution $\widetilde{x}_{t-1}$. This supergradient can be computed in polynomial time in $n, G$ and $\log M$ (Lemma 5.3) by solving a linear program (see Appendix E). Next, depending on the instantiated `OCO` policy, the algorithm either performs a projection onto $\widetilde{\mathcal{X}}_\ell$ (for OGA) which can be done in $\mathcal{O}(n \log n)$ time [Duchi et al., 2008], or solves a strongly convex optimization problem (for FTRL), which can be done in polynomial time in $n$ and $\log M$ using standard first-order or interior point methods [Bubeck, 2015]. Finally, the fractional point obtained from the `OCO` policy is rounded via Randomized Pipage Rounding [Chekuri et al., 2009]. The rounding iteratively selects two fractional coordinates and redistributes their combined mass in a randomized way (for more details see Appendix I). Each such iteration runs in time $\mathcal{O}(n)$ and the algorithm will perform at most $\mathcal{O}(n)$ iterations until all coordinates are integral. Overall, the per-round complexity of `RAOCO` is polynomial in $n, G$ and $\log M$. □

## 6 Experiments

In this section, we evaluate the performance of our algorithm on both real and synthetic datasets. Additional details on each dataset, how it maps to instances of `O2SSM`, our choices of $k, \ell$, and $T$, as well as detailed descriptions of competitor algorithms, can be found in Appendix F.[7]

**Datasets and Problem Instances.** We conduct experiments on seven datasets; five real: `Wikipedia`, `Images`, `MovieRec`, `Influence`, and `HotpotQA`, and two synthetic: `TeamFormation`

---

[7]Our code is available online at https://github.com/jasonNikolaou/online-two-stage-sub-max.

| Dataset | $n$ | $\ell$ | $k$ | $T$ | $m$ | $|C|$ | Description |
|---|---|---|---|---|---|---|---|
| Wikipedia | 407 | 20 | 5 | 88 | 22 | 27.2 | Wikipedia articles representatives |
| Images | 150 | 20 | 5 | 40 | 10 | 20.9 | Image collection summarization |
| MovieRec | 1000 | 10 | 4 | 400 | 100 | 2.5 | Movie recommendations |
| TeamFormation | 100 | 10 | 4 | 50 | 50 | 1.0 | Roster selection |
| Influence | 34 | 8 | 3 | 100 | 100 | 34.0 | Influence maximization |
| Coverage | 100 | 20 | 1 | 500 | 20 | 1.95 | Weighted Coverage |
| HotpotQA | 111,140 | 150 | 4 | 500 | 97,852 | 1 | QA Corpus summarization |

Table 2: Dataset summary: $n$ is the number of elements; $\ell$ is the restricted ground set size; $k$ is the second-stage cardinality; $T$ is the number of time steps; $m$ is the number of objective functions; $|C|$ is the average number of Threshold Potentials per function.

and `Coverage`. The `Wikipedia`, `Images`, and `MovieRec` datasets are drawn from prior work [Balkanski et al., 2016, Stan et al., 2017]. The `Coverage` dataset is constructed adversarially to highlight the differences between one-stage and two-stage submodular optimization (see Appendix F).

Table 2 summarizes the key properties of each dataset.

**Algorithms.** We experiment with different variants of `RAOCO` (Algorithm 1). The variants differ in the `OCO` policy used in the fractional domain; thus, we have `RAOCO-OGA`, `RAOCO-FTRL-L2`, `RAOCO-FTRL-H`, where we use *Online Gradient Ascent (OGA)*, *Follow the Regularized Leader (FTRL)* with $L_2$ regularization and entropy regularization respectively.

**Competitor Algorithms.** We compare our method with the following online competitors: (1) `Random`, which selects a set of $\ell$ elements uniformly at random at each timestep. (2) `One-stage-OGA` (`1S-OGA`), which runs `RAOCO-OGA` as if the second stage cardinality constraint was $k = \ell$, i.e., (one-stage) online `WTP` maximization. We also provide as benchmarks the *offline* algorithms proposed by prior work, i.e., `Continuous-Optimization` (`OFLN-CO`) [Balkanski et al., 2016] and `Replacement-Greedy` (`OFLN-RGR`) [Stan et al., 2017]. Finally, we report the value of the optimal (integral) solution in hindsight (`OPT`).

**Metrics.** For the online algorithms, we measure the cumulative average reward $C_t = \frac{1}{t} \sum_{\tau=1}^{t} F_t(\boldsymbol{x}_t)$. For each dataset, we report the average $C_t$ and its standard deviation over five runs.

**Results.** Figure 1 shows the average $C_t$ over time $t$, for each algorithm and each dataset except `HotpotQA`; the latter, along with additional results with varying $k$ and $\ell$, can be found in Appendix G. In all cases, two-stage variants of `RAOCO` (`RAOCO-OGA`, `RAOCO-FTRL-L2`, `RAOCO-FTRL-H`) outperform or match the one-stage baseline (`1S-OGA`). The performance gap is particularly pronounced on the `Influence`, `Images`, and `Coverage` datasets. In `Coverage`, our algorithms achieve almost double the average reward of `1S-OGA` (see Appendix F). Note that `OFLN-CO` does not apply to `Coverage` since it requires $k > 1$, while $k = 1$ in that case.

## 7 Conclusion

We introduce the `O2SSM` problem and proposed `RAOCO`, a polynomial-time algorithm that achieves sublinear $\alpha$-regret for `WTP` functions under general matroid constraints. Our method leverages online convex optimization over a concave relaxation of the reward and combines it with randomized pipage rounding to obtain integral solutions with strong theoretical guarantees and practical performance.

**Limitations and Future Directions.** Our analysis is restricted to `WTP` objectives and assumes full-information feedback. Extending our framework to general monotone submodular functions and partial-revelation/bandit feedback remains an interesting open direction.

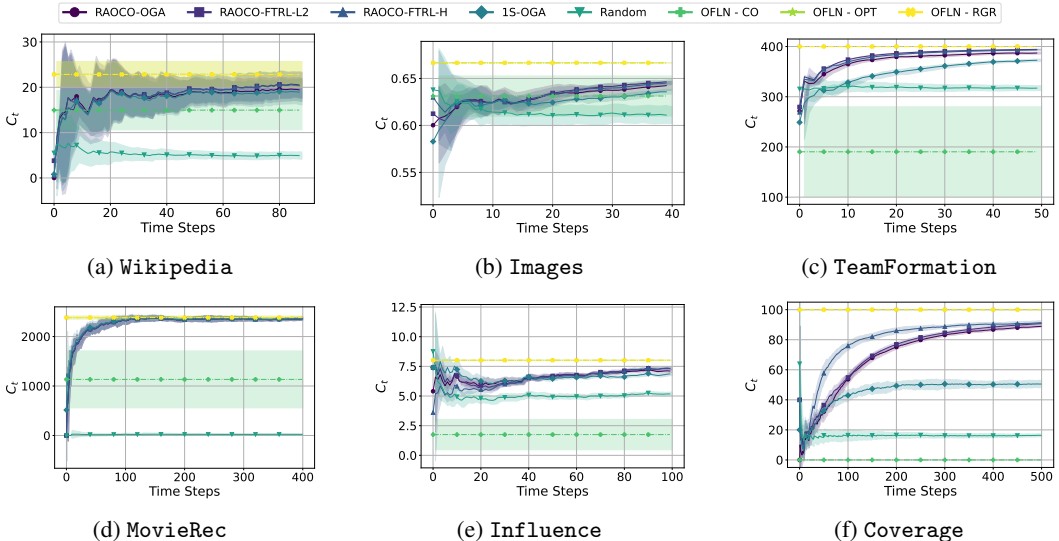

Figure 1: Cumulative average reward $C_t$ of `RAOCO` variants and baselines for each dataset. We report the average value of $C_t$ over 5 runs. The shaded regions correspond to one standard deviation above and below the average. Note that `OFLN-RGR` coincides with `OPT` in all datasets.

## Acknowledgments and Disclosure of Funding

We gratefully acknowledge support to Miltiadis Stouras by the Swiss State Secretariat for Education, Research and Innovation (SERI) under contract number MB22.00054, to Stratis Ioannidis by National Science Foundation grant 2112471, and partial support to Iasonas Nikolaou and Evimaria Terzi through gifts from Microsoft and Google.

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

# A Applications

In this section, we provide several applications of online two-stage submodular maximization (O2SSM), focusing particularly on cases where the second-stage objectives can be modeled via weighted-threshold potential (WTP) functions. Before we present these applications, we first discuss a few abstract objectives that can be cast in WTP form. We then use these objectives to define specific O2SSM applications/instances.

## A.1 Examples of WTP functions

Si Salem et al. [2024] give several instances of submodular maximization whose objective can be expressed as WTP functions. These include influence maximization [Kempe et al., 2003], facility location [Krause and Golovin, 2014, Frieze, 1974], cache networks [Ioannidis and Yeh, 2016, Li et al., 2021], similarity caching [Si Salem et al., 2022], demand forecasting [Ito and Fujimaki, 2016], and team formation [Li et al., 2018]. The objectives of most of these examples are in fact specific cases of weighted coverage, facility location, or quadratic submodular functions. On account of this, we review these three types of objectives in this section, and show that they indeed belong to the WTP class. We refer the interested reader to Si Salem et al. [2024], Karimi et al. [2017], and Stobbe and Krause [2010] for further discussion on WTP functions and applications.

In our description below, we maintain our convention of describing set functions as functions of characteristic (i.e., binary) vectors, i.e., of the form $f : \{0,1\}^n \to \mathbb{R}_+$.

**Weighted Coverage Functions [Karimi et al., 2017].** For $V = [n]$, let $\{S_\ell\}_{\ell \in C}$ be a collection of subsets of $V$, and $c_\ell \in \mathbb{R}_{\geq 0}$ be a weight associated to each subset $S_\ell$. A weighted set coverage function $f : \{0,1\}^n \to \mathbb{R}_+$ receives as input a $\boldsymbol{y} \in \{0,1\}^n$ representing a subset of $V$, and receives value $c_\ell$ if $S_\ell$ is "covered" by an element in $\boldsymbol{y}$; formally,

$$f(\boldsymbol{y}) = \sum_{\ell \in C} c_\ell \min \left\{ 1, \sum_{i \in S_\ell} y_i \right\} = \sum_{\ell \in C} c_\ell \min \left\{ 1, \boldsymbol{y} \cdot \boldsymbol{w}_\ell \right\}, \tag{10}$$

where $\boldsymbol{w}_\ell \in \{0,1\}^n$ is the characteristic vector of $S_\ell$. This is clearly a WTP function of the form (5).

**Facility Location [Krause and Golovin, 2014, Frieze, 1974, Karimi et al., 2017].** In the classic facility location problem, we are given a complete weighted bipartite graph $G(V \cup V', E)$, where $V = [n]$, $E = V \times V'$, with non-negative weights $w_{v,v'} \geq 0$, $v \in V, v \in V'$. We are also given a distibution $\{p_{v'}\}_{v' \in V'}$, where $p_{v'} \geq 0$, for all $v' \in V'$, and $\sum_{v' \in V'} p_{v'} = 1$. The decision maker selects a subset $S \subset V$ and each $v' \in V$ responds by selecting the $v \in S$ with the highest weight $w_{v,v'}$. The goal is to maximize the average weight of these selected edges, i.e. to maximize

$$f(S') = \sum_{v' \in V} p_{v'} \max_{v \in S} w_{v,v'}, \tag{11}$$

given some matroid constraints on $S$. Set $V$ can be considered a set of facilities and $V'$ a set of customers or tasks to be served, while $w_{v,v'}$ is the utility accrued if facility $v$ serves tasks $v'$. The goal is then to maximize the utility in expectation over a random task sampled from distribution $\mathbf{p} \in [0,1]^{|V'|}$), assuming the task is mapped to the best facility in $S$. Karimi et al. [2017] show that this objective is a weighted coverage function, of the form (10), because, for $\boldsymbol{y} \in \{0,1\}^n$ the characteristic vector of $S$, we have

$$\max_{v \in S} w_{v,v'} = \sum_{i=1}^{n-1} \left( w_{\pi_i,v'} - w_{\pi_{i+1},v'} \right) \min \left\{ 1, \textstyle\sum_{j=1}^{i} y_{\pi_j} \right\} + w_{\pi_n,v'} \min \left\{ 1, \textstyle\sum_{j=1}^{n} y_{\pi_j} \right\} \tag{12}$$

where $\pi : V \to V$ is a permutation such that: $w_{\pi_1,v'} \geq w_{\pi_2,v'} \geq \ldots \geq w_{\pi_n,v'}$. Hence, Eq. (11) can be written in WTP form in the same way as Eq. (10).

**Quadratic Submodular Functions [Li et al., 2018, Ito and Fujimaki, 2016].** Consider quadratic monotone submodular functions $f : \{0,1\}^n \to \mathbb{R}_{\geq 0}$ of the form

$$f(\boldsymbol{x}) = \boldsymbol{h}^\intercal \boldsymbol{x} + \frac{1}{2} \boldsymbol{x}^\intercal \boldsymbol{H} \boldsymbol{x}, \tag{13}$$

where, w.l.o.g., $\boldsymbol{H}$ is symmetric and has zeros in the diagonal. Note that $f$ is monotone and submodular if and only if : (a) $\boldsymbol{H} \leq \boldsymbol{0}$, and (b) $\boldsymbol{h} + \boldsymbol{H}\boldsymbol{x} \geq \boldsymbol{0}$, for all $\boldsymbol{x}$ in the feasible domain. We can write the quadratic term as

$$
\begin{aligned}
\boldsymbol{x}^{\intercal}\boldsymbol{H}\boldsymbol{x} &= \sum_{i=1}^{n}\sum_{j=1}^{n} x_i x_j H_{i,j} \\
&= \sum_{i=1}^{n}\sum_{j=1}^{n} (x_i + x_j - \min\{1, x_i + x_j\}) H_{i,j} \\
&= 2\boldsymbol{x}^{\intercal}\boldsymbol{H}\boldsymbol{1} + 2\sum_{i=1}^{n-1}\sum_{j=i+1}^{n} (-H_{i,j})\min\{1, x_i + x_j\},
\end{aligned}
$$

where we used the equality $1 - (1-x)(1-y) = x + y - xy = \min\{1, x+y\}, \forall x, y \in \{0,1\}$. Therefore, we can write $f$ as

$$
f(\boldsymbol{x}) = \min\left\{\infty, (\boldsymbol{h} + \boldsymbol{H}\boldsymbol{1})\cdot\boldsymbol{x}\right\} + \sum_{i=1}^{n-1}\sum_{j=i+1}^{n} (-H_{i,j})\min\{1, x_i + x_j\},
$$

which is a `WTP` function.

### A.2 `O2SSM` Instances and Applications.

**Recommender Systems.** The real-time delivery requirements of modern recommender systems necessitate that they operate in multiple stages [Liu et al., 2022a, Hron et al., 2021, Zheng et al., 2024, Covington et al., 2016]. In particular, upon a user request, a catalog of items to be recommended passes through multiple stages, each thinning the set of recommendation candidates passed on to the next stage. The catalog of items passed to the very first stage is typically restricted heuristically, e.g., through item recency, relation to the recommendation surface, etc. It is exactly the catalog design that passes through the very first stage of the recommendation pipeline that Stan et al. [2017] proposed to optimize via two-stage submodular maximization, presuming knowledge of the distribution of user preferences. As discussed in the introduction, the online version of the problem (`O2SSM`) enables the adaptation of the catalog in an online fashion, without the prior requirement of the distribution of user demand.

The motivation for considering submodular (i.e., diminishing-returns) functions to model user preferences w.r.t. recommendations is natural, and several works model recommender system utilities via submodular functions [Tschiatschek et al., 2017, Nassif et al., 2018, El-Arini et al., 2009, Yue and Guestrin, 2011, Mehrotra and Vishnoi, 2023, Evnine et al., 2024]. Several such models involve `WTP` functions. For example, El-Arini et al. [2009], Tschiatschek et al. [2017] and Yue and Guestrin [2011] model user utilities via weighted coverage functions of the form (10): items recommended "cover" topics or categories $\ell \in C$, and weights $c_\ell$ capture the a user to like that category. In similarity-based recommendations [Haveliwala et al., 2002, Agrawal et al., 1993, Gionis et al., 1999], which often model early-stage rankers [Yan et al., 2022, Yang et al., 2020], queries and items are jointly embedded in the same latent space, and the reward of a recommendated set is the similarity of nearest/most similar item to the user query. This can naturally be modeled via a facility location objective (11), where $v'$ correspond to user/queries, $w_{v,v'}$ is the similarity between their embeddings, and $p'_v$ corresponds to a distribution or weights over queries.

**Team Formation.** In high-reliability settings, such as large-scale software companies or municipal emergency services, organizations must routinely form standby teams capable of reacting to unpredictable, high-stakes events. For instance, a software company may have thousands of engineers, and each day must select a small subset of individuals to be "on-call" in case of operational incidents (e.g., outages, high-severity tickets, etc.). Similarly, emergency response agencies must designate a pool of responders (e.g., firefighters, paramedics, utility crews) to cover incoming incidents throughout the day. Each event or incident, when it occurs, brings specific requirements: geographic proximity or timezone alignment, technical expertise, domain training, or familiarity with particular systems, neighborhoods, or equipment. In both examples, the organization *must commit in advance* to a restricted set of available personnel (i.e., a *roster*) $\boldsymbol{x}_t \in \mathcal{X}_\ell$, from which *a second-stage selection is*

*made once an incident materializes*, i.e., $\boldsymbol{y}_t \in \mathcal{Y}_\mathcal{M}(\boldsymbol{x}_t)$. Since the distribution over incident types is unknown and evolves over time, this naturally constitutes an instance of the O2SSM problem.

Team performance requirements are often modeled in the literature using a weighted coverage function [Nikolakaki et al., 2021, Vombatkere and Terzi, 2023] (see Eq. (10)). In these settings, the utility of a selected team is determined by how well it satisfies task-specific needs, captured by an appropriate bipartite graph. Several works [Lappas et al., 2009, Boon and Sierksma, 2003, Li et al., 2018] model pairwise synergies among team members via quadratic submodular functions (Eq. (13)), where the matrix $\boldsymbol{H}$ encodes compatibility or redundancy among team members. Both cases fall within the WTP class.

**Influence Maximization.**   In social networks and marketing platforms, influence maximization aims to identify a small set of individuals whose activation leads to the largest spread of information, behaviors, or product adoption. A common strategy is to "seed" selected users, e.g. by offering free access, promotional incentives, or early content, so that influence propagates through their social ties. However, running influence maximization over a large-scale network is often computationally prohibitive, particularly when campaigns or objectives evolve over time. A natural approach is to first select a restricted set of influential or cooperative users and later select a campaign-specific subset to activate. Formally, this matches the O2SSM structure: at each round $t$, a platform selects a restricted candidate pool $\boldsymbol{x}_t \in \mathcal{X}_\ell$, from which a seed set $\boldsymbol{y}_t \in \mathcal{Y}_\mathcal{M}(\boldsymbol{x}_t)$ of $k$ users is selected, after observing the influence objective $f_t$ (e.g., a new product or message). Since these objectives arrive sequentially and may reflect different target audiences or goals, the platform must adaptively update $\boldsymbol{x}_t$ based on the influence functions materialized in past rounds, in order to improve the quality of future seed selections.

Classical models of influence propagation, such as the Independent Cascade and Linear Threshold models [Kempe et al., 2003], yield monotone submodular objectives that measure the expected number of activated nodes. In practice, many works study scalable approximations of these models using weighted coverage functions [Borgs et al., 2014, Tang et al., 2014] (see Eq. (10)), where the coverage corresponds to reaching key individuals, groups, or communities within the network. These objectives fall within the WTP class.

## B   Contention Resolution Schemes

### B.1   Background

The framework of Contention Resolution Schemes was formalized by Chekuri et al. [2011] and has since found many applications. Given some feasibility constraints (usually matroid constraints) over some groundset $\mathcal{V}$, a Contention Resolution Scheme (CRS) is an algorithm that accepts a random set $R \subseteq \mathcal{V}$, that is not necessarily feasible, and trims it down to a feasible set $\pi(R) \subseteq R$. The goal of this "rounding" is to ensure that each element is kept with a high enough probability. Most prior work on CRSs has predominantly focused on product distributions, i.e. on random sets $R$ that include each element $e \in \mathcal{V}$, independently, with some probability $x_e$.

In our setting, we aim to use CRS-based roundings to round second-stage solutions (see Appendix C). However, the inclusion of an element in the random set $R$ depends on whether it was included in the rounded groundset by the randomized rounding applied to the first-stage. Since this rounding process (namely Randomized Pipage Rounding) introduces dependencies between elements, the inclusion of elements in $R$ is no longer independent, and standard CRSs for product distributions cannot be directly applied.

Recently, Dughmi [2020, 2022] initiated the study of CRSs under non-product distributions. We build on this line of work and prove that the distribution over sets $R$ induced by our two-stage rounding process satisfies a property known as *submodular dominance* (Definition B.3), which enables the use of CRSs even in the absence of independence (Corollary B.8).

**Notation.**   For the rest of the section, we fix $\mathcal{V} = [n] \equiv \{1, \dots, n\}$ to be a ground set of $n$ elements. Let $\mathcal{M}$ be a matroid over $\mathcal{V}$ and $\boldsymbol{w} \in \mathbb{R}^n_+$ be a non-negative vector. We define the *weighted rank* function $\mathrm{rank}_{\boldsymbol{w},\mathcal{M}} : \{0,1\}^n \to \mathbb{R}_+$, which maps every subset $U \subseteq \mathcal{V}$ to the weight of the maximum-weight independent set of $\mathcal{M}$ that is a subset of $U$. Formally, under our notational convention equating

sets with their characteristic vectors,

$$\text{rank}_{\boldsymbol{w},\mathcal{M}}(\boldsymbol{x}) = \max_{\boldsymbol{y}\in\mathcal{Y}_{\mathcal{M}}(\boldsymbol{x})} \boldsymbol{w}^\top \boldsymbol{y}.$$

When the matroid $\mathcal{M}$ is clear from the context, we will omit it from the subscript and use the notation $\text{rank}_{\boldsymbol{w}}$. We use $\Delta(\{0,1\}^n)$ to denote all possible probability distributions over $\{0,1\}^n$. For any fractional vector $\boldsymbol{x}\in[0,1]^n$, we define the distribution $\mathbf{Ind}(\boldsymbol{x})\in\Delta(\{0,1\}^n)$ which produces a random set $\boldsymbol{z}\sim\mathbf{Ind}(\boldsymbol{x})$ by including every $i\in[n]$, independently, with probability $x_i$. For a distribution $\mathcal{D}\in\Delta(\{0,1\}^n)$, we use $\boldsymbol{\mu}_{\mathcal{D}}\in[0,1]^n$ to denote it's expected value, i.e. $\boldsymbol{\mu}_{\mathcal{D}}=\mathbb{E}_{\boldsymbol{x}\sim\mathcal{D}}[\boldsymbol{x}]$.

We now give a formal definition of a Contention Resolution Scheme.

**Definition B.1** (Contention Resolution Scheme (CRS)). Let $\mathcal{M}=(\mathcal{V},\mathcal{I})$ be a matroid over $\mathcal{V}$. A contention resolution scheme (CRS) $\phi$ is a randomized function from $\{0,1\}^n$ to $\mathcal{I}$, with the property that for all $\boldsymbol{x}\in\{0,1\}^n:\phi(\boldsymbol{x})\leq\boldsymbol{x}$ and $\phi(\boldsymbol{x})\in\mathcal{I}$.

Of course, the above definition by itself is trivially satisfied. For instance, one could define $\phi$ to always return the empty set, which is always feasible but clearly not useful. Intuitively, in CRS-based roundings, we want to retain each element of the input set with a sufficiently high probability. This intuition is captured by the notion of *selectability*, which quantifies the minimum probability with which each element is preserved (conditioned on it being present in the input set).

**Definition B.2** ($\gamma$-selectable CRS). Let $\mathcal{M}=([n],\mathcal{I})$ be a matroid over $[n]$ and $\mathcal{D}\in\Delta(\{0,1\}^n)$ be a distribution over subsets of $[n]$. A contention resolution scheme (CRS) $\phi$ is called $\gamma$-selectable for $\mathcal{D}$, if for any $\boldsymbol{x}\sim\mathcal{D}$ and every $i\in[n]$ it holds that $\mathbf{Pr}_{\boldsymbol{x}\sim\mathcal{D},\phi}[\phi(\boldsymbol{x})_i=1|\boldsymbol{x}_i=1]\geq\gamma$.

To extend the applicability of CRSs beyond product distributions, we need a structural property of distributions that enables approximation guarantees even in the presence of dependencies. One such property is *submodular dominance* [Dughmi, 2020, Qiu and Singla, 2022], which intuitively ensures that a distribution is "at least as good" as its independent counterpart when evaluated against any submodular function. We give the formal definition below.

**Definition B.3** (Submodular dominance). A distribution $\mathcal{D}\in\Delta(\{0,1\}^n)$ has the property of submodular dominance if for every submodular function $f$ it holds that $\mathbb{E}_{\boldsymbol{x}\sim\mathcal{D}}[f(\boldsymbol{x})]\geq\mathbb{E}_{x\sim\mathbf{Ind}(\mu_{\mathcal{D}})}[f(\boldsymbol{x})]$.

We now proceed by stating the properties of some state-of-the-art contention resolution schemes for product distributions, i.e. $\mathbf{Ind}(\boldsymbol{x})$ for some $\boldsymbol{x}\in\mathcal{P}(\mathcal{M})$. We also state a necessary and sufficient condition for the existence of a $\gamma$-selectable CRS for a distribution $\mathcal{D}$.

**Lemma B.4** (Chekuri et al. [2011]). *For any matroid $\mathcal{M}$ and any $\boldsymbol{x}\in\mathcal{P}(\mathcal{M})$, there exists a contention resolution scheme $\phi$ that is $(1-1/e)$-selectable for the distribution $\mathbf{Ind}(\boldsymbol{x})$.*

The above result holds for any matroid and is known to be asymptotically optimal for general matroids under product distributions [Chekuri et al., 2011]. For specific matroids, however, stronger guarantees are possible. In particular, the following lemma gives a tighter selectability bound for uniform matroids of rank $k$.

**Lemma B.5** (Kashaev and Santiago [2023]). *Let $\mathcal{M}'$ be the uniform matroid of rank $k$. There exists a contention resolution scheme $\phi'$ that is $(1-e^{-k}k^k/k!)$-selectable for the matroid $\mathcal{M}'$ and the distribution $\mathbf{Ind}(\boldsymbol{x})$ for any $\boldsymbol{x}\in\mathcal{P}(\mathcal{M}')$.*

Both the aforementioned results (Lemmas B.4 and B.5) admit constructive proofs, i.e. the respective contention resolution schemes are implementable in polynomial time and they are described in Chekuri et al. [2011] and Dughmi [2020] respectively. However, our algorithm does not rely on these constructions, as they are only needed in Appendix C for analysis purposes.

We now step away from product distributions and state a recent result of Dughmi [2020] that gives a necessary and sufficient condition for the existence of a $\gamma$-selectable CRS for a distribution $\mathcal{D}$.

**Lemma B.6** (Dughmi [2020]). *Let $\mathcal{M}$ be a matroid over $[n]$ and $\mathcal{D}\in\Delta(\{0,1\}^n)$ be a distribution over subsets of $[n]$. For any $\gamma\in[0,1]$, the following are equivalent.*

*a) There exists a contention resolution scheme $\phi$ that is $\gamma$-selectable for $\mathcal{D}$.*

*b) For every weight vector $\boldsymbol{w} \in \mathbb{R}_+^n$, it holds that*

$$\mathbb{E}_{\boldsymbol{x} \sim \mathcal{D}}\left[\operatorname{rank}_{\boldsymbol{w}}(\boldsymbol{x})\right] \geq \gamma \cdot \mathbb{E}_{\boldsymbol{x} \sim \mathcal{D}}\left[\sum_{i \in [n]} w_i x_i\right].$$

Using the previous lemma, one can prove that the property of submodular dominance (Definition B.3) is also sufficient for the existence of a $\gamma$-selectable CRS. The latter has been stated informally by Dughmi [2020]. We state it formally in the next lemma and give a small proof for completeness.

**Lemma B.7** (Dughmi [2020]). *Let $\mathcal{M}$ be a matroid over $[n]$ and $\mathcal{D} \in \Delta(2^{[n]})$ be a distribution over subsets of $[n]$. Suppose that $\mathcal{D}$ has the property of submodular dominance (Def. (B.3)). Then, for any $\gamma \in [0, 1]$, if there exists a $\gamma$-selectable CRS for $\textbf{Ind}(\boldsymbol{\mu}_{\mathcal{D}})$, then there also exists a $\gamma$-selectable CRS for $\mathcal{D}$.*

*Proof of Lemma B.7.* Fix any weight vector $\boldsymbol{w} \in \mathbb{R}_+^n$. Notice that

$$\mathbb{E}_{\boldsymbol{x} \sim \textbf{Ind}(\boldsymbol{\mu}_{\mathcal{D}})}\left[\sum_{i \in [n]} w_i x_i\right] = \sum_{i \in [n]} w_i \mathbb{E}_{\boldsymbol{x} \sim \textbf{Ind}(\boldsymbol{\mu}_{\mathcal{D}})}[x_i] = \sum_{i \in [n]} w_i \mathbb{E}_{\boldsymbol{x} \sim \mathcal{D}}[x_i] = \mathbb{E}_{\boldsymbol{x} \sim \mathcal{D}}\left[\sum_{i \in [n]} w_i x_i\right],$$

because $\mathbb{E}_{\boldsymbol{x} \sim \mathcal{D}}[x_i] = \mathbb{E}_{\boldsymbol{x} \sim \textbf{Ind}(\boldsymbol{\mu}_{\mathcal{D}})}[x_i] = (\boldsymbol{\mu}_{\mathcal{D}})_i$, for all $i \in [n]$.

If the distribution $\textbf{Ind}(\boldsymbol{\mu}_{\mathcal{D}})$ has a $\gamma$-selectable CRS, then by Lemma B.6(b), we get that

$$\mathbb{E}_{\boldsymbol{x} \sim \textbf{Ind}(\boldsymbol{\mu}_{\mathcal{D}})}\left[\operatorname{rank}_{\boldsymbol{w}}(\boldsymbol{x})\right] \geq \gamma \cdot \mathbb{E}_{\boldsymbol{x} \sim \textbf{Ind}(\boldsymbol{\mu}_{\mathcal{D}})}\left[\sum_{i \in [n]} w_i x_i\right] = \gamma \cdot \mathbb{E}_{\boldsymbol{x} \sim \mathcal{D}}\left[\sum_{i \in [n]} w_i x_i\right].$$

Since the weighted rank function is submodular for any non-negative vector $\boldsymbol{w}$ [Schrijver, 2003], we can use the submodular dominance property of $\mathcal{D}$ to get

$$\mathbb{E}_{\boldsymbol{x} \sim \mathcal{D}}\left[\operatorname{rank}_{\boldsymbol{w}}(\boldsymbol{x})\right] \geq \mathbb{E}_{\boldsymbol{x} \sim \textbf{Ind}(\boldsymbol{\mu}_{\mathcal{D}})}\left[\operatorname{rank}_{\boldsymbol{w}}(\boldsymbol{x})\right].$$

Combining the latter inequalities gives us that

$$\mathbb{E}_{\boldsymbol{x} \sim \mathcal{D}}\left[\operatorname{rank}_{\boldsymbol{w}}(\boldsymbol{x})\right] \geq \gamma \cdot \mathbb{E}_{\boldsymbol{x} \sim \mathcal{D}}\left[\sum_{i \in [n]} w_i x_i\right],$$

which by Lemma B.6 implies that there exists a $\gamma$-selectable CRS for $\mathcal{D}$. $\qquad\square$

Finally, by combining Lemmas B.7, B.4 and B.5 we get the following corollary, which we will use in our analysis (Appendix C).

**Corollary B.8.** *Let $\mathcal{M} = ([n], \mathcal{I})$ be a matroid and $\mathcal{D} \in \Delta(2^{[n]})$ be a distribution over subsets of $[n]$ for which the following hold: (1) $\boldsymbol{\mu}_{\mathcal{D}} \in \mathcal{P}(\mathcal{M})$, (2) for every submodular function $f$, $\mathbb{E}_{\boldsymbol{x} \sim \mathcal{D}}[f(\boldsymbol{x})] \geq \mathbb{E}_{\boldsymbol{x} \sim \textbf{Ind}(\boldsymbol{\mu}_{\mathcal{D}})}[f(\boldsymbol{x})]$. Then, there exists a $c_{\mathcal{M}}$-selectable CRS for $\mathcal{D}$, where $c_{\mathcal{M}} = 1 - e^{-k} k^k / k!$ if $\mathcal{M}$ is a uniform matroid of rank $k$ and $c_{\mathcal{M}} = 1 - 1/e$ otherwise.*

### B.2 CRS-Based Roundings of WTP Functions

We now state some useful lemmas for CRS-based rounding processes that are used to convert fractional solutions into integral ones; we use this to provide a bound on the expectation of a WTP function w.r.t. its concave relaxation.

First, we state a lemma by Chekuri et al. [2011], which proves that applying a $\gamma$-selectable CRS to a random set $\boldsymbol{x} \sim \mathcal{D}$ produces a feasible solution whose expected value, under any monotone submodular function $f$, is at least a $\gamma$ fraction of the value of $f$ under the independent distribution with the same marginals as $\mathcal{D}$.

**Lemma B.9** (Chekuri et al. [2011]). *Let $f$ be a monotone and non-negative submodular function over $2^{[n]}$ and $\phi$ be a $\gamma$-selectable CRS for a distribution $\mathcal{D}$. Then,*

$$\mathop{\mathbb{E}}_{\boldsymbol{x}\sim\mathcal{D},\phi}[f(\phi(\boldsymbol{x}))] \geq \gamma \cdot \mathop{\mathbb{E}}_{\boldsymbol{x}\sim\boldsymbol{Ind}(\boldsymbol{\mu}_{\mathcal{D}})}[f(\boldsymbol{x})].$$

The following result, due to Si Salem et al. [2024], shows that under a mild negative correlation condition, the expected value of a `WTP` function under a distribution $\mathcal{D}$ is well-approximated by its concave relaxation (as defined in Eq. (6)), evaluated at its expected vector $\boldsymbol{\mu}_{\mathcal{D}}$.

**Lemma B.10** (Si Salem et al. [2024]). *Let $f$ be a `WTP` function and $\widetilde{f}$ be its concave relaxation as defined in Eq. (6). Let also $\mathcal{D} \in \Delta(2^{[n]})$ be a distribution over subsets of $[n]$, such that for any subset $S \subseteq [n]$, $\mathbb{E}_{\boldsymbol{x}\sim\mathcal{D}}\left[\prod_{i\in S} x_i\right] \leq \prod_{i\in S} \mathbb{E}_{\boldsymbol{x}\sim\mathcal{D}}[x_i]$. Then,*

$$\mathop{\mathbb{E}}_{\boldsymbol{x}\sim\mathcal{D}}[f(\boldsymbol{x})] \geq \left(1 - \frac{1}{e}\right) \widetilde{f}(\boldsymbol{\mu}_{\mathcal{D}}).$$

Finally, we combine Lemma B.9 and Lemma B.10 to derive the following approximation guarantee for the expected value of a `WTP` function after applying a $\gamma$-selectable CRS.

**Lemma B.11.** *Let $f$ be a `WTP` function and $\widetilde{f}$ be its concave relaxation as defined in Eq. (6). Let also $\phi$ be a $\gamma$-selectable CRS for a distribution $\mathcal{D} \in \Delta(2^{[n]})$. Then,*

$$\mathop{\mathbb{E}}_{\boldsymbol{x}\sim\mathcal{D},\phi}[f(\phi(\boldsymbol{x}))] \geq \gamma \cdot \left(1 - \frac{1}{e}\right) \widetilde{f}(\boldsymbol{\mu}_{\mathcal{D}}).$$

*Proof of Lemma B.11.* Notice that the distribution $\mathbf{Ind}(\boldsymbol{\mu}_{\mathcal{D}})$ satisfies the properties of Lemma B.10, since for every subset $S \subseteq [n]$ we have that $\mathbb{E}_{\boldsymbol{x}\sim\mathbf{Ind}(\boldsymbol{\mu}_{\mathcal{D}})}\left[\prod_{i\in S} x_i\right] = \prod_{i\in S} \mathbb{E}_{\boldsymbol{x}\sim\mathbf{Ind}(\boldsymbol{\mu}_{\mathcal{D}})}[x_i]$. Therefore, from Lemma B.10 we get

$$\mathop{\mathbb{E}}_{\boldsymbol{x}\sim\mathbf{Ind}(\boldsymbol{\mu}_{\mathcal{D}})}[f(\boldsymbol{x})] \geq \left(1 - \frac{1}{e}\right) \widetilde{f}(\boldsymbol{\mu}_{\mathcal{D}}).$$

The proof is completed by combining the latter inequality with Lemma B.9:

$$\mathop{\mathbb{E}}_{\boldsymbol{x}\sim\mathcal{D},\phi}[f(\phi(\boldsymbol{x}))] \geq \gamma \cdot \mathop{\mathbb{E}}_{\boldsymbol{x}\sim\mathbf{Ind}(\boldsymbol{\mu}_{\mathcal{D}})}[f(\boldsymbol{x})] \geq \gamma \cdot \left(1 - \frac{1}{e}\right) \widetilde{f}(\boldsymbol{\mu}_{\mathcal{D}}). \qquad \square$$

# C Proof of Lemma 5.4

In this section, we prove Lemma 5.4, which shows that the expected reward of the rounded ground set, obtained by applying Randomized Pipage Rounding to a fractional ground set, approximately preserves the value of the initial fractional ground set. Specifically, we aim to show that for any `WTP` function $f$ and any fractional ground set $\widetilde{x} \in \widetilde{X}_\ell$, the expected reward of the rounded ground set $x = \Xi(\tilde{x})$ satisfies

$$\mathbb{E}_\Xi[F(\Xi(\tilde{x}))] \geq c_{\mathcal{M}}(1 - 1/e) \cdot \widetilde{F}(\tilde{x}),$$

where $F(\boldsymbol{x}) = \max_{\boldsymbol{y}\in\mathcal{Y}_{\mathcal{M}}(\boldsymbol{x})} f(\boldsymbol{y})$ is the reward of the rounded ground set (Eq. (7)) and $\widetilde{F}(\widetilde{\boldsymbol{x}}) = \max_{\widetilde{\boldsymbol{y}}\in\widetilde{\mathcal{Y}}_{\mathcal{M}}(\widetilde{\boldsymbol{x}})} \widetilde{f}(\widetilde{\boldsymbol{y}})$ is the concave relaxation of the reward function (Eq. (8)). The constant $c_{\mathcal{M}}$ depends on the structure of the second-stage matroid.

As a reminder, for a ground set $\boldsymbol{x} \in \mathcal{X}_\ell$, the set of feasible second stage solutions, is the restriction of the matroid $\mathcal{M}$ to the set $\boldsymbol{x}$, i.e. $\mathcal{Y}_{\mathcal{M}}(\boldsymbol{x}) = \{\boldsymbol{y} \in \{0,1\}^n : \boldsymbol{y} \in \mathcal{M}, \boldsymbol{y} \leq \boldsymbol{x}\}$. In the same manner, for a fractional ground set $\widetilde{\boldsymbol{x}} \in \widetilde{\mathcal{X}}_\ell$, the corresponding fractional relaxation of the second-stage feasible solutions is $\widetilde{\mathcal{Y}}_{\mathcal{M}}(\widetilde{\boldsymbol{x}}) = \{\widetilde{\boldsymbol{y}} \in [0,1]^n : \widetilde{\boldsymbol{y}} \in \mathcal{P}(\mathcal{M}), \widetilde{\boldsymbol{y}} \leq \widetilde{\boldsymbol{x}}\}$.

**High-level description.** On a high-level, we need to show that the value attained by maximizing a `WTP` function over the fractional ground set remains approximately the same when we transition to the rounded ground set. To establish this, we design a two-stage randomized rounding procedure, used purely for analysis, which transforms any fractional second-stage solution into a feasible integral

one that has, in expectation, approximately the same value. This construction is coupled with the Randomized Pipage Rounding used by `RAOCO` to select the integral ground set. The rounding is performed in two steps: First, a fractional second-stage solution is mapped to an intermediate integral vector that lies within the selected ground set but may violate the matroid constraints. Then, we apply a contention resolution scheme (CRS) to trim the intermediate vector to a feasible solution (for more details on CRSs see Appendix B). A key challenge is that standard CRSs typically need each element of the input vector to have been included in it independently of other elements. This assumption does not hold in our case since the intermediate vector depends on the outcome of the Randomized Pipage Rounding which introduces dependencies across elements. However, we show that the intermediate vector exhibits the property of submodular dominance (Def. (B.3)) which allows us to use state-of-the art CRS guarantees (Corollary B.8). We now formalize this construction and use it to prove Lemma 5.4.

**Rounding Description.** Fix any $\widetilde{x} \in \widetilde{\mathcal{X}}_\ell$ and $x \in \mathcal{X}_\ell$. Let also $\widetilde{y} \in \widetilde{\mathcal{Y}}_{\mathcal{M}}(\widetilde{x})$ be a second stage solution that is consistent with $\widetilde{x}$. To map $\widetilde{y} \in \mathcal{Y}_{\mathcal{M}}(\widetilde{x})$ to a solution $y \in \mathcal{Y}_{\mathcal{M}}(x)$, we first round $\widetilde{y}$ to an intermediate solution $\hat{y}$, through a randomized rounding $Q$. The intermediate solution $\hat{y}$ is always feasible for the new groundset $x$, i.e. $\hat{y} \le x$, but it might violate the matroid constraints. For this reason, we pass it through a contention resolution scheme $\phi$ to trim it down to a solution $y = \phi(\hat{y}) \le \hat{y} \le x$ that also respects the matroid constraints. On a high level, the rounding procedure we described can be seen in the following diagram.

$$\boxed{\widetilde{y} \in \widetilde{\mathcal{Y}}_{\mathcal{M}}(\tilde{x})} \quad \underset{\substack{\text{uses} \\ \widetilde{x},x}}{\xrightarrow{Q}} \quad \hat{y} \quad \xrightarrow{\phi} \quad \boxed{y \in \mathcal{Y}_{\mathcal{M}}(x)}$$

Formally, the rounding $Q$ accepts $\widetilde{x}$, $x$ and $\widetilde{y} \in \widetilde{\mathcal{Y}}_{\mathcal{M}}(\widetilde{x})$ and produces $\hat{y} = Q(\widetilde{y}, \widetilde{x}, x)$. For $i \in [n]$, let $a_i$ be

$$a_i = \begin{cases} \frac{\widetilde{y}_i}{\widetilde{x}_i}, & \text{if } \widetilde{x}_i \neq 0, \\ 0, & \text{otherwise.} \end{cases}$$

The rounding $Q$ is defined follows: first, it samples $n$ independent Bernoulli random variables, $c_1, \ldots, c_n$, with $c_i \sim Ber(a_i)$. Then, it sets the $i$-th coordinate of the resulting vector $\hat{y}$ to be

$$\hat{y}_i = \begin{cases} 1, & \text{if } x_i = 1 \text{ and } c_i = 1, \\ 0, & \text{otherwise.} \end{cases}$$

In other words, $\hat{y} = x \wedge c$. By construction, $\hat{y}$ only uses elements that are in the groundset $x$, i.e. $\hat{y} \le x$. However, it does not necessarily satisfy the matroid constraints. For this reason, we pass $\hat{y}$ through a contention resolution scheme $\phi$ for the matroid $\mathcal{M}$, to obtain $y = \phi(\hat{y})$. We do not instantiate the scheme $\phi$ yet, but we remark that using any contention resolution scheme for the matroid $\mathcal{M}$, will result in a set $y$ that is a feasible second-stage solution for the rounded ground set, i.e. $y \in \mathcal{Y}_{\mathcal{M}}(x)$. We state this formally in the following lemma.

**Lemma C.1.** *Fix any $\widetilde{x} \in \widetilde{\mathcal{X}}_\ell$, any $x \in \mathcal{X}_\ell$ and any $\widetilde{y} \in \widetilde{\mathcal{Y}}_{\mathcal{M}}(\widetilde{x})$. Let $\hat{y} = Q(\widetilde{y}, \widetilde{x}, x)$ and $y = \phi(\hat{y})$, where $\phi$ is a contention resolution scheme for the matroid $\mathcal{M}$. Then, $y \in \mathcal{Y}_{\mathcal{M}}(x)$.*

*Proof of Lemma C.1.* Recall that $\mathcal{Y}_{\mathcal{M}}(x)$ contains all the independent sets of $\mathcal{M}$ that are also subsets of $x$. By definition of the rounding $Q$, if $x_i = 0$ for some coordinate $i \in [n]$, then $\hat{y}_i = 0$. This means that $\hat{y}$ is a subset of $x$, i.e. $\hat{y} \le x$. By the definition of contention resolution schemes (Def. B.1), we know that $y = \phi(\hat{y}) \le \hat{y} \le x$ and $y = \phi(\hat{y}) \in \mathcal{I}$, where $\mathcal{I}$ is the family of independent sets of the matroid $\mathcal{M}$. Therefore, $y \in \mathcal{Y}_{\mathcal{M}}(x)$. $\qquad\square$

Knowing that the resulting vector is always feasible allows us to link the expected value of the transformed solution with the reward of the rounded groundset $x$.

**Lemma C.2.** *Fix any* WTP *function $f$ and its correspoding reward function $F$, as defined in Eq. (7). Then, for any $\widetilde{\boldsymbol{x}} \in \widetilde{\mathcal{X}}_\ell$ and any $\widetilde{\boldsymbol{y}} \in \widetilde{\mathcal{Y}}_\mathcal{M}(\widetilde{\boldsymbol{x}})$ it holds that*

$$\mathbb{E}_{\boldsymbol{x} \sim \Xi(\widetilde{\boldsymbol{x}})} [F(\boldsymbol{x})] \geq \mathbb{E}_{\boldsymbol{x} \sim \Xi(\widetilde{\boldsymbol{x}})} \left[ \mathbb{E}_{\boldsymbol{y} \sim \phi(Q(\widetilde{\boldsymbol{y}}, \widetilde{\boldsymbol{x}}, \boldsymbol{x}))} [f(\boldsymbol{y})] \right].$$

*Proof.* Fix a vector $\boldsymbol{x}_0 \in \mathcal{X}_\ell$ and let $\hat{\boldsymbol{y}} = Q(\widetilde{\boldsymbol{y}}, \widetilde{\boldsymbol{x}}, \boldsymbol{x}_0)$ and $\boldsymbol{y} = \phi(\hat{\boldsymbol{y}})$. The expected value of the produced vector, $\boldsymbol{y}$, is

$$\mathbb{E}_{\boldsymbol{y} \sim \phi(Q(\widetilde{\boldsymbol{y}}, \widetilde{\boldsymbol{x}}, \boldsymbol{x}_0))} [f(\boldsymbol{y})].$$

By Lemma C.1 we know that $\boldsymbol{y} \in \mathcal{Y}_\mathcal{M}(\boldsymbol{x}_0)$. This means that there must exist a $\boldsymbol{y}' \in \mathcal{Y}_\mathcal{M}(\boldsymbol{x}_0)$ whose value is at least the expected value of the produced vector, i.e.

$$f(\boldsymbol{y}') \geq \mathbb{E}_{\boldsymbol{y} \sim \phi(Q(\widetilde{\boldsymbol{y}}, \widetilde{\boldsymbol{x}}, \boldsymbol{x}_0))} [f(\boldsymbol{y})].$$

By the definition of the reward function $F$ (Eq. (7)), we have that

$$F(\boldsymbol{x}_0) \geq f(\boldsymbol{y}') \geq \mathbb{E}_{\boldsymbol{y} \sim \phi(Q(\widetilde{\boldsymbol{y}}, \widetilde{\boldsymbol{x}}, \boldsymbol{x}_0))} [f(\boldsymbol{y})].$$

Notice that the above inequality holds for any $\boldsymbol{x}_0 \in \mathcal{X}_\ell$. We multiply this inequality by the probability that $\boldsymbol{x}_0$ is the result of Randomized Pipage Rounding on $\widetilde{\boldsymbol{x}}$ and then sum up the corresponding inequalities for all $\boldsymbol{x}_0 \in \mathcal{X}_\ell$:

$$\sum_{\boldsymbol{x}_0 \in \mathcal{X}_\ell} F(\boldsymbol{x}_0) \cdot \Pr_{\boldsymbol{x} \sim \Xi(\widetilde{\boldsymbol{x}})} [\boldsymbol{x} = \boldsymbol{x}_0] \geq \sum_{\boldsymbol{x}_0 \in \mathcal{X}_\ell} \mathbb{E}_{\boldsymbol{y} \sim \phi(Q(\widetilde{\boldsymbol{y}}, \widetilde{\boldsymbol{x}}, \boldsymbol{x}_0))} [f(\boldsymbol{y})] \cdot \Pr_{\boldsymbol{x} \sim \Xi(\widetilde{\boldsymbol{x}})} [\boldsymbol{x} = \boldsymbol{x}_0]$$

$$\Rightarrow \mathbb{E}_{\boldsymbol{x} \sim \Xi(\widetilde{\boldsymbol{x}})} [F(\boldsymbol{x})] \geq \mathbb{E}_{\boldsymbol{x} \sim \Xi(\widetilde{\boldsymbol{x}})} \left[ \mathbb{E}_{\boldsymbol{y} \sim \phi(Q(\widetilde{\boldsymbol{y}}, \widetilde{\boldsymbol{x}}, \boldsymbol{x}))} [f(\boldsymbol{y})] \right] \qquad \square$$

Having the above lemma, we will now turn our focus on analyzing the expected value of $f(\boldsymbol{y})$ when $\boldsymbol{x}$ has been produced by Randomized Pipage Rounding. We start by proving some properties for $\hat{\boldsymbol{y}}$ and then "instantiate" the contention resolution scheme $\phi$.

In order to make the notation simpler, we will use $Q(\hat{\boldsymbol{y}}, \widetilde{\boldsymbol{x}}, \Xi(\widetilde{\boldsymbol{x}}))$ to denote the distribution $Q(\hat{\boldsymbol{y}}, \widetilde{\boldsymbol{x}}, \boldsymbol{x})$ where $\boldsymbol{x} \sim \Xi(\widetilde{\boldsymbol{x}})$. We begin by showing that the expected value of $\hat{\boldsymbol{y}}$ is the initial fractional solution $\widetilde{\boldsymbol{y}}$.

**Lemma C.3.** *For any $\widetilde{\boldsymbol{x}} \in \widetilde{\mathcal{X}}_\ell$ and any $\widetilde{\boldsymbol{y}} \in \widetilde{\mathcal{Y}}_\mathcal{M}(\widetilde{\boldsymbol{x}})$ it holds that*

$$\mathbb{E}_{\hat{\boldsymbol{y}} \sim Q(\widetilde{\boldsymbol{y}}, \widetilde{\boldsymbol{x}}, \Xi(\widetilde{\boldsymbol{x}}))} [\hat{\boldsymbol{y}}] = \widetilde{\boldsymbol{y}}.$$

*Proof of Lemma C.3.* Fix an index $i \in [n]$. If $\widetilde{\boldsymbol{x}}_i = 0$ then $\widetilde{\boldsymbol{y}}_i = 0$. This because $\widetilde{\boldsymbol{y}} \in \widetilde{\mathcal{Y}}_\mathcal{M}(\widetilde{\boldsymbol{x}})$ which implies that $\widetilde{\boldsymbol{y}} \leq \widetilde{\boldsymbol{x}}$. In this case, we have that $\boldsymbol{y}_i = 0 = \hat{\boldsymbol{y}}_i$. In the case where $\widetilde{\boldsymbol{x}}_i \neq 0$, the following is true

$$\Pr_{\Xi, Q}[\hat{\boldsymbol{y}}_i = 1] = \Pr_{\Xi, Q}[c_i = 1 \wedge \boldsymbol{x}_i = 1] = \Pr_Q[c_i = 1] \cdot \Pr_\Xi[\boldsymbol{x}_i = 1] \overset{\text{Lem. } I.1}{=} \frac{\widetilde{\boldsymbol{y}}_i}{\widetilde{\boldsymbol{x}}_i} \cdot \widetilde{\boldsymbol{x}}_i = \widetilde{\boldsymbol{y}}_i,$$

where in the second to last inequality we used Lemma I.1 for the marginals of the Randomized Pipage rounding.

In all cases we have that $\mathbb{E}_{\Xi, Q}[\hat{\boldsymbol{y}}_i] = \Pr_{\Xi, Q}[\hat{\boldsymbol{y}}_i = 1] = \widetilde{\boldsymbol{y}}_i$, which concludes the proof. $\qquad \square$

We continue by showing that the distribution $Q(\widetilde{\boldsymbol{y}}, \widetilde{\boldsymbol{x}}, \Xi(\widetilde{\boldsymbol{x}}))$ exhibits the propery of submodular dominance (Def. (B.3)).

**Lemma C.4.** *For any $\widetilde{\boldsymbol{x}} \in \widetilde{\mathcal{X}}_\ell$ and any $\widetilde{\boldsymbol{y}} \in \widetilde{\mathcal{Y}}_\mathcal{M}(\widetilde{\boldsymbol{x}})$, the distribution $Q(\widetilde{\boldsymbol{y}}, \widetilde{\boldsymbol{x}}, \Xi(\widetilde{\boldsymbol{x}}))$ has the property of submodular dominance (Def. B.3), i.e. for every submodular function $f$ it holds that*

$$\mathbb{E}_{\hat{\boldsymbol{y}} \sim Q(\widetilde{\boldsymbol{y}}, \widetilde{\boldsymbol{x}}, \Xi(\widetilde{\boldsymbol{x}}))} [f(\hat{\boldsymbol{y}})] \geq \mathbb{E}_{\hat{\boldsymbol{y}} \sim Ind(\widetilde{\boldsymbol{y}})} [f(\hat{\boldsymbol{y}})].$$

*Proof of Lemma C.4.* Let $\boldsymbol{p} \in [0,1]^n$ be a vector whose $i$-th coordinate is defined as

$$
\boldsymbol{p}_i = \begin{cases} \frac{\widetilde{\boldsymbol{y}}_i}{\widetilde{\boldsymbol{x}}_i}, & \text{if } \widetilde{\boldsymbol{x}}_i \neq 0 \\ 0, & \text{otherwise} \end{cases} \quad .
$$

Fix any submodular function $f$. We define the function $g : 2^{[n]} \to \mathbb{R}$ as

$$
g(\boldsymbol{x}) \overset{\triangle}{=} \underset{\boldsymbol{z} \sim \mathbf{Ind}(\boldsymbol{p})}{\mathbb{E}} [f(\boldsymbol{x} \wedge \boldsymbol{z})].
$$

Intuitively, $g$ maps every set $\boldsymbol{x}$ to its expected value after removing each one of its elements independently with probability $\boldsymbol{p}_i$. Notice that this is exactly what the rounding $Q$ does to every rounded ground set $\boldsymbol{x}$. In other words, $\hat{\boldsymbol{y}}$ is simply the result of setting the $i$-th coordinate of $\boldsymbol{x}$ to zero, independently, with probability $p_i$. From this observation we get that

$$
\underset{\hat{\boldsymbol{y}} \sim Q(\widetilde{\boldsymbol{y}}, \widetilde{\boldsymbol{x}}, \Xi(\widetilde{\boldsymbol{x}}))}{\mathbb{E}} [f(\hat{\boldsymbol{y}})] = \underset{\boldsymbol{x} \sim \Xi(\widetilde{\boldsymbol{x}})}{\mathbb{E}} \left[ \underset{\boldsymbol{z} \sim \mathbf{Ind}(\boldsymbol{p})}{\mathbb{E}} [f(\boldsymbol{x} \wedge \boldsymbol{z})] \right] = \underset{\boldsymbol{x} \sim \Xi(\widetilde{\boldsymbol{x}})}{\mathbb{E}} [g(\boldsymbol{x})].
$$

In a similar manner, we have that

$$
\underset{\hat{\boldsymbol{y}} \sim \mathbf{Ind}(\widetilde{\boldsymbol{y}})}{\mathbb{E}} [f(\hat{\boldsymbol{y}})] = \underset{\boldsymbol{x} \sim \mathbf{Ind}(\widetilde{\boldsymbol{x}})}{\mathbb{E}} \left[ \underset{\boldsymbol{z} \sim \mathbf{Ind}(\boldsymbol{p})}{\mathbb{E}} [f(\boldsymbol{x} \wedge \boldsymbol{z})] \right] = \underset{\boldsymbol{x} \sim \mathbf{Ind}(\widetilde{\boldsymbol{x}})}{\mathbb{E}} [g(\boldsymbol{x})].
$$

We conclude the proof by showing that the function $g$ is submodular and then applying the submodular dominance property of the Randomized Pipage Rounding (Lemma I.2).

To prove the submodularity of $g$, fix a subset $\boldsymbol{z} \in \{0,1\}^n$ and two other subsets $\boldsymbol{x}, \boldsymbol{y} \in \{0,1\}^n$. By the submodularity property of $f$ applied to the sets $(\boldsymbol{x} \wedge \boldsymbol{z})$ and $(\boldsymbol{y} \wedge \boldsymbol{z})$, we have that

$$
f(\boldsymbol{x} \wedge \boldsymbol{z}) + f(\boldsymbol{y} \wedge \boldsymbol{z}) \geq f\big((\boldsymbol{x} \wedge \boldsymbol{z}) \wedge (\boldsymbol{y} \wedge \boldsymbol{z})\big) + f\big((\boldsymbol{x} \wedge \boldsymbol{z}) \vee (\boldsymbol{y} \wedge \boldsymbol{z})\big)
$$
$$
= f\big((\boldsymbol{x} \wedge \boldsymbol{y}) \wedge \boldsymbol{z}\big) + f\big((\boldsymbol{x} \vee \boldsymbol{y}) \wedge \boldsymbol{z}\big).
$$

We write the above inequality for any $\boldsymbol{z}_0 \in \{0,1\}^n$ and multiply both sides with $\mathbf{Pr}_{\boldsymbol{z} \sim \mathbf{Ind}(\boldsymbol{p})}[\boldsymbol{z} = \boldsymbol{z}_0]$. By summing up all the aforementioned inequalities we get

$$
\sum_{\boldsymbol{z}_0 \in \{0,1\}^n} (f(\boldsymbol{x} \wedge \boldsymbol{z}) + f(\boldsymbol{y} \wedge \boldsymbol{z})) \cdot \underset{\boldsymbol{z} \sim \mathbf{Ind}(\boldsymbol{p})}{\mathbf{Pr}}[\boldsymbol{z} = \boldsymbol{z}_0]
$$
$$
\geq \sum_{\boldsymbol{z}_0 \in \{0,1\}^n} \big(f\big((\boldsymbol{x} \wedge \boldsymbol{y}) \wedge \boldsymbol{z}\big) + f\big((\boldsymbol{x} \vee \boldsymbol{y}) \wedge \boldsymbol{z}\big)\big) \cdot \underset{\boldsymbol{z} \sim \mathbf{Ind}(\boldsymbol{p})}{\mathbf{Pr}}[\boldsymbol{z} = \boldsymbol{z}_0]
$$
$$
\Rightarrow \underset{\boldsymbol{z} \sim \mathbf{Ind}(\boldsymbol{p})}{\mathbb{E}}[f(\boldsymbol{x} \wedge \boldsymbol{z})] + \underset{\boldsymbol{z} \sim \mathbf{Ind}(\boldsymbol{p})}{\mathbb{E}}[f(\boldsymbol{y} \wedge \boldsymbol{z})]
$$
$$
\geq \underset{\boldsymbol{z} \sim \mathbf{Ind}(\boldsymbol{p})}{\mathbb{E}}[f\big((\boldsymbol{x} \wedge \boldsymbol{y}) \wedge \boldsymbol{z}\big)] + \underset{\boldsymbol{z} \sim \mathbf{Ind}(\boldsymbol{p})}{\mathbb{E}}[f\big((\boldsymbol{x} \vee \boldsymbol{y}) \wedge \boldsymbol{z}\big)]
$$
$$
\Rightarrow g(\boldsymbol{x}) + g(\boldsymbol{y}) \geq g(\boldsymbol{x} \wedge \boldsymbol{y}) + g(\boldsymbol{x} \vee \boldsymbol{y})
$$

which proves that $g$ is submodular.

We conclude the proof by applying Lemma I.2 to $g$,

$$
\underset{\hat{\boldsymbol{y}} \sim Q(\widetilde{\boldsymbol{y}}, \widetilde{\boldsymbol{x}}, \Xi(\widetilde{\boldsymbol{x}}))}{\mathbb{E}} [f(\hat{\boldsymbol{y}})] = \underset{\boldsymbol{x} \sim \Xi(\widetilde{\boldsymbol{x}})}{\mathbb{E}} [g(\boldsymbol{x})] \overset{\text{Lem. I.2}}{\geq} \underset{\boldsymbol{x} \sim \mathbf{Ind}(\widetilde{\boldsymbol{x}})}{\mathbb{E}} [g(\boldsymbol{x})] = \underset{\hat{\boldsymbol{y}} \sim \mathbf{Ind}(\widetilde{\boldsymbol{y}})}{\mathbb{E}} [f(\hat{\boldsymbol{y}})]. \qquad \square
$$

Using the above properties for $\hat{\boldsymbol{y}}$, we will now show that there exists an "appropriate" CRS $\phi$, which we will use to finally prove Lemma 5.4.

**Lemma C.5.** *For any $\widetilde{\boldsymbol{x}} \in \widetilde{\mathcal{X}}_\ell$ and any $\widetilde{\boldsymbol{y}} \in \widetilde{\mathcal{Y}}_{\mathcal{M}}(\widetilde{\boldsymbol{x}})$, there exists a $c_{\mathcal{M}}$-selectable CRS for the distribution $Q(\widetilde{\boldsymbol{y}}, \widetilde{\boldsymbol{x}}, \Xi(\widetilde{\boldsymbol{x}}))$ and the matroid $\mathcal{M}$, where $c_{\mathcal{M}} = 1 - e^{-k} k^k / k!$ if $\mathcal{M}$ is a uniform matroid with rank $k$ and $c_{\mathcal{M}} = 1 - 1/e$ otherwise.*

*Proof of Lemma C.5.* Let $c_{\mathcal{M}}$ be the constant mentioned in the statement of the lemma. From Corollary B.8, we know that in order for a distribution to have a $c_{\mathcal{M}}$-selectable CRS, it suffices that it's expectation is in the matroid polytope and that is has the property of submodular dominance. Fix any pair of $\widetilde{\boldsymbol{x}} \in \widetilde{\mathcal{X}}_{\ell}$ and $\widetilde{\boldsymbol{y}} \in \widetilde{\mathcal{Y}}_{\mathcal{M}}(\widetilde{\boldsymbol{x}})$. For each such pair, $Q(\widetilde{\boldsymbol{y}}, \widetilde{\boldsymbol{x}}, \Xi(\widetilde{\boldsymbol{x}}))$ defines a distribution over subsets of $[n]$ that depends on the randomness of $Q$ and $\Xi$. From Lemma C.3 we have that

$$\underset{\hat{\boldsymbol{y}} \sim Q(\widetilde{\boldsymbol{y}}, \widetilde{\boldsymbol{x}}, \Xi(\widetilde{\boldsymbol{x}}))}{\mathbb{E}} [\, \hat{\boldsymbol{y}} \,] = \widetilde{\boldsymbol{y}} \in \mathcal{P}(\mathcal{M}),$$

which is in the matroid polytope since $\widetilde{\boldsymbol{y}} \in \widetilde{\mathcal{Y}}_{\mathcal{M}}(\widetilde{\boldsymbol{x}})$. Furthermore, from Lemma C.4, we know that the distribution $Q(\widetilde{\boldsymbol{y}}, \widetilde{\boldsymbol{x}}, \boldsymbol{x})$ has the property of submodular dominance. As a result, using Corollary B.8, we get that there exists a $c_{\mathcal{M}}$-selectable CRS for the distribution $Q(\widetilde{\boldsymbol{y}}, \widetilde{\boldsymbol{x}}, \Xi(\widetilde{\boldsymbol{x}}))$. $\square$

We are now ready to prove Lemma 5.4.

*Proof of Lemma 5.4.* For any solution $\widetilde{\boldsymbol{y}} \in \widetilde{\mathcal{Y}}_{\mathcal{M}}(\widetilde{\boldsymbol{x}})$ and any contention resolution scheme $\phi$, from Lemma C.2 we know that

$$\underset{\boldsymbol{x} \sim \Xi(\widetilde{\boldsymbol{x}})}{\mathbb{E}} [F(\boldsymbol{x})] \geq \underset{\boldsymbol{x} \sim \Xi(\widetilde{\boldsymbol{x}})}{\mathbb{E}} \left[ \underset{\boldsymbol{y} \sim \phi(Q(\widetilde{\boldsymbol{y}}, \widetilde{\boldsymbol{x}}, \boldsymbol{x}))}{\mathbb{E}} [f(\boldsymbol{y})] \,\Big| \, \right] = \underset{\hat{\boldsymbol{y}} \sim Q(\widetilde{\boldsymbol{y}}, \widetilde{\boldsymbol{x}}, \Xi(\widetilde{\boldsymbol{x}}))}{\mathbb{E}} [f(\phi(\hat{\boldsymbol{y}}))] .$$

By selecting $\phi$ to be the $c_{\mathcal{M}}$-selectable contention resolution scheme guaranteed by Lemma C.5 and applying Lemma B.11, we get the following

$$\underset{\hat{\boldsymbol{y}} \sim Q(\widetilde{\boldsymbol{y}}, \widetilde{\boldsymbol{x}}, \Xi(\widetilde{\boldsymbol{x}}))}{\mathbb{E}} [f(\phi(\hat{\boldsymbol{y}}))] \overset{\text{Lem. B.11}}{\geq} c_{\mathcal{M}} \cdot \left( 1 - \frac{1}{e} \right) \cdot \widetilde{f}(\widetilde{\boldsymbol{y}}),$$

where $\widetilde{f}$ is the concave relaxation of the WTP function $f$, as defined in Eq. (6).

Let $\boldsymbol{y}^* \in \operatorname{argmax}_{\boldsymbol{y} \in \widetilde{\mathcal{Y}}_{\mathcal{M}}(\widetilde{\boldsymbol{x}})} \widetilde{f}(\boldsymbol{y})$. By selecting $\widetilde{\boldsymbol{y}} = \boldsymbol{y}^*$ in the above inequality we get

$$\underset{\boldsymbol{x} \sim \Xi(\widetilde{\boldsymbol{x}})}{\mathbb{E}} [F(\boldsymbol{x})] \geq c_{\mathcal{M}} \cdot \left( 1 - \frac{1}{e} \right) \cdot \widetilde{f}(\boldsymbol{y}^*) = c_{\mathcal{M}} \cdot \left( 1 - \frac{1}{e} \right) \cdot \widetilde{F}(\widetilde{\boldsymbol{x}}),$$

where in the last equality we used that $\widetilde{F}(\widetilde{\boldsymbol{x}}) = \widetilde{f}(\boldsymbol{y}^*)$ by the definition of the fractional reward function (Eq. (8)). $\square$

# D   Regret of OCO Policies in Fractional O2SSM

In this section, we prove Lemma 5.2 which states that any standard OCO policy (e.g., FTRL or OGA, see Appendix H for more details) has no-regret in the fractional relaxation of O2SSM.

**Notation.** We introduce some extra notation regarding matroids that we will use in our proofs in this section. For a matroid $\mathcal{M} = (\mathcal{V}, \mathcal{I})$, the rank function of $\mathcal{M}$, rank : $2^{\mathcal{V}} \to \mathbb{N}$, maps each set $U \subseteq [n]$ to the size of the largest independent set contained in $U$. Also, the matroid polytope $\mathcal{P}(\mathcal{M})$ can equivalently be written as: $\mathcal{P}(\mathcal{M}) = \{\widetilde{\boldsymbol{y}} \in [0, 1]^n : \sum_{i \in S} \widetilde{y}_i \leq \operatorname{rank}(S), \, \forall S \subseteq \mathcal{V}\}$.

First, we will show that the fractional reward function $\widetilde{F}$ associated with a WTP function $f$ is concave and Lipschitz. These two properties are proven in the next two lemmas (Lemmas D.1 and D.2).

**Lemma D.1.** *Let $f$ be any WTP function, $\widetilde{f}$ be its concave relaxation as defined in Eq. (6) and $\widetilde{F}$ be the concave relaxation of the reward function associated with $\widetilde{f}$, as defined in Eq. (8). Then, $\widetilde{F}$ is concave over the domain $\widetilde{\mathcal{X}}_{\ell}$.*

*Proof of Theorem D.1.* Recall that $\widetilde{F}$ is defined in Eq. (8) as

$$\widetilde{F}(\widetilde{\boldsymbol{x}}) = \max_{\widetilde{\boldsymbol{y}} \in \widetilde{\mathcal{Y}}_{\mathcal{M}}(\widetilde{\boldsymbol{x}})} \widetilde{f}(\widetilde{\boldsymbol{y}}), \, \forall \widetilde{\boldsymbol{x}} \in \widetilde{\mathcal{X}}_{\ell}.$$

Let $\widetilde{\boldsymbol{x}}_1, \widetilde{\boldsymbol{x}}_2 \in \widetilde{\mathcal{X}}_\ell$ be any two fractional restricted ground sets. We will show that for any $\lambda \in [0,1]$, the following holds

$$\widetilde{F}_t(\lambda \widetilde{\boldsymbol{x}}_1 + (1-\lambda)\widetilde{\boldsymbol{x}}_2) \geq \lambda \widetilde{F}_t(\widetilde{\boldsymbol{x}}_1) + (1-\lambda)\widetilde{F}_t(\widetilde{\boldsymbol{x}}_2).$$

We define the vectors $\widetilde{\boldsymbol{y}}_1 \in \widetilde{\mathcal{Y}}_{\mathcal{M}}(\widetilde{\boldsymbol{x}}_1)$ and $\widetilde{\boldsymbol{y}}_2 \in \widetilde{\mathcal{Y}}_{\mathcal{M}}(\widetilde{\boldsymbol{x}}_2)$ as follows

$$\widetilde{\boldsymbol{y}}_1 = \operatorname*{argmax}_{\widetilde{\boldsymbol{y}} \in \widetilde{\mathcal{Y}}_{\mathcal{M}}(\widetilde{\boldsymbol{x}}_1)} \widetilde{f}_t(\widetilde{\boldsymbol{y}}) \qquad \text{and} \qquad \widetilde{\boldsymbol{y}}_2 = \operatorname*{argmax}_{\widetilde{\boldsymbol{y}} \in \widetilde{\mathcal{Y}}_{\mathcal{M}}(\widetilde{\boldsymbol{x}}_2)} \widetilde{f}_t(\widetilde{\boldsymbol{y}}).$$

It holds that

$$\begin{aligned}
\lambda \widetilde{F}(\widetilde{\boldsymbol{x}}_1) + (1-\lambda)\widetilde{F}(\widetilde{\boldsymbol{x}}_2) &= \lambda \widetilde{f}(\widetilde{\boldsymbol{y}}_1) + (1-\lambda)\widetilde{f}(\widetilde{\boldsymbol{y}}_2) \\
&\leq \widetilde{f}(\lambda \widetilde{\boldsymbol{y}}_1 + (1-\lambda)\widetilde{\boldsymbol{y}}_2) \\
&\leq \max_{\widetilde{\boldsymbol{y}} \in \widetilde{\mathcal{Y}}_{\mathcal{M}}(\lambda \widetilde{\boldsymbol{x}}_1 + (1-\lambda)\widetilde{\boldsymbol{x}}_2)} \widetilde{f}(\widetilde{\boldsymbol{y}}) \\
&= \widetilde{F}(\lambda \widetilde{\boldsymbol{x}}_1 + (1-\lambda)\widetilde{\boldsymbol{x}}_2),
\end{aligned}$$

where first inequality is true by the concavity of $\widetilde{f}$ and to get the second inequality we used the fact that $\lambda \widetilde{\boldsymbol{y}}_1 + (1-\lambda)\widetilde{\boldsymbol{y}}_2 \in \widetilde{\mathcal{Y}}_{\mathcal{M}}(\lambda \widetilde{\boldsymbol{x}}_1 + (1-\lambda)\widetilde{\boldsymbol{x}}_2)$, which we prove below.

The memberships $\widetilde{\boldsymbol{y}}_1 \in \widetilde{\mathcal{Y}}_{\mathcal{M}}(\widetilde{\boldsymbol{x}}_1)$ and $\widetilde{\boldsymbol{y}}_2 \in \widetilde{\mathcal{Y}}_{\mathcal{M}}(\widetilde{\boldsymbol{x}}_2)$ imply the following inequalities:

$$\begin{cases} \lambda \widetilde{\boldsymbol{y}}_1 \leq \lambda \widetilde{\boldsymbol{x}}_1, \\ \lambda \sum_{i \in S} \widetilde{y}_{1,i} \leq \lambda \operatorname{rank}(S), \forall S \subseteq \mathcal{V} \end{cases} \quad \text{and} \quad \begin{cases} (1-\lambda)\widetilde{\boldsymbol{y}}_2 \leq (1-\lambda)\widetilde{\boldsymbol{x}}_2, \\ (1-\lambda) \sum_{i \in S} \widetilde{y}_{2,i} \leq (1-\lambda)\operatorname{rank}(S), \quad \forall S \subseteq \mathcal{V}. \end{cases}$$

Adding up the above inequalities gives us that $\lambda \widetilde{\boldsymbol{y}}_1 + (1-\lambda)\widetilde{\boldsymbol{y}}_2 \in \widetilde{\mathcal{Y}}_{\mathcal{M}}(\lambda \widetilde{\boldsymbol{x}}_1 + (1-\lambda)\widetilde{\boldsymbol{x}}_2)$, which concludes the proof. $\square$

Note that the proof of Lemma D.1 does not actually require that $f$ is a `WTP` function; the lemma follows for any $f$ that admits a concave relaxation $\widetilde{f}$.

The following lemma establishes the Lipschitz property of $\widetilde{F}$:

**Lemma D.2.** *Let $f \in \mathcal{F}_{G,M}$ be a `WTP` function, as defined in Eq. (5). Let also $\widetilde{f}$ be its concave relaxation, as defined in Eq. (6) and $\widetilde{F}$ be the corresponding first-stage relaxation associated with $\widetilde{f}$, as defined in Eq. (8). Then, $\widetilde{F}$ is $GM^2\sqrt{n}$-Lipschitz.*

*Proof of Lemma D.2.* Recall the definitions of $\widetilde{f}$ (Eq. (6)) and $\widetilde{F}$ (Eq. (8)):

$$\widetilde{f}(\widetilde{\boldsymbol{y}}) = \sum_{j \in \mathcal{C}} c_j \min\{b_j, \widetilde{\boldsymbol{y}} \cdot \mathbf{w}_j\}, \widetilde{\boldsymbol{y}} \in [0,1]^n \qquad \text{and} \qquad \widetilde{F}(\widetilde{\boldsymbol{x}}) = \max_{\widetilde{\boldsymbol{y}} \in \widetilde{\mathcal{Y}}(\widetilde{\boldsymbol{x}})} \widetilde{f}(\widetilde{\boldsymbol{y}}), \widetilde{\boldsymbol{x}} \in \widetilde{\mathcal{X}}_\ell.$$

Recall that the family $\mathcal{F}_{G,M}$ consists of the `WTP` functions whose index set $\mathcal{C}$ has size at most $G$ and whose parameters $c_j, b_j, w_{ij}$ are upper bounded by $M$.

Fix an $\widetilde{\boldsymbol{x}} \in \widetilde{\mathcal{X}}_\ell$ and let $\widetilde{\boldsymbol{y}} \in \operatorname{argmax}_{\boldsymbol{z} \in \widetilde{\mathcal{Y}}_{\mathcal{M}}(\widetilde{\boldsymbol{x}})} \widetilde{f}(\boldsymbol{z})$. As we show in Appendix E, computing the reward $\widetilde{F}(\widetilde{\boldsymbol{x}})$ can be cast into a linear program. We focus on a dimension $i \in [n]$ and employ a sensitivity analysis to upper bound how much the objective value of this linear program can increase when we increase the $i$-th coordinate of $\widetilde{\boldsymbol{x}}$ by $\epsilon$, i.e. $\widetilde{\boldsymbol{x}}' = \widetilde{\boldsymbol{x}} + \epsilon \cdot e_i$. Notice that $\widetilde{x}_i$ appears only as the right-hand side in the constraint $\widetilde{y}_i \leq \widetilde{x}_i$. Therefore, by increasing $\widetilde{x}_i$ to $\widetilde{x}'_i = \widetilde{x}_i + \epsilon$, the objective function can increase by at most:

$$\sum_{j \in \mathcal{C}} \epsilon |c_j||w_{ji}| \leq \epsilon GM^2.$$

Furthermore, by decreasing a coordinate of $\widetilde{\boldsymbol{x}}$, the objective value can only stay the same or decrease. Therefore, for two points $\widetilde{\boldsymbol{x}}_1, \widetilde{\boldsymbol{x}}_2 \in \widetilde{\mathcal{X}}_\ell$ with $\widetilde{F}(\widetilde{\boldsymbol{x}}_2) \geq \widetilde{F}(\widetilde{\boldsymbol{x}}_1)$ we get that

$$
\begin{aligned}
|\widetilde{F}(\widetilde{\boldsymbol{x}}_2) - \widetilde{F}(\widetilde{\boldsymbol{x}}_1)| = \widetilde{F}(\widetilde{\boldsymbol{x}}_2) - \widetilde{F}(\widetilde{\boldsymbol{x}}_1) \\
\leq GM^2 \cdot \sum_{i \in [n]} (x_{2i} - x_{1i})^+ \\
\leq GM^2 \cdot \sum_{i \in [n]} |x_{2i} - x_{1i}| \\
= GM^2 \cdot \|\widetilde{\boldsymbol{x}}_2 - \widetilde{\boldsymbol{x}}_1\|_1 \\
\leq GM^2 \sqrt{n} \cdot \|\widetilde{\boldsymbol{x}}_2 - \widetilde{\boldsymbol{x}}_1\|_2. \qquad \square
\end{aligned}
$$

**Lemma D.3.** *The diameter of the convex set $\widetilde{\mathcal{X}}_\ell$ is at most $2\ell$.*

*Proof.* For the diameter $D$ of the set $\widetilde{\mathcal{X}}_\ell$ it holds that

$$
\begin{aligned}
D = \max_{\widetilde{\boldsymbol{x}}_1, \widetilde{\boldsymbol{x}}_2 \in \widetilde{\mathcal{X}}_\ell} \|\widetilde{\boldsymbol{x}}_2 - \widetilde{\boldsymbol{x}}_1\|_2 \\
\leq \max_{\widetilde{\boldsymbol{x}}_1, \widetilde{\boldsymbol{x}}_2 \in \widetilde{\mathcal{X}}_\ell} \|\widetilde{\boldsymbol{x}}_2 - \widetilde{\boldsymbol{x}}_1\|_1 \\
\leq \max_{\widetilde{\boldsymbol{x}}_1, \widetilde{\boldsymbol{x}}_2 \in \widetilde{\mathcal{X}}_\ell} \|\widetilde{\boldsymbol{x}}_2\|_1 + \|\widetilde{\boldsymbol{x}}_1\|_1 \\
\leq 2\ell,
\end{aligned}
$$

where for the last inequality we used that $\|\widetilde{\boldsymbol{x}}\|_1 \leq \ell, \ \forall \widetilde{\boldsymbol{x}} \in \widetilde{\mathcal{X}}_\ell$. $\qquad \square$

We are now ready to prove Lemma 5.2, which we restate below for convenience.

**Lemma 5.2.** *Let $\{f_t\}_{t=1}^T \subseteq \mathcal{F}_{G,M}$ be a sequence of `WTP` functions, let $\{F_t\}_{t=1}^T$ be the corresponding reward functions as defined in Eq. (7) and $\{\widetilde{F}_t\}_{t=1}^T$ be the corresponding concave relaxations of the reward functions as defined in Eq. (8). Suppose that $\{\widetilde{\boldsymbol{x}}_t\}_{t=1}^T$ is the sequence of fractional restricted ground sets computed by the `OCO` policy of Algorithm 1. Then,*

$$
\max_{\boldsymbol{x} \in \mathcal{X}_\ell} \sum_{t=1}^T F_t(\boldsymbol{x}) - \sum_{t=1}^T \widetilde{F}_t(\widetilde{\boldsymbol{x}}_t) = \mathcal{O}\left(\ell GM^2 \sqrt{n}\sqrt{T}\right).
$$

*Proof of Lemma 5.2.* The fractional reward functions $\widetilde{F}$ are concave (Lemma D.1) and Lipschitz (Lemma D.2) and the set of actions, $\widetilde{\mathcal{X}}_\ell$ has diameter at most $2\ell$ (Lemma D.3). Therefore, by the standard regret analysis (Theorem 3.1) we get that

$$
\max_{\widetilde{\boldsymbol{x}} \in \widetilde{\mathcal{X}}_\ell} \sum_{t=1}^T \widetilde{F}_t(\widetilde{\boldsymbol{x}}) - \sum_{t=1}^T \widetilde{F}(\widetilde{\boldsymbol{x}}_t) = O(\ell GM^2 \sqrt{n}\sqrt{T}).
$$

The proof of the lemma is completed by noticing that

$$
\max_{\widetilde{\boldsymbol{x}} \in \mathcal{X}_\ell} \sum_{t=1}^T F_t(\widetilde{\boldsymbol{x}}) \leq \max_{\widetilde{\boldsymbol{x}} \in \mathcal{X}_\ell} \sum_{t=1}^T \widetilde{F}_t(\widetilde{\boldsymbol{x}}) \leq \max_{\widetilde{\boldsymbol{x}} \in \widetilde{\mathcal{X}}_\ell} \sum_{t=1}^T \widetilde{F}_t(\widetilde{\boldsymbol{x}}),
$$

because $\widetilde{F}(\widetilde{\boldsymbol{x}}) = F(\widetilde{\boldsymbol{x}}), \ \forall \widetilde{\boldsymbol{x}} \in \mathcal{X}_\ell$ and $\mathcal{X}_\ell \subset \widetilde{\mathcal{X}}_\ell$. $\qquad \square$

# E  Supergradient Computation

In this section, we show how to efficiently compute the supergradient of $\widetilde{F}(\widetilde{\boldsymbol{x}})$ for any $\widetilde{\boldsymbol{x}}$. See Algorithm 2 for the pseudocode. In particular, we are going to prove Lemma 5.3, which we restate below for convenience.

**Lemma 5.3.** *Let $f \in \mathcal{F}_{G,M}$ be a WTP function and let $\widetilde{F}$ be the corresponding fractional reward function, as defined in Eq. (8). For any fractional restricted ground set $\widetilde{\boldsymbol{x}} \in \widetilde{\mathcal{X}}_\ell$, both the time to compute $\widetilde{F}$ and the time to compute a supergradient of $\widetilde{F}$ at $\widetilde{\boldsymbol{x}}$ are polynomial in $n, G$ and $\log M$.*

*Proof of Lemma 5.3.* Consider the concave program:

$$\widetilde{F}(\widetilde{\boldsymbol{x}}) = \max_{\widetilde{\boldsymbol{y}} \in \widetilde{\mathcal{Y}}_\mathcal{M}(\widetilde{\boldsymbol{x}})} \sum_{j \in \mathcal{C}} c_j \min\{b_j, \, \widetilde{\boldsymbol{y}} \cdot \boldsymbol{w}_j\},$$

where the feasible region is given by $\widetilde{\mathcal{Y}}_\mathcal{M}(\widetilde{\boldsymbol{x}}) = \{\widetilde{\boldsymbol{y}} \in [0,1]^n : \widetilde{\boldsymbol{y}} \in \mathcal{P}(\mathcal{M}), \, \widetilde{\boldsymbol{y}} \leq \widetilde{\boldsymbol{x}}\}$. Let $\widetilde{\boldsymbol{y}}^\star$ be an optimal solution to this program. We will show that the dual variables $\boldsymbol{\lambda} = (\lambda_1, \ldots, \lambda_n)$ associated with the constraints $\widetilde{y}_i \leq \widetilde{x}_i$ are a valid supergradient of $\widetilde{F}$ at $\widetilde{\boldsymbol{x}}$.

First, we linearize the concave objective. For each $j \in \mathcal{C}$, we introduce an auxiliary variable $z_j$ and enforce $z_j \leq b_j$ and $z_j \leq \widetilde{\boldsymbol{y}} \cdot \boldsymbol{w}_j$. This yields the equivalent linear program:

$$\max_{\widetilde{\boldsymbol{y}},\boldsymbol{z}} \quad \sum_{j \in \mathcal{C}} c_j z_j$$
$$\text{s.t.} \quad z_j \leq b_j, \quad \forall j \in \mathcal{C},$$
$$z_j \leq \widetilde{\boldsymbol{y}} \cdot \boldsymbol{w}_j, \quad \forall j \in \mathcal{C},$$
$$\widetilde{\boldsymbol{y}} \in \mathcal{P}(\mathcal{M}),$$
$$\widetilde{\boldsymbol{y}} \leq \widetilde{\boldsymbol{x}}.$$

Next, we show how to solve the above linear program using the ellipsoid method. The main requirement is to construct a separation oracle for the feasible set. Crucially, for any $\widetilde{\boldsymbol{y}} \in [0,1]^n$, we can check membership in $\mathcal{P}(\mathcal{M}) = \{\widetilde{\boldsymbol{y}} \in [0,1]^n : \sum_{i \in S} \widetilde{y}_i \leq \text{rank}(S), \, \forall S \subseteq \mathcal{V}\}$ by solving the submodular maximization problem

$$\max_{S \subseteq [n]} \left( \sum_{i \in S} \widetilde{y}_i - r(S) \right),$$

where $r(\cdot)$ is the matroid rank function. This can be done efficiently via the greedy algorithm [Edmonds, 2003, Schrijver, 2003]. If $\widetilde{\boldsymbol{y}} \notin \mathcal{P}(\mathcal{M})$, we return the most violated constraint. The rest of the constraints $z_j \leq b_j, z_j \leq \widetilde{\boldsymbol{y}} \cdot \boldsymbol{w}_j, \widetilde{\boldsymbol{y}} \leq \widetilde{\boldsymbol{x}}$ can be explicitly checked in polynomial time.

We solve the linearized primal LP using the ellipsoid method. During the process, the ellipsoid method generates only a polynomial number of constraints from the exponential family that are necessary to define the optimal face of the feasible region. We then reconstruct a reduced LP consisting of these constraints along with the explicit ones (i.e., $z_j \leq b_j, z_j \leq \widetilde{\boldsymbol{y}} \cdot \boldsymbol{w}_j$, and $\widetilde{\boldsymbol{y}} \leq \widetilde{\boldsymbol{x}}$). This approach follows a standard technique for solving LPs with exponentially many constraints via the ellipsoid method and extracting a reduced LP from the active constraints [Grötschel et al., 2012, Schrijver, 1998].

This reduced LP *has polynomial size and can be solved in polynomial time* using an interior-point method [Nesterov and Nemirovskii, 1994, Boyd, 2004], or efficiently in practice using the simplex method. Both types of solvers return the primal and dual optimal solutions. In particular, we obtain the vector of dual variables $\boldsymbol{\lambda} = (\lambda_1, \ldots, \lambda_n)$ corresponding to the constraints $\widetilde{y}_i \leq \widetilde{x}_i$.

These dual variables constitute a valid supergradient of $\widetilde{F}$ at $\widetilde{\boldsymbol{x}}$ [Boyd, 2004]. That is, for any $\widetilde{\boldsymbol{x}}' \in \mathbb{R}^n$,

$$\widetilde{F}(\widetilde{\boldsymbol{x}}') \leq \widetilde{F}(\widetilde{\boldsymbol{x}}) + \langle \boldsymbol{\lambda}, \, \widetilde{\boldsymbol{x}}' - \widetilde{\boldsymbol{x}} \rangle.$$

We give the pseudocode of the above procedure in Algorithm 2. $\qquad \square$

For completeness, we show that the vector of dual variables $\boldsymbol{\lambda}$ is a valid supergradient. This result is proven in Chapter 5.6 of Boyd [2004], which establishes the correspondence between dual variables and subgradients in parametric convex optimization. Consider the following optimization problem:

$$\max_{\widetilde{\boldsymbol{y}}} \quad f(\widetilde{\boldsymbol{y}})$$
$$\text{s.t.} \quad \widetilde{\boldsymbol{y}} \in \mathcal{P}(\mathcal{M})$$
$$\widetilde{\boldsymbol{y}} \leq \widetilde{\boldsymbol{x}}$$
$$\boldsymbol{0} \leq \widetilde{\boldsymbol{y}} \leq \boldsymbol{1}.$$

**Algorithm 2** Compute Supergradient of $\widetilde{F}(\widetilde{x})$

---

**Require:** Point $\widetilde{x} \in [0, 1]^n$, matroid $\mathcal{M}$
1: Define the LP:

$$\max_{\widetilde{y}, \mathbf{z}} \quad \sum_{j \in \mathcal{C}} c_j z_j$$

$$\text{s.t.} \quad z_j \leq b_j, \quad z_j \leq \widetilde{y} \cdot w_j \quad \forall j \in \mathcal{C},$$
$$\widetilde{y} \in \mathcal{P}(\mathcal{M}), \quad \widetilde{y} \leq \widetilde{x}$$

2: **if** $\mathcal{P}(\mathcal{M})$ has polynomially many constraints **then**
3:      Solve the LP using a solver that returns both primal and dual solutions (e.g., interior-point)
4: **else**
5:      Solve the LP using the ellipsoid method with a separation oracle for $\mathcal{P}(\mathcal{M})$
6:      Record the violated constraints identified by the oracle during optimization
7:      Construct a reduced LP with:
   - the recorded matroid constraints,
   - the explicit constraints: $z_j \leq b_j$, $z_j \leq \widetilde{y} \cdot w_j$, and $\widetilde{y} \leq \widetilde{x}$
8:      Solve the reduced LP using a solver that provides access to the dual variables
9: **end if**
10: **return** dual variables $\lambda = (\lambda_1, \ldots, \lambda_n)$ corresponding to constraints $\widetilde{y}_i \leq \widetilde{x}_i$

---

The Lagrangian is:

$$\mathcal{L}(\widetilde{y}, \lambda, \nu, \mu) = f(\widetilde{y}) + \sum_i \lambda_i (x_i - y_i) + \sum_i \nu_i (1 - y_i) + \sum_{\mathcal{S} \subseteq [n]} \mu_{\mathcal{S}} \mathbf{1}_{\mathcal{S}}(\widetilde{y}),$$

where $\mathbf{1}_{\mathcal{S}}(\widetilde{y})$ is an indicator function that is 1 if $\widetilde{y} \in \mathcal{S}$ and 0 otherwise.

The dual function is:

$$g(\lambda, \nu, \mu) = \min_{\widetilde{y}} \mathcal{L}(\widetilde{y}, \lambda, \nu, \mu).$$

The optimal value of the primal problem is:

$$\widetilde{F}(\widetilde{x}) = \max_{\lambda \geq 0, \nu \geq 0, \mu \geq 0} g(\lambda, \nu, \mu).$$

Now, suppose that we perturb $\widetilde{x}$ to $\widetilde{x}' = \widetilde{x} + u$. The perturbed optimization problem becomes:

$$\max_{\widetilde{y}} \quad f(\widetilde{y})$$

$$\text{s.t.} \quad \widetilde{y} \in \mathcal{P}(\mathcal{M})$$
$$\widetilde{y} \leq \widetilde{x}' = \widetilde{x} + u$$
$$0 \leq \widetilde{y} \leq 1.$$

The Lagrangian for the perturbed problem is:

$$\mathcal{L}'(\widetilde{y}, \lambda, \nu, \mu) = f(\widetilde{y}) + \sum_i \lambda_i \left( (x_i + u_i) - y_i \right) + \sum_i \nu_i (1 - y_i) + \sum_{\mathcal{S} \in [n]} \mu_{\mathcal{S}} \mathbf{1}_{\mathcal{S}}(y),$$

which simplifies to:

$$\mathcal{L}'(\widetilde{y}, \lambda, \nu, \mu) = \mathcal{L}(\widetilde{y}, \lambda, \nu, \mu) + \sum_i \lambda_i u_i.$$

Minimizing over $\widetilde{y}$, the perturbed dual function is:

$$g'(\lambda, \nu, \mu) = g(\lambda, \nu, \mu) + \sum_i \lambda_i u_i.$$

The optimal value of the perturbed problem is:

$$\widetilde{F}(\widetilde{\boldsymbol{x}}') = \max_{\boldsymbol{\lambda} \geq 0, \boldsymbol{\nu} \geq 0, \boldsymbol{\mu} \geq 0} g'(\boldsymbol{\lambda}, \boldsymbol{\nu}, \boldsymbol{\mu}) = \max_{\boldsymbol{\lambda} \geq 0, \boldsymbol{\nu} \geq 0, \boldsymbol{\mu} \geq 0} \left( g(\boldsymbol{\lambda}, \boldsymbol{\nu}, \boldsymbol{\mu}) + \sum_i \lambda_i u_i \right).$$

Thus, we get the inequality:

$$\widetilde{F}(\widetilde{\boldsymbol{x}}') \leq \widetilde{F}(\widetilde{\boldsymbol{x}}) + \sum_i \lambda_i u_i.$$

The dual variables $\boldsymbol{\lambda} = (\lambda_1, \lambda_2, \ldots, \lambda_n)$ are supergradients of the function $\widetilde{F}(\widetilde{\boldsymbol{x}})$ since they satisfy the following inequality for any $\widetilde{\boldsymbol{x}}$ and $\widetilde{\boldsymbol{x}}'$:

$$\widetilde{F}(\widetilde{\boldsymbol{x}}') \leq \widetilde{F}(\widetilde{\boldsymbol{x}}) + \boldsymbol{\lambda}^{\intercal}(\widetilde{\boldsymbol{x}}' - \widetilde{\boldsymbol{x}}),$$

where we substituted $\boldsymbol{u} = \widetilde{\boldsymbol{x}}' - \widetilde{\boldsymbol{x}}$.

# F   Experiments

## F.1   Datasets and Problem Instances

We run experiments on six datasets: datasets `Wikipedia`, `Images`, and `MovieRec` were taken from the experimental setups of Balkanski et al. [2016], Stan et al. [2017]; an influence maximization dataset based on the *Zachary's Karate Club* graph Zachary [1977], `Influence`; and two synthetic datasets: a team formation dataset, `TeamFormation`, and a weighted coverage dataset, `Coverage`.

As in `O2SSM`, the objective is to select, in an online fashion, a set of $\ell$ items corresponding to articles, images, movies, nodes, experts, or sets and receive a reward equal to that of the best $k$ among them. For `Wikipedia`, `Images`, and `MovieRec`, $\ell$ and $k$ were selected based on the experiments of prior work [Balkanski et al., 2016, Stan et al., 2017]. For `Influence` and `TeamFormation` the choice was arbitrary, while for the `Coverage` dataset the choice was adversarial to the `One-stage-OGA` algorithm.

In all cases, we use $n$ to denote the dimension of the decision space, $m$ the number of distinct objective functions, and $T$ the time horizon. For `Wikipedia`, `Images`, and `MovieRec`, we construct a sequence of $T$ objective functions by sampling $T = 4 \cdot m$ functions uniformly at random from the $m$ available functions. For the remaining datasets (`Influence`, `TeamFormation`, and `Coverage`) we set $T = m$ and use each of the $m$ functions exactly once, in sequence.

We summarize the basic information for each dataset in Table 2.

**Wikipedia.** In the `Wikipedia` dataset, the goal is to maintain a set of articles for Machine Learning that are highly relevant for a diverse set of subtopics (e.g., Markov Models, Neural Networks). The function $f_i(S)$ measures how relevant the set of articles $S$ is to the subtopic $i$. In total, we have $n = 407$ articles and $m = 22$ subtopics. We define $f_i(S)$ as the number of Wikipedia pages in the subtopic $i$ that have a link to at least one article in $S$. Note that $f_i$ is a *coverage* function. We set $f_t = f_i$ by sampling a subtopic $i$ uniformly at random. We consider a total of $T = 4 \cdot m = 88$ time steps.

**Images.** In the `Images` dataset, the goal is to maintain a set of images that contain a diverse set of visual elements. We used the VOC2012 dataset Everingham and Winn [2011] where each image contains a set of categories (e.g., chair, car, bird, etc.). For each image $i$ we define $f_i(S)$ as the maximum similarity (minimum $\ell_2$ distance) between the image and an image in $S$. This objective is named Clustered Facility Location in Balkanski et al. [2016] and Exemplar-Based Clustering in Stan et al. [2017]. We have a total of $m = 10$ different functions $f_i$. We set $f_t = f_i$ by sampling an image $i \in [n]$ uniformly at random. We consider a total of $T = 4 \cdot m = 40$ time steps.

**MovieRec.** In the `MovieRec` dataset Harper and Konstan [2015], we consider a movie recommender system. We use the *MovieLens* dataset that contains movies from different genres and user ratings. The goal is to maintain a small set of movies so that every user can find enjoyable movies in this set. For each user $i$, we define the function $f_i(S)$ as follows. Let $r_{ik}$ be the rating given by user $i$ to movie $k$. Let $R_{ij}(S) = \{\max_{k \in S} r_{ik} \mid k \text{ in genre } j\}$ be the highest rating that user $i$ has given to a

movie in genre $j$ that belongs in $S$. Let $w_{ij}$ be the percent of movies in genre $j$ that user $i$ has rated out of all their rated movies. Then,

$$f_i(S) = \sum_{j \in \text{Genres}} w_{ij} R_{ij}(S).$$

Note that $f_i$ is a *facility location* function [Si Salem et al., 2024, Karimi et al., 2017]. We have a total of $m = 100$ different functions $f_i$. We set $f_t = f_i$ by sampling a user $i \in [n]$ uniformly at random. We consider a total of $T = 4 \cdot m = 400$ time steps.

**Team Formation.** In the `TeamFormation` dataset, we consider a roster selection problem. The goal is to maintain a set of $\ell$ individuals (i.e., a roster), so that when a new task $f_t$ arrives, there exists a subset of $k$ individuals within the roster that can handle the task. In order to model the performance of a team when faced with a task $f_t$, we use a quadratic function of the form

$$f_t(\boldsymbol{y}) = \boldsymbol{h}_t^\mathsf{T} \boldsymbol{y} + \frac{1}{2} \boldsymbol{y}^\mathsf{T} \boldsymbol{H}_t \boldsymbol{y}.$$

The linear/modular part of the function captures the strength of each team member, and the quadratic part captures the complementarity/overlaps between their skills. Note that $f_t$ is monotone and submodular, iff $\boldsymbol{H} \leq \boldsymbol{0}$ and $\boldsymbol{h} + \boldsymbol{H}\boldsymbol{y} \geq 0$, for all $\boldsymbol{y} \in \mathcal{Y}_\mathcal{M}(\boldsymbol{x})$. We generate $h_t$ by sampling each coordinate from a Gaussian distribution with mean $\mu = 30$ and standard deviation $\sigma = 20$. Then, we generate $H_t$ by sampling each entry from a Gaussian distribution with mean $\mu = -20$ and standard deviation $\sigma = 10$, while ensuring the all entries are non-positive. If the constraint $\boldsymbol{h}_t + \boldsymbol{H}_t \boldsymbol{y} \geq 0, \forall \boldsymbol{y} \in \mathcal{Y}_\mathcal{M}(\boldsymbol{x})$ is violated, we shrink the entries of $H_t$ until it is satisfied. We construct a total of $m = 50$ functions and consider a total of $T = m = 50$ time steps.

**Influence.** In the `Influence` dataset, we address a two-stage influence maximization problem using *Zachary's Karate Club* as the underlying graph. The dataset assumes the presence of 5 distinct topics (e.g., sports, politics, etc.), with each node in the graph being assigned a subset of these topics. The objective is to maintain a set of $\ell$ nodes such that, at any time step $t$, it is possible to select a subset of $k$ nodes from this set to maximize the influence on a specific topic. We generate a total of $T = 100$ random cascades by sampling each edge of the graph with probability $p = 0.25$. In addition, a topic $i \in [5]$ is selected uniformly at random. A node can influence other nodes only within its connected component in the sampled graph, provided that it has been assigned the selected topic. Note that the influence function is a weighted coverage function, since every node covers all the nodes in its connected component [Kempe et al., 2003].

**HotpotQA.** The `HotpotQA`[9] [Yang et al., 2018] dataset consists of approximately 97K questions along with ground-truth answers and Wikipedia articles that support the ground-truth answer. We create an `O2SSM` instance with the goal of maintaining a small collection of Wikipedia articles that can be used to answer questions from the `HotpotQA` dataset. The universe of elements is chosen to be the union of all Wikipedia articles that are supporting any question and its size is approximately $n = 111K$ articles. For every question $q$ in the `HotpotQA` dataset, let $S(q) \subseteq [n]$ be the set of its supporting articles and $e_{S(q)} \in \{0,1\}^n$ be its characteristic vector. We associate $q$ with a coverage-like `WTP` function $f_q$ defined as:

$$f_q(\boldsymbol{x}) = \min(1, e_{S(q)} \cdot x).$$

For our experiment, we create an arbitrary ordering of the above `WTP` functions and sample $T = 500$ functions in the following way: with probability $0.5$ we pick a uniformly random function (with replacement) from the first 30 functions of the order, otherwise we pick a uniformly random function (with replacement) from the rest.

**Coverage.** In the `Coverage` dataset, we consider a weighted coverage problem with $n = 100$ sets $S_1, \ldots, S_n$. We set $\ell = 10$ and $k = 1$. Consider the $\ell$ sets $S_1, S_2, \ldots, S_\ell$ defined as follows. The first set covers the first $\ell$ elements of the ground set and receives reward $M$ for every covered element:

$$f_1(\boldsymbol{x}) = \sum_{i=1}^{\ell} M \min(1, \boldsymbol{e}_i \cdot \boldsymbol{x}),$$

---

[9] https://huggingface.co/datasets/BeIR/hotpotqa

where $M = 100$ is the value of covering an element of the groundset and $e_i$ is an indicator vector with 1 at coordinate $i$ and 0 everywhere else.

The rest of the sets $i = 2, \ldots, \ell$, cover one, different, element each:

$$f_2(\boldsymbol{x}) = M \min(1, \boldsymbol{e}_{\ell+1} \cdot \boldsymbol{x}),$$

$$\vdots$$

$$f_\ell(\boldsymbol{x}) = M \min(1, \boldsymbol{e}_{2\ell} \cdot \boldsymbol{x}),$$

The online sequence of functions follows the pattern $f_1, f_2, ..., f_\ell$ and then cycles back to $f_1$. We consider a total of 50 cycles, which results in $T = \ell \cdot 50 = 500$ iterations.

This dataset is constructed in such a way so that `One-stage-OGA` algorithm performs poorly. Specifically, when `One-stage-OGA` receives $f_1$ it aggressively updates all coordinates $x_1, ..., x_\ell$ without taking into account that only one element can eventually be used for $f_1$ due to the second-stage cardinality constraint of $k = 1$. Eventually, `One-stage-OGA` converges to a vector having equal weight on the first $2\ell$ coordinates and all other coordinates equal to 0, i.e.,

$$\widetilde{\boldsymbol{x}}_T \approx (\underbrace{{}^1\!/_2, {}^1\!/_2, \ldots, {}^1\!/_2}_{2\ell}, 0, \ldots, 0).$$

After rounding the above vector with Randomized Pipage Rounding we get a restricted dataset that has an expected reward of $M/2$ for the functions $f_2, \ldots, f_\ell$ and approximately $M$ for $f_1$.

The optimal solution in hindsight, however, only puts weight $\ell$ coordinates, using exactly one element for each second-stage objective, due to the inner constraint $k = 1$. That is, the optimal solution in hindsight is

$$\widetilde{\boldsymbol{x}}^\star = (\underbrace{0, \ldots, 0}_{\ell-1}, \underbrace{1, \ldots, 1}_{\ell}, 0, \ldots, 0)$$

and achieves value $M$ for every function.

On the other hand, `RAOCO` eventually converges to a vector that has a uniform mass of 1 element spread across the elements of function $f_1$ and also takes 1 more element for every other function $f_2, \ldots, f_\ell$:

$$\widetilde{\boldsymbol{x}}_T \approx \left( \underbrace{\frac{1}{\ell}, \ldots, \frac{1}{\ell}}_{\ell}, \underbrace{1, \ldots, 1}_{\ell-1}, 0, \ldots, 0 \right).$$

Rounding the above vector with Randomized Pipage Rounding yields a restricted ground set that has expected value $M$ for every function.

The above also explain the result we observe in the `Coverage` experiment in Figure 1. As the time evolves, `RAOCO` improves and converges to the offline optimum in hindsight achieving cumulative average reward approximately $M = 100$, while the cumulative average of `One-stage-OGA` remains constant at approximately $M/2 = 50$.

### F.2 Algorithms

**Online Algorithms.** We provide several implementations of the `RAOCO` algorithm (Algorithm 1). Implementations differ only in the `OCO` policy used in the fractional domain. In particular, we have implemented `RAOCO-OGA`, `RAOCO-FTRL-L2`, `RAOCO-FTRL-H`, where we use *Online Gradient Ascent (OGA)*, *Follow the Regularized Leader (FTRL)* with $L_2$ regularization and entropy regularization, respectively. For each algorithm and dataset, we tested different values for the learning rate $\eta \in \{0.0001, 0.001, 0.01, 0.1, 1, 10\}$ and reported the best results. In Table 3, we report the $\eta$ we used for each algorithm and dataset.

**Online Competitors.** We implement the following online competitors: `Random` selects a set of $\ell$ elements uniformly at random at each timestep; `One-stage-OGA` (1S-OGA) runs `RAOCO-OGA` as if the second stage cardinality constraint was $k = \ell$, i.e., (single stage) online submodular maximization. This is the algorithm presented by Si Salem et al. [2024] for (one-stage) online submodular maximization.

| Dataset | RAOCO-OGA | RAOCO-FTRL-L2 | RAOCO-FTRL-H | 1S-OGA |
|---|---|---|---|---|
| Wikipedia | 0.1 | 0.1 | 1 | 1 |
| Images | 10 | 10 | 10 | 0.1 |
| MovieRec | 0.001 | 0.001 | 0.01 | 1 |
| TeamFormation | 0.01 | 1 | 1 | 0.1 |
| Influence | 0.01 | 0.01 | 0.01 | 0.01 |
| Coverage | 0.01 | 0.001 | 0.01 | 0.1 |

Table 3: Optimal learning rates for each combination of dataset and algorithm.

**Offline Benchmarks.** As offline benchmarks, we implemented the `Continuous-Optimization` (`OFLN-CO`) algorithm proposed by Balkanski et al. [2016], and the `Replacement-Greedy` (`OFLN-RGR`) algorithm proposed by Stan et al. [2017]. For completeness, we present the algorithms here.

`Continuous-Optimization` [Balkanski et al., 2016] follows the relax-then-round paradigm. First, they solve the following program:

$$\max_{\widetilde{\boldsymbol{x}}, \widetilde{\boldsymbol{y}}_1, \ldots, \widetilde{\boldsymbol{y}}_T} \quad \sum_{t=1}^{T} \widetilde{f}_t(\widetilde{\boldsymbol{y}}_t)$$

$$\text{s.t.} \quad \widetilde{\boldsymbol{y}}_t \in \mathcal{P}(\mathcal{M}), \forall t \in [T]$$

$$\widetilde{\boldsymbol{y}}_t \leq \widetilde{\boldsymbol{x}}$$

$$\boldsymbol{0} \leq \widetilde{\boldsymbol{y}}_t \leq \boldsymbol{1}, \forall t \in [T].$$

Let $(\widetilde{\boldsymbol{x}}^\star, \widetilde{\boldsymbol{y}}_1^\star, \ldots, \widetilde{\boldsymbol{y}}_T^\star)$ be the solution of the program. Then, they reduce $\widetilde{\boldsymbol{x}}^\star$ by a factor $(1 - \epsilon')$, where $\epsilon' = 1/k^{1/2-\epsilon}$, for $0 < \epsilon < 1/2$. We set $\epsilon = 1/4$. Next, they round each coordinate of $(1 - \epsilon')\widetilde{\boldsymbol{x}}^\star$ independently to get an integral solution $\boldsymbol{x}$:

$$x_i = \begin{cases} 1, \text{w.p. } \tilde{x}_i^\star \\ 0, \text{otherwise} \end{cases}, \forall i \in [n].$$

If $\boldsymbol{x}$ satisfies the cardinality constraint, we return $\boldsymbol{x}$. Otherwise, if $\boldsymbol{x}$ violates the cardinality constraint ($\sum_{i=1}^{n} x_i \leq \ell$), we return a uniformly random solution.

`Replacement-Greedy` [Stan et al., 2017] is based on a local search approach. First, we start with a series of useful definitions. Let $\Delta_t(e, A) = f_t(A \cup \{e\}) - f_t(A)$ denote the marginal gain of adding $e$ to the set $A$ if we consider the function $f_t$. Let $I(e, A) = \{e' \in A \mid A \cup \{e\} \setminus \{e'\} \in \mathcal{I}\}$ be the set of all elements in $A$ that can be replaced by $e$ without violating the matroid constraint. Let

$$\nabla_i(e, A) = \begin{cases} \Delta_t(e, A), & \text{if } A \cup \{e\} \in \mathcal{I} \\ \max\left(0, \max_{e' \in I(e, A)} f_i(A \cup \{e\} \setminus \{e'\}) - f_i(A)\right), & \text{otherwise} \end{cases}$$

be the gain of either inserting $e$ into $A$ or replacing $e$ with one element of $A$ while keeping $A$ an independent set. Let

$$\text{Rep}_t(e, A) = \begin{cases} \emptyset & \text{if } A \cup \{e\} \in \mathcal{I} \\ \arg\max_{e' \in I(e, A)} f_t(A \cup \{e\} \setminus \{e'\}) - f_t(A) & \text{otherwise} \end{cases}$$

be the element that should be replaced by $e$ to maximize the gain and stay independent. Using the above definitions, we give the pseudocode of `Replacement-Greedy` in Algorithm 3.

**Metrics.** For the online algorithms we measure the cumulative average reward.

$$C_t = \frac{1}{t} \sum_{\tau=1}^{t} F_t(\boldsymbol{x}_t).$$

In order to calculate $F_t(\boldsymbol{x}_t)$ we use Gurobi's integer linear program solver. We also compute the optimal (integral) solution in hindsight (`OPT`):

$$F^\star = \frac{1}{T} \max_{\boldsymbol{x} \in \mathcal{X}_\ell} \sum_{t=1}^{T} F_t(\boldsymbol{x}).$$

**Algorithm 3** `Replacement-Greedy`

---

1: $S \leftarrow \emptyset$
2: $B_t \leftarrow \emptyset$ for all $1 \leq t \leq T$
3: **for** $j = 1$ to $\ell$ **do**
4:     $e^\star \leftarrow \text{argmax}_{e \in \mathcal{V}} \sum_{t=1}^{T} \nabla_t(e, B_t)$
5:     $S \leftarrow S \cup \{e^\star\}$
6:     **for** $t = 1$ to $T$ **do**
7:         **if** $\nabla_t(e^\star, B_t) > 0$ **then**
8:             $B_t \leftarrow B_t \cup \{e^\star\} \setminus \text{Rep}_t(e^\star, B_t)$
9:         **end if**
10:     **end for**
11: **end for**
12: **return** $S$

---

We repeat each experiment 5 times and report the average $C_t$ along with one standard deviation above and below the average depicted via the shaded regions. The sources of randomness are: (i) Randomized Pipage rounding, and (ii) sampling of the objective functions $f_t$ for datasets `Wikipedia`, `Images`, and `MovieRec`.

**Setup.** All the experiments were ran on an Apple M2 Macbook with 16GB RAM. The code is written in Python 3.11.2. We make the code and data used for the experiments publicly available on GitHub. Each experiment runs efficiently on a personal laptop: every problem instance could be computed until the time horizon $T$ in less than one minute.

## G   Additional Experimental Results

### G.1   Exploring the $k$ vs $C_T$ tradeoff

In this section, we explore the tradeoff between $k$ and $C_T$, where $T$ is defined for each dataset at Table 2. Recall that when $\ell = k$, the problem becomes equivalent to one-stage online submodular maximization. We focus on the following algorithms: `RAOCO-OGA`, `One-stage-OGA`. We also report the value of the optimal (integral) solution in hindsight `OPT`. In Figure 2, for each dataset, we show the cumulative average reward $C_T = \frac{1}{T} \sum_{t=1}^{T} F_t(\boldsymbol{x}_t)$ of each algorithm for each value of $k$. For each dataset and each $k$, we run each experiment 5 times and report the average $C_T$ as well as the standard deviation. For each dataset, we fix $\ell$ to the value reported in Table 2.

We observe that for most datasets, varying $k$ does not significantly affect the performance gap between the two algorithms. In many cases, the two algorithms perform similarly, with the `MovieRec` dataset being a notable exception: `RAOCO` consistently outperforms `One-stage-OGA` and its performance remains close to that of the optimal solution in hindsight.

However, in the `Coverage` dataset, we observe that as $k$ increases, the gap between the two algorithms narrows. This is expected due to the dataset's construction: `One-stage-OGA` tends to greedily select multiple elements for the first coverage function $f_1$, whereas `RAOCO` learns to pick one element per function. Consequently, for small values of $k$, many of the elements selected by `One-stage-OGA` cannot be used in the second stage for $f_1$. As $k$ grows, these elements become usable, and the performance of `One-stage-OGA` begins to recover.

### G.2   Exploring the $\ell$ vs $C_T$ tradeoff

In this section, we explore the tradeoff between $\ell$ and $C_T$, where $T$ is defined for each dataset at Table 2. We focus on the following algorithms: `RAOCO-OGA`, `One-stage-OGA`. We also report the value of the optimal (integral) solution in hindsight `OPT`. In Figure 3, for each dataset, we show the cumulative average reward $C_T = \frac{1}{T} \sum_{t=1}^{T} F_t(\boldsymbol{x}_t)$ of each algorithm for each value of $\ell$. For each dataset and each $\ell$, we run each experiment 5 times and report the average $C_T$ as well as the standard deviation. For each dataset, we fix $k$ to the value reported in Table 2 and we vary $\ell$ as a multiple of $k$, that is $\ell = k, 2k, \ldots, 5k$. We observe that varying $\ell$ while keeping the second-stage cardinality constraint $k$ fixed does not significantly affect the performance gap between the two online

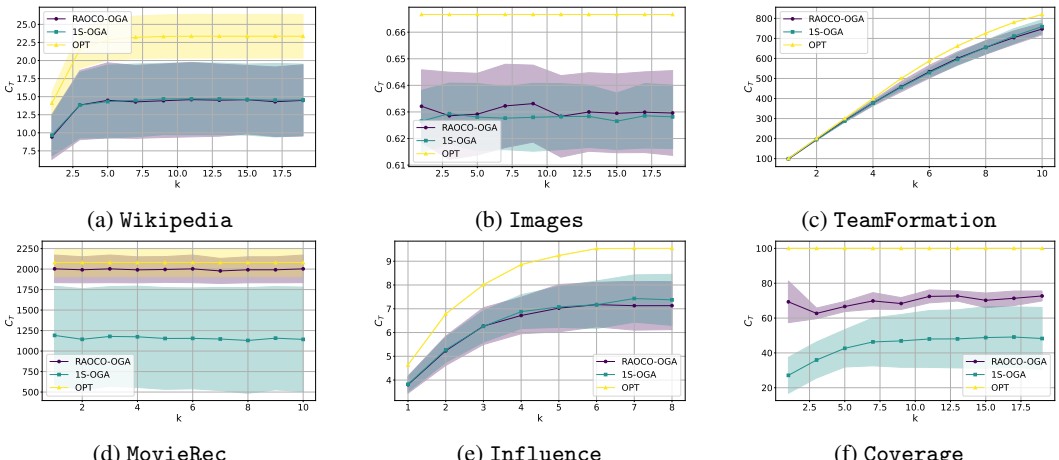

Figure 2: $k$ vs $C_T$ tradeoff for `RAOCO-OGA` and `One-stage-OGA`, where $T$ is defined for each dataset at Table 2. We report the average value of $C_T$ over 5 runs. The shaded regions correspond to one standard deviation above and below the average.

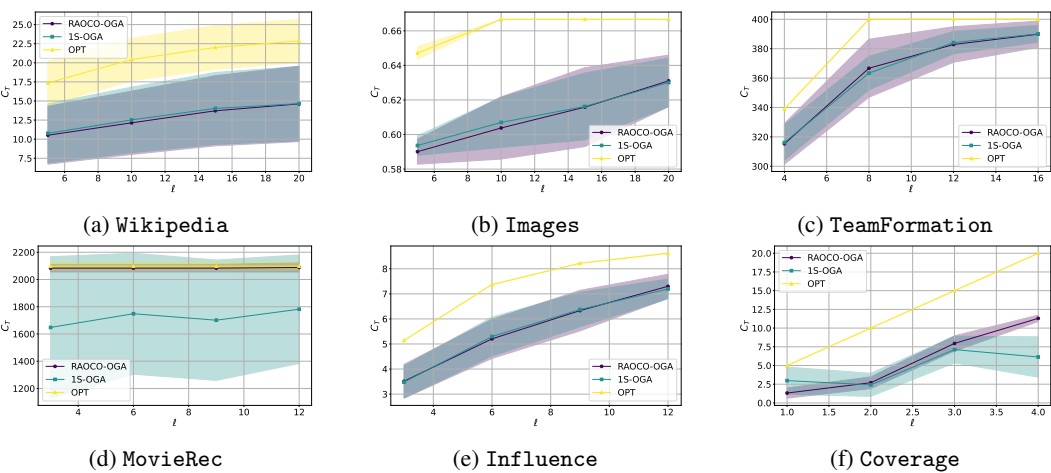

Figure 3: $\ell$ vs $C_T$ tradeoff for `RAOCO-OGA` and `One-stage-OGA`, where $T$ is defined for each dataset at Table 2. We report the average value of $C_T$ over 5 runs. The shaded regions correspond to one standard deviation above and below the average.

algorithms in most datasets. Across these settings, both algorithms exhibit nearly overlapping curves, indicating similar behavior. A notable exception is the `MovieRec` dataset, where `RAOCO` consistently outperforms `One-stage-OGA` and approaches the performance of the optimal fixed restricted ground set in hindsight.

In contrast, the `Coverage` dataset highlights a clear divergence: for small values of $\ell$, the two algorithms perform similarly, but as $\ell$ increases, the performance of `One-stage-OGA` deteriorates while that of `RAOCO` improves. This behavior is expected given the construction of the dataset; `One-stage-OGA` tends to allocate more mass to elements that benefit the first coverage function $f_1$, despite the second-stage constraint allowing only one element to be used. In contrast, `RAOCO` learns to include elements that are useful for different functions, resulting in a higher cumulative reward.

### G.3 Scalability of `RAOCO`

To further assess the scalability of our approach, we experiment on a substantially larger dataset, namely `HotpotQA`[10] [Yang et al., 2018], which consists of $97,852$ questions with ground-truth

---
[10]https://huggingface.co/datasets/BeIR/hotpotqa

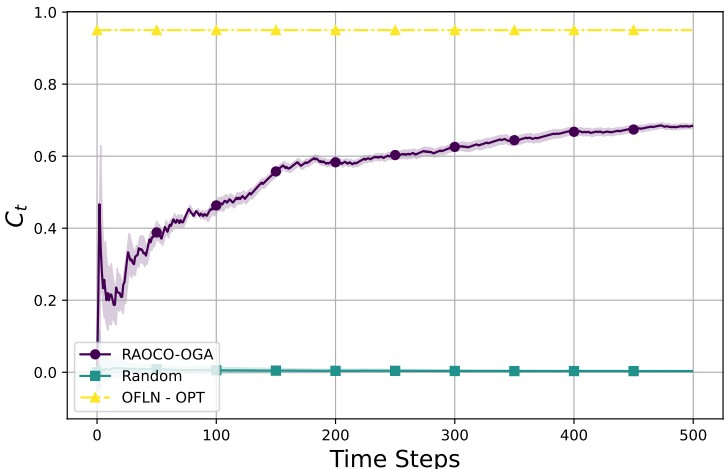

Figure 4: Cummulative reward of `RAOCO-OGA`, `Random` and `OPT` on HotpotQA

answers and supporting Wikipedia articles. We construct an `O2SSM` instance by associating each question with a coverage-like function as described in Appendix F.1. The union of all supporting articles forms a ground set of $111{,}140$ articles. Our goal is to identify a restricted ground set of size $\ell = 150$, from which $k = 4$ articles are selected in the second stage for each question.

We generate $T = 500$ objectives $f_1, \ldots, f_T$ using the following sampling scheme: with probability $0.5$ we select a random question from a fixed pool of $30$ questions, and with probability $0.5$ we select one uniformly at random from the remaining questions. In Figure 4 we compare the cumulative reward of `RAOCO-OGA` against the fractional offline optimum (OPT) and `Random`.

Each iteration of `RAOCO-OGA` requires solving one linear program, which we do via Gurobi (using a free academic license), followed by randomized pipage rounding of the fractional solution. On an Apple M1 MacBook Pro (16GB RAM), each step completes in approximately $30$ seconds on average.

## H   Online Convex Optimization (OCO)

In this section, we describe the the `OCO` policies $\mathcal{P}_{\widetilde{\mathcal{X}}}$ used by our algorithm, i.e., *Online Gradient Ascent (OGA)* and *Follow the Regularized Leader (FTRL)*. Both algorithms enjoy sublinear regret guarantees ($\mathcal{O}(\sqrt{T})$). We refer the interested reader to Hazan [2016] for a more detailed treatment.

### H.1   Online projected (super-)Gradient Ascent (OGA)

*Online projected (super-)Gradient Ascent (OGA)* [Hazan, 2016] comprises the following update rule:

$$\widetilde{\boldsymbol{x}}_{t+1} = \Pi_{\widetilde{\mathcal{X}}_\ell}(\widetilde{\boldsymbol{x}}_t + \eta \boldsymbol{g}_t),$$

where $\eta$ is the learning rate, $\boldsymbol{g}_t$ is a supergradient of $\widetilde{F}_t$ at $\widetilde{\boldsymbol{x}}_t$, and $\Pi_{\widetilde{\mathcal{X}}_\ell}(\cdot)$ is the orthogonal projection on the set $\widetilde{\mathcal{X}}_\ell$.

### H.2   Follow the Regularized Leader (FTRL)

*Follow-the-Regularized-Leader (FTRL)* [Hazan, 2016] comprises the following update rule:

$$\widetilde{\boldsymbol{x}}_{t+1} = \operatorname*{argmax}_{\widetilde{\boldsymbol{x}} \in \widetilde{\mathcal{X}}_\ell} \left\{ \eta \sum_{\tau=1}^{t} \langle \boldsymbol{g}_\tau, \widetilde{\boldsymbol{x}} \rangle - \mathcal{R}(\widetilde{\boldsymbol{x}}) \right\},$$

where $\eta$ is the learning rate, $\boldsymbol{g}_t$ is a supergradient of $\widetilde{F}_t$ at $\widetilde{\boldsymbol{x}}_\tau$, and $\mathcal{R}$ is a strongly convex regularizer.

# I  Randomized Pipage Rounding

In this section, we describe the *Randomized Pipage Rounding* algorithm for the case of cardinality constraints. The algorithm proceeds by iteratively rounding the fractional entries of $\widetilde{x}$ until all components are either 0 or 1. Let $\mathcal{H}_{\widetilde{x}} = \{\widetilde{x}_i \mid 0 < \widetilde{x}_i < 1\}$ be the set of fractional components. At each iteration, two cases are considered:

If only one element in $\widetilde{x}$ is fractional (i.e., $|\mathcal{H}_{\widetilde{x}}| = 1$), we simply round this element to 1. This operation is guaranteed to be feasible.

If multiple fractional elements remain (i.e. $|\mathcal{H}_{\widetilde{x}}| > 1$) , two components $\widetilde{x}_i$ and $\widetilde{x}_j$ are arbitrarily selected from $\mathcal{H}_{\widetilde{x}}$. For these selected indices, we define two quantities $\epsilon_1$ and $\epsilon_2$ as follows:

$$\epsilon_1 = \min(1 - \widetilde{x}_i, \widetilde{x}_j), \quad \epsilon_2 = \min(\widetilde{x}_i, 1 - \widetilde{x}_j)$$

These quantities represent the maximum possible adjustment to $\widetilde{x}_i$ and $\widetilde{x}_j$ while ensuring that the values remain within the range $[0, 1]$. Then a random decision is made based on the ratio of $\epsilon_2/(\epsilon_1 + \epsilon_2)$. Specifically, the following actions are performed:

- With probability $\frac{\epsilon_2}{\epsilon_1 + \epsilon_2}$, we increase $\widetilde{x}_i$ by $\epsilon_1$ and decrease $\widetilde{x}_j$ by $\epsilon_1$.
- With the complementary probability, we decrease $\widetilde{x}_i$ by $\epsilon_2$ and increase $\widetilde{x}_j$ by $\epsilon_2$.

For a more detailed analysis, we defer the interested reader to Chekuri et al. [2009]. We state two lemmas that we use during our analysis in Appendix C.

**Lemma I.1.** *[Chekuri et al., 2009] Let $\widetilde{x} \in \widetilde{\mathcal{X}}$ be a fractional vector and $x = \Xi(\widetilde{x})$ be resulting vector after running Randomized Pipage Rounding on $\widetilde{x}$. Then, the following is true*

$$\mathbf{E}_{\Xi}[x] = \widetilde{x}.$$

**Lemma I.2.** *[Chekuri et al., 2009] Let $\widetilde{x} \in \widetilde{\mathcal{X}}$ be a fractional vector and $x = \Xi(\widetilde{x})$ be resulting vector after running Randomized Pipage Rounding on $\widetilde{x}$. Then, for any submdular function $f$, the following holds*

$$\mathbf{E}_{x \sim \Xi(\widetilde{x})}[f(x)] \geq \mathbf{E}_{x \sim Ind(\widetilde{x})}[f(x)].$$

# J  `RAOCO` in the offline `2SSM` problem

For the sake of completeness, we give a small proof that the ingredients of `RAOCO` can be used to obtain a $c_{\mathcal{M}}(1 - 1/e)$-approximation for the offline `2SSM` problem when the input functions are `WTP`, where $c_{\mathcal{M}} = (1 - e^{-k}k^k/k!)$ if $\mathcal{M}$ is a uniform matroid of rank $k$ and $c_{\mathcal{M}} = (1 - 1/e)$ otherwise.

The aforementioned guarantee improves upon the state-of-the-art for the case cadinality constraints and `WTP` functions. Specifically, while Balkanski et al. [2016] achieve an approximation guarantee of $1 - 1/e - \Omega(1/\sqrt{k})$, our bound converges to the optimal value of $1 - 1/e$ more sharply as $k$ increases, and yields strictly better guarantees for small values of $k$. We illustrate this comparison in Figure 5.

We now prove our offline guarantee formally in the following lemma.

**Lemma J.1.** *Let $[n]$ be a universe of elements and $\mathcal{M}$ be a matroid over $[n]$. Let also $\ell, k \in \mathbb{N}$ and $f_1, \ldots, f_m \in \mathcal{F}_{G,M}$ be the input of the offline `2SSM` problem. There exists a randomized algorithm that runs in polynomial time in $n, m, G$ and $\log M$ and outputs a ground set $x \in \mathcal{X}_\ell$ for which*

$$\mathbb{E}\left[\sum_{t=1}^{m} F_t(x)\right] \geq c_{\mathcal{M}}(1 - 1/e) \cdot \max_{x \in \mathcal{X}_\ell} \sum_{t=1}^{m} F_t(x),$$

*where the expectation is taken over the randomness of the algorithm and $c_{\mathcal{M}} = (1 - e^{-k}k^k/k!)$ if $\mathcal{M}$ is a uniform matroid of rank $k$ and $c_{\mathcal{M}} = (1 - 1/e)$ otherwise.*

*Proof of Lemma J.1.* **Description of the algorithm.** The offline algorithm is the following: first, construct the fractional reward functions $\widetilde{F}_1, \ldots, \widetilde{F}_m$, as defined in Eq. (8), of the given `WTP` functions. Then, solve the following concave program

$$\max_{\widetilde{x} \in \widetilde{\mathcal{X}}_\ell} \sum_{i=1}^{m} \widetilde{F}_i(\widetilde{x}).$$

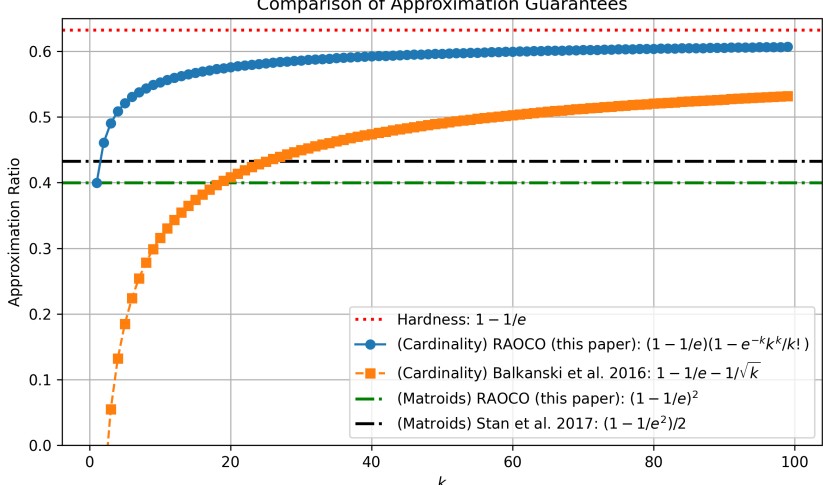

Figure 5: Comparison of approximation guarantees as a function of $k$ for the (offline) 2SSM. The guarantee of both our algorithm and the one by Balkanski et al. [2016] converge to the optimal value $1 - 1/e$ as $k \to \infty$, but our algorithm achieves strictly better guarantees even for finite $k \in \mathbb{N}$. Notably, our guarantee for cardinality constraints also outperforms the state-of-the-art bound for matroid constraints from Stan et al. [2017] for all $k \geq 2$. For general matroids, our guarantee for 2SSM ($(1 - 1/e)^2 \approx 0.40$) is slightly worse than the one by Stan et al. [2017] ($1/2(1 - 1/e^2) \approx 0.43$).

Finally, round the solution of the above program using Randomized Pipage Rounding (see Appendix I).

**Analysis**. We know that each fractional reward function $\widetilde{F}_t$ is concave over $\widetilde{\mathcal{X}}_\ell$ (Lemma D.1) and their supergradient, at any point in $\widetilde{\mathcal{X}}_\ell$, can be computed in polynomial time in $n, G$ and $\log M$ (Lemma 5.3).

Therefore, the above concave program can be solved in polynomial time in $n, m, G$ and $\log M$ using standard first-order or interior point methods [Bubeck, 2015]. Let $\widetilde{\boldsymbol{x}}^*$ be a maximizer of the above program. In the last step, the algorithm rounds $\widetilde{\boldsymbol{x}}^*$ using Randomized Pipage Rounding. Let $\boldsymbol{x} = \Xi\left(\widetilde{\boldsymbol{x}}^*\right)$ be the resulting integral vector.

The analysis is concluded as follows:

$$\mathbf{E}_\Xi \left[ \sum_{t=1}^m F_t(\boldsymbol{x}) \right] = \sum_{t=1}^m \mathbf{E}_\Xi \left[ F_t(\Xi(\widetilde{\boldsymbol{x}}^*)) \right]$$

$$\overset{\text{Lem. } 5.4}{\geq} c_\mathcal{M}(1 - 1/e) \cdot \sum_{t=1}^m \widetilde{F}_i(\widetilde{\boldsymbol{x}}^*)$$

$$= c_\mathcal{M}(1 - 1/e) \cdot \max_{\widetilde{\boldsymbol{x}} \in \widetilde{\mathcal{X}}_\ell} \sum_{t=1}^m \widetilde{F}_t(\widetilde{\boldsymbol{x}})$$

$$\overset{\mathcal{X}_\ell \subset \widetilde{\mathcal{X}}_\ell}{\geq} c_\mathcal{M}(1 - 1/e) \cdot \max_{\widetilde{\boldsymbol{x}} \in \mathcal{X}_\ell} \sum_{t=1}^m \widetilde{F}_t(\widetilde{\boldsymbol{x}})$$

$$\overset{\substack{F_t(\boldsymbol{x}) = \widetilde{F}_t(\boldsymbol{x}) \\ \forall \boldsymbol{x} \in \mathcal{X}_\ell}}{=} c_\mathcal{M}(1 - 1/e) \cdot \max_{\widetilde{\boldsymbol{x}} \in \mathcal{X}_\ell} \sum_{t=1}^m F_t(\widetilde{\boldsymbol{x}}). \qquad \square$$

