# OpenReview forum: "Online Two-Stage Submodular Maximization"
_NeurIPS.cc/2025/Conference — NeurIPS 2025 poster_

### Official Review · Reviewer_SdED · 2025-06-20

**Clarity:** 2
**Significance:** 2
**Originality:** 3
**Rating:** 4
**Confidence:** 2

**Summary:**

This paper addresses the problem of two-stage submodular maximization in an online setting. At each round $t$, the algorithm selects a subset $x_t$ of size $\ell$ in the first stage, after which the submodular objective function $f_t$ is revealed. In the second stage, the algorithm performs submodular maximization over the selected subset. The authors propose an algorithm that leverages techniques from online convex optimization and rounding methods, and demonstrate that it achieves sublinear regret when the objective functions $f_t$ belong to the class of Weighted Threshold Potentials (WTP).

**Questions:**

1. The proofs in the paper appear to rely heavily on techniques from prior work. Could the authors clarify the main technical contributions and specify the key challenges they had to overcome in establishing the primary results?
2. Given the computational overhead introduced by the first stage, could the authors elaborate on why the two-stage online submodular maximization framework is more efficient and advantageous compared to traditional online submodular maximization methods?

**Ethical Concerns:**

["NO or VERY MINOR ethics concerns only"]

**Final Justification:**

I would like to thank the authors for the response. Most of my concerns are addressed, while the practical applicability of the proposed approach still remains unclear to me. Therefore, I maintain my current evaluation.

**Limitations:**

yes.

**Quality:**

3

**Strengths And Weaknesses:**

Strengths
1. The studied problem is well-motivated and significant. In many online scenarios, the large size of the ground set can pose computational challenges for the submodular optimization algorithms. In this sense, the idea of optimization in two-stages, which means first choose a subset and then perform submodular optimization on the restricted set, offers a practical and compelling solution to this issue.
2. The paper is well-written and easy to follow.
3. The result that $\tilde{F}$ is also concave and Lipschitz, and thus the OCO policies can be applied is also interesting.

Weakness:
1. The main weakness of the paper lies in the practicality of the proposed approach. While the two-stage setting is designed to address scenarios where the ground set is large and optimizing over a smaller subset is computationally efficient, the method introduced in this work is itself computationally intensive. Although the authors show that the algorithm runs in polynomial time, the overall computational overhead may limit its applicability in real-world online settings. As such, the practicality of the proposed method for large-scale problems remains questionable.

---

> ### Author Rebuttal · Authors · 2025-07-31
>
> We thank the reviewer for their feedback.
>
> __“Could the authors clarify the main technical contributions and specify the key challenges they had to overcome in establishing the primary results?"__
>
> First of all, we would like to thank the reviewer for the opportunity to re-iterate the main technical contributions of our work. These are split into two “categories”, those regarding the function F(x) and those regarding the two-stage rounding.
>
> __a) On $F(x)$:__
>
> Recall that the way that prior work measures the quality of a restricted groundset $x$ is by associating it to the maximal value a function $f$ can attain when optimized over it.
> This is captured by $F(x) = \max_{\substack{y \leq x, \\ ||y||_1 \leq k}} f(y)$.
> This is non-concave and, as we discuss in the paper, is NP-hard to compute. Thus, contrary to other online settings (like OCO and OSM), we cannot assume oracle access to the reward function when designing an online algorithm.
>
> Instead, we work with the fractional relaxation $\widetilde{F}$ of F, which simply relaxes y to be fractional and replaces $f$ with its fractional relaxation.
>
> An immediate challenge is that the behavior of $\widetilde{F}(x)$ is far from obvious; its input dictates the feasible region of the inner optimization problem and the relation between $x$ and $\widetilde{F}(x)$ is highly non-trivial.
> Surprisingly, we show that $\widetilde{F}$ is concave over the domain of $x$ and, even more importantly, that its super-gradients can be computed efficiently. To do the latter, we write $\widetilde{F}$ as a linear program and show that the dual variables corresponding to the constraints $y_i \leq x_i$ are actually a super-gradient of $\widetilde{F}$.
>
> Both of the above structural insights are non-trivial and essential in order to leverage existing tools from the OCO literature.
>
> __b) On the two-stage rounding:__
>
> Designing a two-stage rounding procedure that handles matroid constraints is particularly challenging. Notably, no prior continuous optimization approach for the offline two-stage submodular maximization problem has succeeded in handling matroid constraints (the work of Balkanski et al. ICML 2016 only had a very simple rounding that works for a single cardinality constraint in the second stage).
>
> The difficulty stems from the fact that the primary tool for rounding under matroid constraints, i.e. contention resolution schemes (CRSs), requires that coordinates are rounded independently at random. This is inherently incompatible with the two-stage setting: in our case, each coordinate $y_i$ in the second-stage solution must satisfy $y_i \leq x_i$​, where $x$ is the rounded first-stage solution. Consequently, the distribution over $y$ is inherently dependent on $x$, and since the first-stage solution $x$ must satisfy a global cardinality or matroid constraint, its coordinates are also correlated. This breaks the independence assumption required by standard CRSs and rules out their direct application.
>
> To overcome this, we leverage a recent characterization theorem by Dughmi (ITCS’20). We notice that Dughmi’s result implies that it is sufficient for the rounding distribution to satisfy a structural property known as submodular dominance (Definition B3) in order for a CRS to exist for this sampling distribution. This implication is hinted at in Dughmi’s work, but we formally state and prove it as Lemma B.7. We believe that this connection could find applications beyond our setting.
>
> Equipped with this insight, we show that, if we apply Randomized Pipage Rounding to the fractional first-stage solution and then sample an "intermediate" second-stage solution conditioned on the first stage (as detailed in Appendix C), the resulting distribution satisfies submodular dominance (Lemma C.4). This crucial property allows us to apply a CRS in the second stage and obtain a feasible solution with provable approximation guarantees.
>
>
> For a higher-level discussion of the two-stage rounding, we refer you to our response to Reviewer SbnU and Appendix C.
>
> We have discussed such issues in Section 5 (lines 239-244) and Appendix C (lines 727-732). We will revise the paper to make sure these technical contributions are more evident.
>
> __“The main weakness of the paper lies in the practicality of the proposed approach. While the two-stage setting is designed to address scenarios where the ground set is large and optimizing over a smaller subset is computationally efficient, the method introduced in this work is itself computationally intensive. … Given the computational overhead […] by the first stage, could the authors elaborate on why the two-stage online submodular maximization framework is more efficient and advantageous compared to traditional online submodular maximization methods?”__
>
> We appreciate the opportunity to discuss this. As the reviewer is asking us to compare two-stage submodular optimization to standard submodular optimization (online or otherwise), this is a question pertaining to the motivation of the two-stage submodular maximization setting in general (online and offline).
>
> It would be good to clarify that a multi-stage approach is indeed often used for computational advantages. For example, many of today’s recommender systems involve multiple stages, progressively reducing the set of items to be recommended. Efficient early-stage recommender algorithms are run initially, and subsequently, more computationally expensive late-stage recommenders are applied at later stages (see, e.g., Liu et al., 2022a; Hron et al., 2021; Zheng et al., 2024; Covington et al., 2016).
>
> However, in many circumstances, the 2-stage approach is inevitable for reasons that do not have to do with computational considerations. One example is caching/pre-fetching (see also Appendix A.2 in the paper):  a system may need to pre-fetch content (e.g., a movie, a large file) to a CDN server close to the user, in advance of the arrival of the user and the expression of their demand. Beyond the computational issue of computing user preferences over a large content catalog when the user arrives (and expresses their demand), the server is restricted to delivering (quickly) only items they have prefetched, and this pre-fetching (that takes time) has to happen before demand manifests.
>
> Appendix A.2 gives several more examples, where decision making (ground set restriction) needs to happen in advance of the manifestation of the demand (the revelation of the objective function). Another simple and intuitive example is putting together a team roster: the players in a team, programmers in a software company, etc., need to be hired before tasks to be performed (playing games or coding a feature requested by clients) are revealed, manifesting any skills required.
>
> Finally, going back to the computational motivation for having multiple stages,  the computational burden/overhead of the first stage is not as large as alluded to by the reviewer.
> In fact, approaching the problem as an online submodular maximization problem, as suggested by the reviewer, will not yield computational dividends as OSM algorithms have similar computational complexity (e.g. Si Salem et al. AAAI 2024).
>
>
> We have made an effort to highlight the scalability/computational complexity of our algorithm in Section 6 and the necessity of the multi-stage approach beyond computational reasons in Appendix A. We will revise these points to make these issues more prominent.

---

> > ### Comment · Reviewer_SdED · 2025-08-07
> >
> > Thank you for addressing my question regarding the technical challenges tackled in the paper. However, I remain somewhat uncertain about the practical applicability of the proposed approach. Therefore, I will maintain my current evaluation.

---

> ### Comment · Area_Chair_n1At · 2025-08-04
> **Please respond to author rebuttals**
>
> Dear Reviewer SdED,
>
> Would you please check if the authors address your major concerns, and if you have further comments?
>
> Regards,
>
> AC

---

### Official Review · Reviewer_SbnU · 2025-06-27

**Clarity:** 3
**Significance:** 3
**Originality:** 3
**Rating:** 4
**Confidence:** 3

**Summary:**

This paper studies the online version of the Two-Stage Submodular Maximization (2SSM) problem, introduced by Balkanski et al. [2016].
2SSM is an interesting problem that aims to select a subset $S$ of core elements from a ground set $V$ (1st stage), such that the expected utility over a collection of submodular functions $\mathcal{F}$ on the selected subset $S$ is maximized (2nd stage).
It can be seen as a special kind of coreset problems.

The main challenge is the objective function:

$$\max_S E_{f \sim \mathcal{F}} [\max_{S' \subseteq S} f(S)],$$

where the maximization over $S'$ is taken *inside* the expectation.
That means each submodular function $f$ is allowed to take on a different subset $S'$ of $S$.
This renders the expectation non-submodular, and difficult to handle.

Balkanski et al. [2016] proposed to optimize a continuous relaxation and do randomized rounding to get a coreset $S$.
This paper follows the same idea but adapts it to the online setting.
Besides, the authors leverage standard algorithms for online convex optimization and more powerful existing rounding techniques to achieve a sublinear regret bound for a specific class of submodular functions.

I think this is a nice paper by combining the ideas of previous works on 2SSM and online convex optimization.
But relying heavily on existing components does reduce the novelty of the paper a bit.
In my opinion, the main discovery is the theoretical properties of the weighted threshold class of submodular functions that enable the analysis.
Moreover, I have a question about the rounding technique (see below).

Representation issues:
- Typo: line 59 "'"
- It would be great if the authors could briefly explain how to comprehend the size of $(1-1/e) (1-e^{-k}k^k/k!)$ compared to $(1-1/e)^2$.

**Questions:**

The authors state that "we design a two-stage randomized rounding procedure, used only in our analysis, that is coupled with the rounding performed by RAOCO."
I am not sure if I understand this correctly.
If the advanced rounding technique is used virtually, then what rounding is actually used in the algorithm?
Does the given algorithm still achieve the claimed regret bound?

**Ethical Concerns:**

["NO or VERY MINOR ethics concerns only"]

**Final Justification:**

I thank the authors for the rebuttal and clarification.
My concerns have been addressed.
I improved my score.

**Limitations:**

yes

**Quality:**

3

**Strengths And Weaknesses:**

Strengths:
1. The authors study an interesting problem, that is relevant to multiple applications such as recommenders.
2. The authors are the first to study the online version of 2SSM.
3. The authors leverage existing tools to achieve a sublinear regret bound for a specific class of submodular functions.

Weaknesses:
1. Concerns about the rounding techniques and the claimed bounds.
2. Given the heavy theoretical machinery, the algorithm is not scalable (up to thousands of elements).

---

> ### Author Rebuttal · Authors · 2025-07-31
>
> We thank the reviewer for their feedback.
>
> __“Explain how to comprehend the size of $(1-1/e)(1-e^{-k}k^k/k!)$ compared to $(1-1/e)^2$”__
>
> Please refer to Figure 4 in Appendix J. In this figure, we plot the aforementioned quantities as well as the approximation ratios of the prior work in the offline problem. We observe that $(1-1/e)(1-e^{-k}k^k/k!)$ is larger than $(1-1/e)^2$ for all $k \geq 1$.  Also, compared with $1-1/e-1/\sqrt{k}$, which is the previous state-of-the-art approximation ratio for a cardinality constraint in the second stage, our guarantee is better for all $k\geq 1$.
>
> __“The authors state that "we design a two-stage randomized rounding procedure, used only in our analysis, that is coupled with the rounding performed by RAOCO." I am not sure if I understand this correctly. If the advanced rounding technique is used virtually, then what rounding is actually used in the algorithm? Does the given algorithm still achieve the claimed regret bound?”__
>
> We thank the reviewer for the opportunity to clarify this point regarding the algorithm’s rounding procedure. The point raised by the reviewer is important and is also related to the technical novelty of our contribution.
>
> There are indeed two rounding procedures used in our analysis:
>
> __1) Rounding used by Alg. 1 (Pipage Rounding)__
> Recall that Algorithm 1 must output a subset of size $\ell$ (the “restricted groundset”). To do so, it runs an Online Convex Optimization (OCO) policy (e.g., FTRL or OGA) to obtain fractional values for the elements (a “fractional restricted groundset”) and then applies Randomized Pipage Rounding to produce the final integral restricted groundset (see Appendix I). This is, in fact, the only rounding procedure used by the algorithm. The rounding is very simple, efficient, and easily implementable. No other rounding needs to be implemented to execute the algorithm; __the regret bound stated in the paper (Theorem 5.1) is correct, and is indeed achieved using exactly this procedure.__
>
> __2) A second rounding, introduced only in the regret analysis__
> We stress that this second rounding __is not used by the algorithm, and is only used in the analysis as a tool to prove a lower bound__. We describe this in lines 246-250 & 717-721, but we will attempt to clarify this further here.
>
> Observe that, by Lemma 5.2, the fractional restricted groundset that the OCO policy returns is “good enough” : if the algorithm could output that, it would have no regret.
>
> However, as discussed above, the algorithm needs to round this restricted groundset, say $\widetilde{x}$, to an integral groundset $x$. It does so by running Randomized Pipage Rounding. In order to prove that the algorithm’s output has sublinear $\alpha$-Regret, we need to prove that the “value” of the rounded groundset is at least an $\alpha$ factor of the “value” of the original fractional groundset $\widetilde{x}$.
>
> The “value” of a restricted groundset is measured by the maximal value a function $f$ can attain when optimized over the restricted groundset, i.e. $F(x) = \max_{\substack{y \leq x, \\ ||y||_1 \leq k}} f(y)$.
>
> As $\widetilde{x}$ is rounded to $x$, it is highly nontrivial to track what is the relation between $F(\widetilde{x})$ and $F(x)$, as the input of $F$ changes the feasible region of the inner maximization problem.
>
> To prove the desired relation, we do the following: Fix $\widetilde{x}$, the algorithm’s output $x$, and a function $f$.
>
> Let $y^{\*}$ be the maximal solution of $f$ on the restricted groundset, i.e. $F(\widetilde{x}) = f(y^{\*})$.
>
> We construct a new solution $y’$ that lies in the rounded restricted groundset $x$, for which it holds that $f(y’) \geq \alpha \cdot f(y^*) = \alpha \cdot F(\widetilde{x})$.
>
> Note that the maximal value of $f$ in the groundset $x$, namely $F(x)$ (!),  is going to be at least $f(y’)$, because $y’$ is just one feasible selection.
>
> Therefore this concludes our proof that: $F(x) \geq f(y’) \geq \alpha \cdot f(y^*) \geq \alpha F(\widetilde{x})$.
>
>
> This is our proof of Lemma 5.3 on a high level. In order to make this proof possible, we need to come up with a way to construct $y’$ given $\widetilde{x}, x$ and $y^*$. This is exactly what the second ”virtual” rounding does.
>
>
>
> This rounding first constructs an intermediate (possibly infeasible) integral solution $\hat{y}$​ and then applies a contention resolution scheme (CRS) to obtain a feasible solution $y’$. We describe this procedure in detail in Appendix C. Lemmas C.1 through C.5 build up to the guarantee that $f(y’) \geq \alpha f(y^*)$ as we described above.
>
>
> This argument is central to proving Lemma 5.4 and, ultimately, our regret bounds.
>
> To summarize, __the second rounding, as technical as it may be, is not used by the algorithm and is only used in the analysis as a tool to prove a lower bound.__
>
> We hope that this clarifies the role and purpose of the two-stage rounding in our analysis.
>
> __“Given the heavy theoretical machinery, the algorithm is not scalable (up to thousands of elements).”__
>
> All components of our algorithm to scale to larger datasets. The online optimization routines (FTRL, Gradient Ascent) compute gradients efficiently in polynomial time via linear programming, and the rounding step (pipage rounding) runs in $O(n)$ time for cardinality constraints. As analyzed in detail in Section 5 (“Computational Complexity”), these components do not pose practical bottlenecks.
>
> To further strengthen the paper, we will include a large-scale experiment in the final version. In this experiment, based on the HotpotQA dataset __with a ground set of $10^5$ articles__, our algorithm selects, at each iteration, a set of articles intended to “cover” an unknown topic revealed only after the selection. Despite the large ground set and the online nature of the task, __each iteration, i.e., solving a convex program over the dataset and rounding the result via Randomized Pipage Rounding, completes in under 30 seconds__, demonstrating the practical scalability of our approach.
> Notably, the algorithm of Stan et al. needs more than 15 minutes to find a restricted groundset in the above dataset, even for a small number of second-stage functions (e.g. less than 10)!
>
>
> We include a table below showing the cumulative reward of our algorithm, the fractional optimal set in hindsight, and Random (i.e., selecting the restricted groundset randomly).
>
>
> | Algorithm                   | t = 0 | t = 20 | t = 40 | t = 60 | t = 80 | t = 100 | t = 120 | t = 140 | t = 160 | t = 180 | t = 200 |
> |-----------------------------|-------|--------|--------|--------|--------|---------|---------|---------|---------|---------|---------|
> | Fractional Offline Optimal  | 1.0   | 1.0    | 1.0    | 1.0    | 1.0    | 1.0     | 1.0     | 1.0     | 1.0     | 1.0     | 1.0     |
> | RAOCO OGA                   | 0.0   | 0.571  | 0.659  | 0.590  | 0.593  | 0.594   | 0.612   | 0.596   | 0.578   | 0.564   | 0.560   |
> | Random                      | 0.0   | 0.0    | 0.0    | 0.0    | 0.0    | 0.0     | 0.0     | 0.007   | 0.006   | 0.006   | 0.005
>
>
> Table: Cumulative reward of different algorithms over HotpotQA dataset, as a function of online steps
>
>
>
> __Typos:__
>
> Thank you for spotting the typo in line 59.  We will fix this and will correct all other typos pointed out by reviewers.

---

> > ### Comment · Reviewer_SbnU · 2025-08-03
> >
> > I thank the authors for the rebuttal and clarification.
> > My concerns have been addressed.
> > I improved my score.

---

> > > ### Author Response · Authors · 2025-08-05
> > >
> > > We would like to thank the reviewer for re-evaluating the paper and for raising their score.

---

### Official Review · Reviewer_ToR4 · 2025-07-02

**Clarity:** 3
**Significance:** 1
**Originality:** 2
**Rating:** 4
**Confidence:** 3

**Summary:**

This paper focuses on the two stage submodular maximization problem in different settings and subject to cardinality and matroid constraint.

The first result in this paper follows the general steps of Balkanski et. al.   and slightly improves the approximation ratio in the offline setting. The improvement is mainly for small values of k.

The authors also consider the online setting and provide a 0.4 approximate algorithm for the general matroid constraint setting but only for a specific class of functions (WTP). In this online setting the functions are revealed in an online fashion and not the elements.

**Questions:**

1- Can the proposed algorithms be applied on much larger datasets?
2- The motivation behind the online setting is not strong in my opinion, can you elaborate on its importance and applicability?

**Ethical Concerns:**

["NO or VERY MINOR ethics concerns only"]

**Final Justification:**

I missed the point about the assumptions on the submodular functions. I also appreciate the more details on experiments and applications. The datasets considered here are still quite small compared to the large ones experimented on in submodular maximization literature (by an order of magnitude or so). I still have some concerns regarding the setting and techniques.

**Limitations:**

yes

**Quality:**

3

**Strengths And Weaknesses:**

The main strength of the paper is overcoming the challenge of computing the function F. This function is not submodular and also it cannot be computed in polytime. The techniques used here are interesting but seem to be limited to this specific problem.

The online setting considered in this work is not that interesting and the motivations provided for it are not strong enough. This combined with the marginal improvements in the offline setting raises concern about the importance and strength of this paper. This is also not for general submodular functions and only for WTP. Most of the previous works cited in this paper (see Table 1) are for general submodular function and in more practical settings.

The experimental section requires significant changes. Currently it is not possible to conclude that the proposed algorithms can be used in practice. The main reason is that the datasets considered are tiny.

---

> ### Author Rebuttal · Authors · 2025-07-31
>
> We thank the reviewer for their feedback. However, we believe that there is a fundamental misunderstanding regarding the main contributions and focus of our work.
>
> __“This paper focuses on the two stage submodular maximization problem [...] The first result of this paper [...] in the offline setting. [...] The authors ALSO consider the online setting.”__
>
> The paper’s primary focus is the __online__ two-stage submodular maximization (O2SSM) problem, where the second-stage functions are revealed sequentially. Our main contribution is the first algorithm with sublinear $\alpha$-regret guarantees for this setting. The offline results (including the improved approximation for small $k$) are a (fortunate) by-product of our online framework, but not the central contribution. The online version of this problem was explicitly left as an open question by Stan et al. (ICML 2017), making our result the first to close this gap.
>
> __“The online setting considered in this work is not that interesting and the motivations provided for it are not strong enough”__
>
> The online setting we study is __more natural and realistic than the offline two-stage submodular maximization problem in nearly all applications considered in prior work.__
>
> Consider the recommender system case, for instance, in which a platform must proactively maintain and update a restricted catalog $S$ (of size $\ell$) __before user requests materialize__. As a result, as new users, preferences, and content (movies, videos, or music) arrive continuously, __user utility functions are not known ahead of time__, and decisions on how to maintain this restricted catalog need to be made in advance.
>
> Note that this setting, and the pertinent challenge, describe the (offline) 2SSM problem, as studied by Stan et al. (ICML 2017) and Balkanski et al. (ICML 2016). How is the challenge handled in the offline setting? Stan et al. and Balkanski et al. __assume that a distribution over user utilities is known__. In the online setting, __we remove this assumption: we do not assume that utilities come from a known distribution__. Instead, utilities are picked in an arbitrary (potentially adversarial) fashion. Clearly, __making fewer assumptions than the offline setting makes the online setting more practical and better motivated, not less.__
>
> Providing guarantees in the online setting, and thereby lifting distributional assumptions, is a common theme in online algorithms; it is in fact the main reason they are studied, and of great interest to the machine learning community. Papers on online convex optimization and online submodular maximization have appeared in NeurIPS (Zhang et al. 2019; Harvey et al. 2020; Zhao et al. 2020; Ghai et al. 2022; Nie et al. 2024; Xu et al. 2024), ICML (Fotakis et al. 2021; Gergatsouli et al. 2022; Wan et al. 2023;), and COLT (Ito et al. 2024; Kozachinskiy et al. 2024; Appel et al. 2025; Mhammedi 2025), to name a few. The online setting is thus both of interest to the community as well as the original authors of (offline) 2SSM.
>
> Note that, precisely because they make no distributional assumptions, online problems are more technically challenging than their offline counterparts. This is another reason why online submodular maximization and online convex optimization are active areas of research, and why Stan et al. themselves pose online 2SSM as an open problem.
>
> In conclusion, while the offline two-stage framework has been justified in prior work, our online framework strictly generalizes it, __removing the strong assumption that a distribution over objectives is known in advance__. The extension to the online setting was also highlighted as an open problem by Stan et al. __Finally, all other reviewers recognized the problem’s significance and strong motivation.__
>
> __“This is also not for general submodular functions and only for WTP. Most of the previous works cited in this paper (see Table 1) are for general submodular function and in more practical settings.”__
>
> We stress that previous works in Table 1 are for the __offline__ 2SSM problem, __not__ the online O2SSM problem. __Our work is the first to provide any guarantees in the online setting for any submodular functions (WTP or otherwise)__. We also stress that our online setting is a more challenging and realistic setting than the offline setting: the offline setting requires distributional assumptions; the online setting does not. Hence, we do not agree that the offline setting is more practical.
>
> Finally, Weighted Threshold Potentials (WTPs) form a broad subclass of monotone submodular functions that subsumes all applications studied in prior two-stage maximization work, including Weighted Coverage, Influence Maximization, Facility Location, and Team Formation. __Our results directly apply to all applications presented as a motivation by Balkanski et al. (ICML 2016) and Stan et al. (ICML 2017), despite focusing on WTPs.__ Note that several works on submodular maximization focus on restricted classes of functions, both in the offline and online setting (Karimi et al. NeurIPS’17, Bai et al. NeurIPS’18, Bian et al. ICML’17, Hassani et al. NeurIPS’17, etc.)
>
> __"“The techniques used here are interesting but seem to be limited to this specific problem.”__
>
> We stress that we solve O2SSM for arbitrary WTP functions and general matroid constraints in the second stage, both of which include many important instances, as listed in Appendix A.2.
>
> Our techniques follow the general template of solving a continuous problem first, and then identifying an appropriate rounding. We design a novel scheme for 2-stage Submodular Maximization, when the feasible set of the second-stage maximization is restricted by a matroid constraint and depends on a first-stage decision. Such techniques are of broader interest and often find applications in different problems. As we also point out in the response to Reviewer SdED, the challenge we face lies in the incompatibility between standard contention resolution schemes (CRSs), which are essential for rounding continuous relaxations under matroid constraints, and the two-stage nature of the problem. Specifically, traditional CRSs require independent rounding, which is impossible when rounding across nested feasible sets (i.e., subsets selected in the first stage and constrained in the second).
>
> To address this, we develop a two-stage rounding scheme based on a recent characterization theorem by Dughmi (ITCS 2020) concerning non-product CRSs. To our knowledge, this is the first time such a tool has been applied in a multi-stage stochastic setting. We believe this connection opens up new directions for multi-stage optimization under matroid and combinatorial constraints. Moreover, the rounding framework we introduce is not just an ad hoc fix: it may lead to closing the current approximation gap in offline 2SSM. Thus, we view this contribution as advancing rounding for structured, non-product settings, with potential applicability beyond 2SSM.
>
> __“The experimental section requires significant changes. [...] the datasets considered are tiny.”__
>
> We recognize that our main contribution is theoretical, providing the first formal sublinear $alpha$-regret guarantee for O2SSM. Our experimental evaluation is intended to validate our algorithm’s behavior and to compare it directly with prior work, rather than to establish scalability. To ensure a fair comparison, __we follow the experimental setup of prior work__ [Balkanski et al., ICML’16; Stan et al., ICML’17], __using the same datasets__ (Wikipedia, Images, and MovieRec), to directly benchmark against established baselines. To highlight the broad applicability of our framework, we also include the Influence and TeamFormation datasets, while the Coverage dataset is designed to illustrate the benefits of two-stage optimization over one-stage formulations.
>
> Nevertheless, to address the reviewer’s concerns, we have conducted a new experiment on a dataset whose groundset is two orders of magnitude bigger than the ones tested by our paper and prior work. We discuss this in the next paragraph.
>
> __“Can the proposed algorithms be applied on much larger datasets?”__
>
> All components of our algorithm scale to larger datasets. The online optimization routines (FTRL, Gradient Ascent) compute gradients efficiently in polynomial time via linear programming, and the rounding step (pipage rounding) runs in $O(n)$ time for cardinality constraints. As discussed in Section 5 (“Computational Complexity”), these components do not cause bottlenecks. To strengthen the paper, we include a large-scale experiment based on the HotpotQA dataset with __a ground set of $10^5$ articles__. Our algorithm selects, at each iteration, a set of articles that “cover” an unknown topic revealed after the selection. Despite the large ground set, __each iteration solving a convex program over the dataset and rounding the result via Randomized Pipage Rounding, completes in under 30 seconds__, demonstrating the practical scalability of our approach. Notably, __the algorithm of Stan et al. needs more than 15 minutes to find a restricted groundset in the above dataset, even for a small number of second-stage functions (e.g. less than 10)!__
>
> We include a table below showing the fractional optimal set in hindsight, the cumulative reward of our algorithm, and Random.
>
> | Alg.   | t = 0 | t = 20 | t = 40 | t = 60 | t = 80 | t = 100 | t = 120 | t = 140 | t = 160 | t = 180 | t = 200 |
> |---|----|----|----|----|----|---|---|----|----|---|---|
> | Fractional Offline Optimal | 1.0 | 1.0  | 1.0  | 1.0  | 1.0   | 1.0    | 1.0   | 1.0  | 1.0  | 1.0   | 1.0  |
> | RAOCO OGA | 0.0   | 0.571  | 0.659  | 0.590  | 0.593  | 0.594   | 0.612   | 0.596   | 0.578   | 0.564   | 0.560   |
> | Random   | 0.0   | 0.0    | 0.0    | 0.0    | 0.0    | 0.0     | 0.0     | 0.007   | 0.006   | 0.006   | 0.005   |
>
> Table: Cumulative reward vs. time over HotpotQA dataset.

---

> > ### Comment · Reviewer_ToR4 · 2025-08-04
> >
> > I want to thank the authors for their detailed rebuttal.
> >
> > I missed the point about the assumptions on the submodular functions. I also appreciate the more details on experiments and applications. The datasets considered here are still quite small compared to the large ones experimented on in submodular maximization literature (by an order of magnitude or so).  I still have some concerns regarding the setting and techniques.
> >
> > I raised my score accordingly.

---

> > > ### Author Response · Authors · 2025-08-05
> > >
> > > We sincerely thank the reviewer for re-evaluating the paper and for raising their score.
> > >
> > > We appreciate the reviewer’s concern on dataset size. It is worth reiterating that the two-stage submodular maximization problem is fundamentally different from the classical one-stage variant (i.e., submodular maximization), both in structure and motivation. As noted in prior work (Stan et al., ICML 2017; Balkanski et al., ICML 2016) and detailed in our paper, this two-stage setting is well-motivated by several real-world applications. Our experimental setup exactly follows these prior works, using the same benchmark datasets, and adding more. The common datasets have also appeared in the (one-stage) online submodular maximization literature (see, e.g., Si Salem et al. AAAI 2024).
> > >
> > > That said, we fully agree on the importance of demonstrating scalability. To that end, in the final version of the paper, we will include a new experiment on a dataset that is two orders of magnitude larger than those used in previous two-stage submodular maximization studies. We would be very happy to add any additional datasets that the reviewer would like to recommend.

---

> ### Comment · Area_Chair_n1At · 2025-08-04
> **Please respond to author rebuttals**
>
> Dear Reviewer ToR4,
>
> Would you please check if the authors address your major concerns, and if you have further comments?
>
> Regards,
>
> AC

---

### Official Review · Reviewer_2YLy · 2025-07-03

**Clarity:** 3
**Significance:** 3
**Originality:** 4
**Rating:** 5
**Confidence:** 5

**Summary:**

The (stochastic) submodular maximization problem aims to maximize an objective subject to some constraint defined over a ground set, where the objective is picked u.a.r. from a collection of monotone submodular functions. The two-stage submodular maximization problem (2SSM) aims to find a subset of size at most $\ell$ from the ground set such that when restricted to this subset, the previous submodular maximization problem attains a high maximal value in expectation. This paper studies the online version of 2SSM. For weighted threshold potential functions, an important subclass of monotone submodular functions, it provides an online algorithm that achieves an $O(\sqrt{T})$ $(1-1/e)^2$-regret under a matroid constraint and the ratio is slightly improved for uniform matroid.

**Questions:**

The (offline) 2SSM admits a better approximation ratio for coverage functions, a subclass of weighted threshold potential functions. Can one obtain the same ratio for online 2SSM by applying the approach of the paper?

**Ethical Concerns:**

["NO or VERY MINOR ethics concerns only"]

**Final Justification:**

Thanks for the authors' reply. My concerns have been addressed. I will retain my score.

**Limitations:**

yes

**Quality:**

4

**Strengths And Weaknesses:**

Quality. The paper is technically sound. The main result is supported by theoretical analysis. Besides, the performance of the proposed algorithm is validated by experiments on both real and synthetic datasets.

Clarity. The paper is organized well. The algorithm involves several components. The paper thoroughly explains each of them and lists the corresponding lemmas used to prove the final theorem.

Significance. The paper is the first to consider the online version of the well-studied 2SSM problem and attains sublinear regret for the problem. Though built on submodular maximization, the problem is not a standard online submodular maximization (OIM) problem. Therefore, the result extends previous results on OIM to a non-submodular setting. This could be of interest to researchers. On the other hand, the approach is somewhat limited since it works only for a subclass of submodular functions. Besides, it looks difficult to extend the algorithm to general submodular functions since they do not admit computable concave extensions.

Originality. The paper exploits an algorithmic framework developed for online submodular maximization in a non-standard submodular maximization problem. Although the overall algorithmic framework derives from prior work, this paper contains sufficient technical contributions, as the successful application of the framework requires key insights and technical lemmas.

---

> ### Author Rebuttal · Authors · 2025-07-31
>
> We kindly thank the reviewer for their detailed feedback. We appreciate their positive view of the paper’s contributions.
>
> __“The (offline) 2SSM admits a better approximation ratio for coverage functions, a subclass of weighted threshold potential functions. Can one obtain the same ratio for online 2SSM by applying the approach of the paper?”__
>
> Thank you for bringing up this point. This requires a somewhat subtle comparison to prior work, that we clarify below, and can add to the paper as needed.
>
> To the best of our knowledge, the only results that pertain to coverage functions in the context of two-stage submodular optimization appear in Balkanski et al. (ICML 2016). There are two guarantees provided by Ballkanski et al. in the context of coverage functions.
>
> - First, along with their algorithm for general submodular functions, they propose a local search method that offers a $1/2 (1− 1/e) \approx 0.31$-approximation for the special case of coverage functions, under cardinality constraints.
> For large $k$, this guarantee is worse than the  $1-1/e-1/\sqrt{k}$ guarantee of the continuous optimization algorithm also proposed by Balkanski et al. The latter algorithm also works for general submodular functions. The $ \approx 0.31$ guarantee is also worse than the  $(1-1/e^2)/2 \approx 0.43$ guarantee provided by Stan et al. (ICML 2017), which also works for arbitrary submodular functions over general matroid constraints.  This is why we did not include the local search algorithm of Balkanski et al. in our literature review.
>
> Our framework already achieves the state-of-the-art approximation ratio ($(1-1/e)(1-e^{-k}k^k/k!)$)  both in the offline and online settings, for WTP functions and thus also for coverage functions, in the case of cardinality constraints. For more details, we refer the reviewer to Appendix J and Fig. 4. In particular, our guarantee is better than the guarantee by Stan et al. (ICML 2017) for $k \geq 2$. It is also better than the
> $1-1/e-1/\sqrt{k}$ guarantee of Balkanski et al. for all $k \geq 1$.
>
> - Second, Balkanski et al. also point out that, for constant $k$ , a variant of their local search algorithm yields an improved $0.5$ approximation, still for coverage functions. This algorithm has a running time that is exponential in $k$. Note that our guarantee is better than $0.5$ for all $k > 3$, and does not require that $k$ is a constant. In fact, our algorithm’s running time is independent of $k$.
>
> In summary, the guarantees for coverage in the offline case, under cardinality constraints, are not better than the ones we obtain here (either online or offline).
>
> In the case of general matroid constraints, there are no special guarantees for coverage functions. The state-of-the-art approximation ratio in the offline problem is by Stan et al. (ICML 2017), who achieve a $(1-1/e^2)/2 \approx 0.43$ approximation, for general submodular functions. In the case of general matroid constraints, the offline version of our algorithm yields a $(1-1/e)^2$ guarantee.  We discuss how our (offline) guarantee relates to prior work over matroid constraints in Table 1 and Appendix J. Note that the local search algorithm of Stan et al. heavily relies on the knowledge of all second-stage functions for its decisions, while our algorithm works in the online setting and operates without prior knowledge of second-stage functions.

---

> > ### Comment · Reviewer_2YLy · 2025-08-08
> >
> > Thanks for the authors' reply. My concerns have been addressed. I will retain my score.

---

> ### Comment · Area_Chair_n1At · 2025-08-04
> **Please respond to author rebuttals**
>
> Dear Reviewer 2YLy,
>
> Would you please check if the authors address your major concerns, and if you have further comments?
>
> Regards,
>
> AC

---

### Decision · Program_Chairs · 2025-09-17

**Decision:**

Accept (poster)

**Comment:**

This paper initiates the study of online version of two-stage submodular maximization, and gives regret results for WTP functions. The result is competitive, and the techniques are solid. The main weakness seems to be the experiments, and the formulation of the problem may also be argued. As raised by some reviewers, a potential flaw is the scalability of the algorithm, but this seems clarified/resolved during the rebuttal. The review is overall positive, and I recommend to accept the paper.